# Variance-Aware Off-Policy Evaluation with Linear Function Approximation

**Yifei Min**[*]
Department of Statistics and Data Science
Yale University
CT 06511
yifei.min@yale.edu

**Tianhao Wang**[*]
Department of Statistics and Data Science
Yale University
CT 06511
tianhao.wang@yale.edu

**Dongruo Zhou**
Department of Computer Science
University of California, Los Angeles
CA 90095
drzhou@cs.ucla.edu

**Quanquan Gu**
Department of Computer Science
University of California, Los Angeles
CA 90095
qgu@cs.ucla.edu

## Abstract

We study the off-policy evaluation (OPE) problem in reinforcement learning with linear function approximation, which aims to estimate the value function of a target policy based on the offline data collected by a behavior policy. We propose to incorporate the variance information of the value function to improve the sample efficiency of OPE. More specifically, for time-inhomogeneous episodic linear Markov decision processes (MDPs), we propose an algorithm, `VA-OPE`, which uses the estimated variance of the value function to reweight the Bellman residual in Fitted Q-Iteration. We show that our algorithm achieves a tighter error bound than the best-known result. We also provide a fine-grained characterization of the distribution shift between the behavior policy and the target policy. Extensive numerical experiments corroborate our theory.

## 1 Introduction

Reinforcement learning (RL) has been a hot spot in both theory and practice in the past decade. Many efficient algorithms have been proposed and theoretically analyzed for finding the optimal policy adopted by an agent to maximize the long-term cumulative rewards. In contrast to online RL where the agent actively interacts with the environment, offline RL (a.k.a., batch RL) [25, 24] aims to extract information from past data and use this information to learn the optimal policy. There has been much empirical success of offline RL in various application domains [4, 6, 37, 40, 36].

Among various tasks of offline RL, an important task is called *off-policy evaluation* (OPE), which evaluates the performance of a target policy $\pi$ given offline data generated by a behavior policy $\bar{\pi}$. Most existing theoretical works on OPE are in the setting of tabular MDPs [34, 26, 11, 16, 45, 47–49], where the state space $\mathcal{S}$ and the action space $\mathcal{A}$ are both finite. However, real-world applications often have high-dimensional or even infinite-dimensional state and action spaces, where function approximation is required for computational tractability and generalization. While provably efficient online RL with linear function approximation has been widely studied recently [46, 17, 50, 15, 3, 54], little work has been done for analyzing OPE with linear function approximation, with one notable exception by Duan et al. [10]. More specifically, Duan et al. [10] analyzed a regression-based

---

[*]Equal contribution.

35th Conference on Neural Information Processing Systems (NeurIPS 2021).

Fitted Q-Iteration method (FQI-OPE) that achieves an $\widetilde{\mathcal{O}}(H^2\sqrt{(1+d(\pi,\bar{\pi}))/N})$ error for linear MDPs [46, 17], where $H$ is the planning horizon, $N$ is the sample size, and $d(\pi,\bar{\pi})$ represents the distribution shift between the behavior policy and the target policy. They also proved a sample complexity lower bound for a subclass of linear MDPs, for which their algorithm is nearly minimax optimal. However, as we will show later, the $H^2$ dependence is not tight since they discard the useful variance information contained in the offline data. Consequently, their result is only optimal for a small class of MDPs of which the value functions have large variance. The $H^2$ dependence in the sample complexity also makes their algorithm less sample-efficient for long-horizon problems, which is one of the major challenges in RL.

Extracting useful information from the data is particularly important for offline RL since the agent cannot sample additional data by interacting with the environment, as compared to online RL. In this paper, we propose a new algorithm that incorporates the variance information of the value functions to improve the sample efficiency of OPE. This allows us to achieve a deeper understanding and tighter error bounds of OPE with linear function approximation. In detail, we consider time-inhomogeneous linear MDPs [46, 17] where the transition probability and reward function are assumed to be linear functions of a known feature mapping and may vary from stage to stage.

The main contributions of this paper are summarized as follows:

- We develop VA-OPE (Variance-Aware Off-Policy Evaluation), an algorithm for OPE that effectively utilizes the variance information from the offline data. The core idea behind the proposed algorithm is to calibrate the Bellman residual in the regression by an estimator of the conditional variance of the value functions, such that data points of higher quality can receive larger important weights.

- We show that our algorithm achieves $\widetilde{\mathcal{O}}(\sum_h (\mathbf{v}_h^\top \mathbf{\Lambda}_h^{-1} \mathbf{v}_h)^{1/2}/\sqrt{K})$ policy evaluation error, where $\mathbf{v}_h$ is the expectation of the feature vectors under target policy and $\mathbf{\Lambda}_h$ is the uncentered covariance matrix under behavior policy weighted by the conditional variance of the value function. Our algorithm achieves a tighter error bound and milder dependence on $H$ than FQI-OPE [10], and provides a tighter characterization of the distribution shift between the behavior policy and the target policy, which is also verified by extensive numerical experiments.

- Our analysis is based on a novel two-step proof technique. In the first step, we use backward induction to establish worst-case uniform convergence[2] results for the estimators of the value functions. In the second step, the convergence of OPE estimator is proved by tightening the uniform convergence result based on an average-case analysis. Our proof strategy provides a generic way for analyzing (weighted) ridge regression methods that are carried out in a backward and iterated fashion. The analyses in both steps might be of independent interest.

**Notation** We use lower case letters to denote scalars and use lower and upper case boldface letters to denote vectors and matrices respectively. For any vector $\mathbf{x} \in \mathbb{R}^d$ and any positive semi-definite matrix $\mathbf{\Sigma} \in \mathbb{R}^{d \times d}$, we denote by $\|\mathbf{x}\|_2$ the Euclidean norm and $\|\mathbf{\Sigma}\|$ the operator norm, and define $\|\mathbf{x}\|_{\mathbf{\Sigma}} = \sqrt{\mathbf{x}^\top \mathbf{\Sigma} \mathbf{x}}$. For any positive integer $n$, we denote by $[n]$ the set $\{1, \ldots, n\}$. For any finite set $A$, we denote by $|A|$ the cardinality of $A$. For two sequences $\{a_n\}$ and $\{b_n\}$, we write $a_n = \mathcal{O}(b_n)$ if there exists an absolute constant $C$ such that $a_n \leq C b_n$, and we write $a_n = \Omega(b_n)$ if there exists an absolute constant $C$ such that $a_n \geq C b_n$. We use $\widetilde{\mathcal{O}}(\cdot)$ to further hide the logarithmic factors.

## 2 Preliminaries

### 2.1 Markov Decision Processes

We consider the time-inhomogeneous episodic Markov Decision Process (MDP), which is represented by a tuple $M(\mathcal{S}, \mathcal{A}, H, \{r_h\}_{h=1}^H, \{\mathbb{P}_h\}_{h=1}^H)$. In specific, we denote the state space by $\mathcal{S}$ and the action space by $\mathcal{A}$, and $H > 0$ is the horizon length of each episode. At each stage $h \in [H]$, $r_h : \mathcal{S} \times \mathcal{A} \to [0,1]$ is the reward function, and $\mathbb{P}_h(s'|s,a)$ is the transition probability function which represents the probability for state $s$ to transit to state $s'$ given action $a$. A policy $\pi$ consists of $H$ mappings $\{\pi_h\}_{h=1}^H$ from $\mathcal{S}$ to the simplex on $\mathcal{A}$, such that for any $(h,s) \in [H] \times \mathcal{S}$, $\pi_h(\cdot|s)$ is a probability distribution over $\mathcal{A}$. Here a policy can be either deterministic (point mass) or stochastic.

---

[2]By uniform convergence we mean the convergence of the estimated value functions in $\ell_\infty$-norm to their true values, which is different from the uniform convergence over all policies in Yin et al. [48].

For any policy $\pi$, we define the associated action-value function $Q_h^\pi(s, a)$ and value function $V_h^\pi(s)$ at each stage $h \in [H]$ as follows:

$$Q_h^\pi(s, a) = \mathbb{E}_\pi\left[ \sum_{i=h}^H r_i(s_i, a_i) \bigg| s_h = s, a_h = a \right], \qquad V_h^\pi(s) = \int_\mathcal{A} Q_h^\pi(s, a) \mathrm{d}\pi_h(a|s), \qquad (2.1)$$

where $a_i \sim \pi_i(\cdot|s_i)$ and $s_{i+1} \sim \mathbb{P}_i(\cdot|s_i, a_i)$. For any function $V : \mathcal{S} \to \mathbb{R}$, we introduce the following shorthand notation for the conditional expectation and variance of $V$:

$$[\mathbb{P}_h V](s, a) = \mathbb{E}_{s' \sim \mathbb{P}_h(\cdot|s,a)}[V(s')], \qquad [\mathbb{V}_h V](s, a) = [\mathbb{P}_h V^2](s, a) - ([\mathbb{P}_h V](s, a))^2. \qquad (2.2)$$

**Time-inhomogeneous linear MDPs.** We consider a special class of MDPs called *linear MDPs* [46, 17]. Note that most of the existing works on RL with linear function approximation rely on this assumption.

**Assumption 2.1.** $M(\mathcal{S}, \mathcal{A}, H, \{r_h\}_{h=1}^H, \{\mathbb{P}_h\}_{h=1}^H)$ is called a linear MDP with a *known* feature mapping $\phi : \mathcal{S} \times \mathcal{A} \to \mathbb{R}^d$, if for any $h \in [H]$, there exist $\gamma_h$ and $\mu_h \in \mathbb{R}^d$, such that for any state-action pair $(s, a) \in \mathcal{S} \times \mathcal{A}$, it holds that

$$\mathbb{P}_h(\cdot \mid s, a) = \langle \phi(s, a), \mu_h(\cdot) \rangle, \qquad r_h(s, a) = \langle \phi(s, a), \gamma_h \rangle. \qquad (2.3)$$

We assume that at any stage $h$, for any state-action pair $(s, a) \in \mathcal{S} \times \mathcal{A}$, the reward received by the agent is given by $r = r_h(s, a) + \epsilon_h(s, a)$, where $r_h(s, a) \in [0, 1]$ is the expected reward and $\epsilon_h(s, a)$ is the random noise. We assume that the noise is zero-mean and independent of anything else.

Without loss of generality, we assume that $\|\gamma_h\|_2 \leq 1$ and $\|\phi(s, a)\|_2 \leq 1$ for all $(s, a) \in \mathcal{S} \times \mathcal{A}$. We also assume that $r_h(s, a) + \epsilon_h(s, a) \leq 1$, $|\epsilon_h(s, a)| \leq 1$ almost surely and thus $\mathrm{Var}(\epsilon_h(s, a)) \leq 1$ for all $h \in [H]$ and $(s, a) \in \mathcal{S} \times \mathcal{A}$. Moreover, we assume that $\max_{h \in [H]} \left\| \int_\mathcal{S} f(s) \mathrm{d}\mu_h(s) \right\|_2 \leq \sqrt{d}$ for all bounded function $f : \mathcal{S} \to \mathbb{R}$ such that $\sup_{s \in \mathcal{S}} |f(s)| \leq 1$.

The above assumption on linear MDPs implies the following proposition for the action-value functions.

**Proposition 2.2** (Proposition 2.3, [17]). For a linear MDP, for any policy $\pi$, there exist weights $\{\mathbf{w}_h^\pi, h \in [H]\}$ such that for any $(s, a, h) \in \mathcal{S} \times \mathcal{A} \times [H]$, we have $Q_h^\pi(s, a) = \langle \phi(s, a), \mathbf{w}_h^\pi \rangle$. Moreover, we have $\|\mathbf{w}_h^\pi\|_2 \leq 2H\sqrt{d}$ for all $h \in [H]$.

Following this proposition, we may further show that the value functions are also linear functions, but of different features. We define $\phi_h^\pi(s) = \int_\mathcal{A} \phi(s, a) \mathrm{d}\pi_h(a|s)$ for all $s \in [S]$ and $h \in [H]$. Then by (2.1) we have

$$V_h^\pi(s) = \int_\mathcal{A} \phi(s, a)^\top \mathbf{w}_h^\pi \mathrm{d}\pi(a|s) = \langle \phi_h^\pi(s), \mathbf{w}_h^\pi \rangle.$$

## 2.2 Off-policy Evaluation

The purpose of OPE is to evaluate a (known) target policy $\pi$ given an offline dataset generated by a different (unknown) behavior policy $\bar\pi$. In this paper, our goal is to estimate the expectation of the value function induced by $\pi$ over a fixed initial distribution $\xi_1$, i.e.,

$$v_1^\pi = \mathbb{E}_{s \sim \xi_1}[V_1^\pi(s)].$$

To faciliate the presentation, we further introduce some important notations. For all $h \in [H]$, let $\nu_h$ be the occupancy measure over $\mathcal{S} \times \mathcal{A}$ at stage $h$ induced by the transition $\mathbb{P}$ and the behavior policy $\bar\pi$, that is, for any $E \subseteq \mathcal{S} \times \mathcal{A}$,

$$\nu_h(E) = \mathbb{E}\left[ (s_h, a_h) \in E \mid s_1 \sim \xi_1, \ a_i \sim \bar\pi(\cdot|s_i), \ s_{i+1} \sim \mathbb{P}_i(\cdot|s_i, a_i), \ 1 \leq i \leq h \right]. \qquad (2.4)$$

For simplicity, we write $\mathbb{E}_h[f(s, a)] = \mathbb{E}_{\bar\pi,h}[f(s, a)] = \int_{\mathcal{S} \times \mathcal{A}} f(s, a) \mathrm{d}\nu_h(s, a)$ for any function $f$ on $\mathcal{S} \times \mathcal{A}$. Similarly, we use $\mathbb{E}_{\pi,h}[f(s, a)]$ to denote the expectation of $f$ with respect to the occupancy measure at stage $h$ induced by the transition $\mathbb{P}$ and the target policy $\pi$.

We define the following uncentered covariance matrix under behavior policy for all $h \in [H]$:

$$\Sigma_h = \mathbb{E}_{\bar\pi,h}\left[ \phi(s, a)\phi(s, a)^\top \right]. \qquad (2.5)$$

Intuitively, these matrices measure the coverage of the offline data in the state-action space. It is known that the success of OPE necessitates a good coverage [10, 43]. Therefore here we make the same coverage assumption on the offline data.

**Assumption 2.3** (Coverage). For all $h \in [H]$, $\kappa_h := \lambda_{\min}(\mathbf{\Sigma}_h) > 0$. Denote $\kappa = \min_{h \in [H]} \kappa_h$.

A key difference in our result is that, instead of depending on $\mathbf{\Sigma}_h$ directly, the error bound depends on the following weighted version of the covariance matrices defined as

$$\mathbf{\Lambda}_h := \mathbb{E}_{\bar{\pi}, h} \left[ \sigma_h(s, a)^{-2} \phi(s, a) \phi(s, a)^\top \right], \tag{2.6}$$

for all $h \in [H]$, where each $\sigma_h : \mathcal{S} \times \mathcal{A} \to \mathbb{R}$ is defined as

$$\sigma_h(s, a) := \sqrt{\max\{1, \mathbb{V}_h V_{h+1}^\pi(s, a)\} + 1}. \tag{2.7}$$

Note that in the definition of $\sigma_h(\cdot, \cdot)$, taking the maximum and adding an extra 1 is purely for technical reason and is related to its estimator $\widehat{\sigma}_h(\cdot, \cdot)$, which we will introduce and explain later in Section 3.2. In general, one can think of $\sigma_h^2(s, a) \approx \mathbb{V}_h V_{h+1}^\pi(s, a)$. Therefore, compared with the raw covariance matrix $\mathbf{\Sigma}_h$, $\mathbf{\Lambda}_h$ further incorporates the variance of the value functions under the target policy. This is the key to obtaining a tighter instance-dependent error bound.

**Definition 2.4** (Variance-aware coverage). We define $\iota_h := \lambda_{\min}(\mathbf{\Lambda}_h)$ and $\iota = \min_{h \in [H]} \iota_h$.

Since $\sup_{(s,a) \in \mathcal{S} \times \mathcal{A}} \sigma_h(s, a)^2$ is bounded from above, by (2.6) and Assumption 2.3, we immediately have $\iota_h \geq \kappa_h / [\sup_{(s,a) \in \mathcal{S} \times \mathcal{A}} \sigma_h(s, a)^2] > 0$ for all $h \in [H]$, and thus $\iota > 0$. Even if Assumption 2.3 does not hold, we can always restrict to the subspace $\text{span}\{\phi(s_h, a_h)\}$. For convenience of presentation, we make Assumption 2.3 in this paper.

Next, we introduce the assumption on the sampling process of the offline data.

**Assumption 2.5** (Stage-sampling Data). We have two offline datasets $\mathcal{D}$ and $\check{\mathcal{D}}$ where each dataset consists of data from $H$ stages: $\mathcal{D} = \{\mathcal{D}_h\}_{h \in [H]}$ and $\check{\mathcal{D}} = \{\check{\mathcal{D}}_h\}_{h \in [H]}$. For the dataset $\mathcal{D}$, we assume $\mathcal{D}_{h_1}$ is independent of $\mathcal{D}_{h_2}$ for $h_1 \neq h_2$. For each stage $h$, we have $\mathcal{D}_h = \{(s_{k,h}, a_{k,h}, r_{k,h}, s'_{k,h})\}_{k \in [K]}$, where we assume for each $k \in [K]$, the data point $(s_{k,h}, a_{k,h}, r_{k,h}, s'_{k,h})$ is sampled identically and independently in the following way: $(s_{k,h}, a_{k,h}) \sim \nu_h(\cdot, \cdot)$ where $\nu_h(\cdot, \cdot)$ is the occupancy measure defined in (2.4), and $s'_{k,h} \sim \mathbb{P}_h(\cdot | s_{k,h}, a_{k,h})$. The same holds for $\check{\mathcal{D}}$, and we write $\check{\mathcal{D}}_h = \{(\check{s}_{k,h}, \check{a}_{k,h}, \check{r}_{k,h}, \check{s}'_{k,h})\}_{k \in [K]}$. Note that here $s'_{k,h} \neq s_{k,h+1}$.

Assumptions 2.5 is standard in the offline RL literature [48, 10]. Note that in the assumption, there is a data splitting, i.e., one can view it as the whole dataset $\mathcal{D} \cup \check{\mathcal{D}}$ being split into two halves. The datasets $\mathcal{D}$ and $\check{\mathcal{D}}$ will then be used for two different purposes in Algorithm 1 as will be made clear in the next section. We would like to remark that the only purpose of the splitting is to avoid a lengthy analysis. There is no need to perform the data splitting in practice. Also, in our implementation and experiments, we do not split the data.

## 3 Algorithm

To ease the notation, we denote $\phi_{k,h} = \phi(s_{k,h}, a_{k,h})$, $\check{\phi}_{k,h} = \phi(\check{s}_{k,h}, \check{a}_{k,h})$, $\widehat{\sigma}_{k,h} = \widehat{\sigma}_h(s_{k,h}, a_{k,h})$ and $r_{k,h} = r_h(s_{k,h}, a_{k,h}) + \epsilon_{k,h}$ for all $(h, k) \in [H] \times [K]$. Recall that we use the check mark to denote the other half of the splitted dataset. How the splitted data is utilized will be clear in Section 3.2 when we introduce the proposed algorithm.

### 3.1 Regression-Based Value Function Estimation

By Proposition 2.2, it suffices to estimate the vectors $\{\mathbf{w}_h^\pi, h \in [H]\}$. A popular approach is to apply the Least-Square Value Iteration (LSVI) [17] which relies on the Bellman equation, $Q_h^\pi(s, a) = r_h(s, a) + [\mathbb{P}_h V_{h+1}^\pi](s, a)$, that holds for all $h \in [H]$ and $(s, a) \in \mathcal{S} \times \mathcal{A}$. By viewing $V_{h+1}^\pi(s'_{k,h})$ as an unbiased estimate of $[\mathbb{P}_h V_{h+1}^\pi](s_{k,h}, a_{k,h})$, the idea of the LSVI-type method is to solve the following ridge regression problem:

$$\widehat{\mathbf{w}}_h^\pi := \underset{\mathbf{w} \in \mathbb{R}^d}{\text{argmin}} \, \lambda \|\mathbf{w}\|_2^2 + \sum_{k=1}^K \left[ \langle \phi_{k,h}, \mathbf{w} \rangle - r_{k,h} - V_{h+1}^\pi(s'_{k,h}) \right]^2, \tag{3.1}$$

for some regularization parameter $\lambda > 0$. Since we do not know the exact values of $V_{h+1}^\pi$ in (3.1), we replace it by an estimator $\widehat{V}_{h+1}^\pi$, and then recursively solve the lease-square problem in a backward manner, which enjoys a closed-form solution as follows

$$\widehat{\mathbf{w}}_h^\pi = \left[ \sum_{k=1}^K \boldsymbol{\phi}_{k,h} \boldsymbol{\phi}_{k,h}^\top + \lambda \mathbf{I}_d \right]^{-1} \sum_{k=1}^K \boldsymbol{\phi}_{k,h} \left[ r_{k,h} + \widehat{V}_{h+1}^\pi(s_{k,h}') \right].$$

This has been used in the LSVI-UCB algorithm proposed by Jin et al. [17] and the FQI-OPE algorithm studied by Duan et al. [10], for online learning and OPE of linear MDPs respectively. For this kind of algorithms, the key difficulty in the analysis lies in bounding the Bellman error:

$$\left[ \sum_{k=1}^K \boldsymbol{\phi}_{k,h} \boldsymbol{\phi}_{k,h}^\top + \lambda \mathbf{I}_d \right]^{-1} \sum_{k=1}^K \boldsymbol{\phi}_{k,h} \left( [\mathbb{P}_h \widehat{V}_{h+1}^\pi](s_{k,h}, a_{k,h}) - \widehat{V}_{h+1}^\pi(s_{k,h}') \right).$$

Jin et al. [17] applied a Hoeffding-type inequality to bound the Bellman error. Although Duan et al. [10] applied Freedman's inequality in their analysis, their algorithm design overlooks the variance information in the data and consequently they can only adopt a crude upper bound on the conditional variance of the value function, i.e., $\mathbb{V}_h V_{h+1}^\pi \leq (H-h)^2$, which simply comes from $\sup_{s \in \mathcal{S}} V_{h+1}^\pi(s) \leq H-h$. Therefore, it prevents [10] from getting a tight instance-dependent error bound for OPE. This is further verified by our numerical experiments in Appendix A which show that the performance of FQI-OPE degrades for large $H$. This motivates us to utilize the variance information in the data for OPE.

## 3.2 The Proposed Algorithm

In particular, we present our main algorithm as displayed in Algorithm 1. Due to the greedy nature of the value functions, we adopt a backward estimation scheme.

**Weighted ridge regression.** For any $h \in [H]$, let $\widehat{\mathbf{w}}_{h+1}^\pi$ be the estimate of $\mathbf{w}_{h+1}^\pi$ computed at the previous step, and correspondingly $\widehat{V}_{h+1}^\pi(\cdot) = \langle \boldsymbol{\phi}_{h+1}^\pi(\cdot), \widehat{\mathbf{w}}_{h+1}^\pi \rangle$. Instead of the ordinary ridge regression (3.1), we consider the following weighted ridge regression:

$$\widehat{\mathbf{w}}_h^\pi := \operatorname*{argmin}_{\mathbf{w} \in \mathbb{R}^d} \lambda \|\mathbf{w}\|_2^2 + \sum_{k=1}^K \left[ \langle \boldsymbol{\phi}_{k,h}, \mathbf{w} \rangle - r_{k,h} - \widehat{V}_{h+1}^\pi(s_{k,h}') \right]^2 / \widehat{\sigma}_{k,h}^2, \tag{3.2}$$

where $\widehat{\sigma}_{k,h} = \widehat{\sigma}_h(s_{k,h}, a_{k,h})$ for all $(h,k) \in [H] \times [K]$ with $\widehat{\sigma}_h(\cdot, \cdot)$ being a proper estimate of $\sigma_h(\cdot, \cdot)$ defined in (2.7). We then have the following closed-form solution (Line 9 and 7 of Alg. 1):

$$\widehat{\mathbf{w}}_h^\pi = \widehat{\boldsymbol{\Lambda}}_h^{-1} \sum_{k=1}^K \boldsymbol{\phi}_{k,h} \left( r_{k,h} + \widehat{V}_{h+1}^\pi(s_{k,h}') \right) / \widehat{\sigma}_{k,h}^2, \text{ with } \widehat{\boldsymbol{\Lambda}}_h = \sum_{k=1}^K \widehat{\sigma}_{k,h}^{-2} \boldsymbol{\phi}_{k,h} \boldsymbol{\phi}_{k,h}^\top + \lambda \mathbf{I}_d. \tag{3.3}$$

In the above estimator, we use the dataset $\mathcal{D}$ to estimate the value functions. Next, we apply an LSVI-type method to estimate $\sigma_h$ using the dataset $\check{\mathcal{D}}$.

**Variance estimator.** By (2.2), we can write

$$[\mathbb{V}_h V_{h+1}^\pi](s,a) = [\mathbb{P}_h (V_{h+1}^\pi)^2](s,a) - \left( [\mathbb{P}_h V_{h+1}^\pi](s,a) \right)^2. \tag{3.4}$$

For the first term in (3.4), by Assumption 2.1 we have

$$[\mathbb{P}_h (V_{h+1}^\pi)^2](s,a) = \int_{\mathcal{S}} V_{h+1}^\pi(s')^2 d\mathbb{P}_h(s'|s,a) = \boldsymbol{\phi}(s,a)^\top \int_{\mathcal{S}} V_{h+1}^\pi(s')^2 \, d\boldsymbol{\mu}_h(s'),$$

which suggests that $\mathbb{P}_h (V_{h+1}^\pi)^2$ also has a linear representation. Thus we adopt a linear estimator $\langle \boldsymbol{\phi}(s,a), \widehat{\boldsymbol{\beta}}_h^\pi \rangle$ where $\widehat{\boldsymbol{\beta}}_h^\pi$ (Line 4) is the solution to the following ridge regression problem:

$$\widehat{\boldsymbol{\beta}}_h^\pi = \operatorname*{argmin}_{\boldsymbol{\beta} \in \mathbb{R}^d} \sum_{k=1}^K \left[ \langle \check{\boldsymbol{\phi}}_{k,h}, \boldsymbol{\beta} \rangle - [\widehat{V}_{h+1}^\pi]^2(\check{s}_{k,h}') \right]^2 + \lambda \|\boldsymbol{\beta}\|_2^2 = \widehat{\boldsymbol{\Sigma}}_h^{-1} \sum_{k=1}^K \check{\boldsymbol{\phi}}_{k,h} \widehat{V}_{h+1}^\pi(\check{s}_{k,h}')^2. \tag{3.5}$$

---

**Algorithm 1** Variance-Aware Off-Policy Evaluation (VA-OPE)

---

1: **Input:** target policy $\pi = \{\pi_h\}_{h \in [H]}$, datasets $\mathcal{D} = \{\{(s_{k,h}, a_{k,h}, r_{k,h}, s'_{k,h})\}_{h \in [H]}\}_{k \in [K]}$ and $\check{\mathcal{D}} = \{\{(\check{s}_{k,h}, \check{a}_{k,h}, \check{r}_{k,h}, \check{s}'_{k,h})\}_{h \in [H]}\}_{k \in [K]}$, initial distribution $\xi_1$, $\widehat{\mathbf{w}}^\pi_{H+1} = \mathbf{0}$
2: **for** $h = H, H-1, \ldots, 1$ **do**
3:     $\widehat{\boldsymbol{\Sigma}}_h \leftarrow \sum_{k=1}^K \check{\boldsymbol{\phi}}_{k,h} \check{\boldsymbol{\phi}}^\top_{k,h} + \lambda \mathbf{I}_d$
4:     $\widehat{\boldsymbol{\beta}}_h \leftarrow \widehat{\boldsymbol{\Sigma}}^{-1}_h \sum_{k=1}^K \check{\boldsymbol{\phi}}_{k,h} \widehat{V}^\pi_{h+1}(\check{s}'_{k,h})^2$
5:     $\widehat{\boldsymbol{\theta}}_h \leftarrow \widehat{\boldsymbol{\Sigma}}^{-1}_h \sum_{k=1}^K \check{\boldsymbol{\phi}}_{k,h} \widehat{V}^\pi_{h+1}(\check{s}'_{k,h})$
6:     $\widehat{\sigma}_h(\cdot, \cdot) \leftarrow \sqrt{\max\{1, \widehat{\mathbb{V}}_h \widehat{V}^\pi_{h+1}(\cdot, \cdot)\} + 1}$
7:     $\widehat{\boldsymbol{\Lambda}}_h \leftarrow \sum_{k=1}^K \boldsymbol{\phi}_{k,h} \boldsymbol{\phi}^\top_{k,h} / \widehat{\sigma}^2_{k,h} + \lambda \mathbf{I}_d$
8:     $Y_{k,h} \leftarrow r_{k,h} + \langle \boldsymbol{\phi}^\pi_h(s'_{k,h}), \widehat{\mathbf{w}}^\pi_{h+1} \rangle$
9:     $\widehat{\mathbf{w}}^\pi_h \leftarrow \widehat{\boldsymbol{\Lambda}}^{-1}_h \sum_{k=1}^K \boldsymbol{\phi}_{k,h} Y_{k,h} / \widehat{\sigma}^2_{k,h}$
10:    $\widehat{Q}^\pi_h(\cdot, \cdot) \leftarrow \langle \boldsymbol{\phi}(\cdot, \cdot), \widehat{\mathbf{w}}^\pi_h \rangle$,    $\widehat{V}^\pi_h(\cdot) \leftarrow \langle \boldsymbol{\phi}^\pi_h(\cdot), \widehat{\mathbf{w}}^\pi_h \rangle$
11: **end for**
12: **Output:** $\widehat{v}^\pi_1 \leftarrow \int_{\mathcal{S}} \widehat{V}^\pi_1(s) \, \mathrm{d}\xi_1(s)$

---

Similarly, we estimate the second term in (3.4) by $\langle \boldsymbol{\phi}(s, a), \widehat{\boldsymbol{\theta}}^\pi_h \rangle$, where $\widehat{\boldsymbol{\theta}}^\pi_h$ (Line 5) is given by

$$\widehat{\boldsymbol{\theta}}_h = \operatorname*{argmin}_{\boldsymbol{\theta} \in \mathbb{R}^d} \sum_{k=1}^K \left[ \langle \check{\boldsymbol{\phi}}_{k,h}, \boldsymbol{\theta} \rangle - \widehat{V}^\pi_{h+1}(\check{s}'_{k,h}) \right]^2 + \lambda \|\boldsymbol{\theta}\|^2_2 = \widehat{\boldsymbol{\Sigma}}^{-1}_h \sum_{k=1}^K \check{\boldsymbol{\phi}}_{k,h} \widehat{V}^\pi_{h+1}(\check{s}'_{k,h}), \qquad (3.6)$$

with $\widehat{\boldsymbol{\Sigma}}_h = \sum_{k=1}^K \check{\boldsymbol{\phi}}_{k,h} \check{\boldsymbol{\phi}}^\top_{k,h} + \lambda \mathbf{I}_d$. Combining (3.5) and (3.6), we estimate $\mathbb{V}_h V^\pi_{h+1}$ by

$$[\widehat{\mathbb{V}}_h \widehat{V}^\pi_{h+1}](\cdot, \cdot) = \langle \boldsymbol{\phi}(\cdot, \cdot), \widehat{\boldsymbol{\beta}}^\pi_h \rangle_{[0, (H-h+1)^2]} - \left[ \langle \boldsymbol{\phi}(\cdot, \cdot), \widehat{\boldsymbol{\theta}}^\pi_h \rangle_{[0, H-h+1]} \right]^2, \qquad (3.7)$$

where the subscript $[0, (H-h+1)^2]$ denotes the clipping into the given range, and similar for the subscript $[0, H-h+1]$. We do such clipping due to the fact that $V^\pi_{h+1} \in [0, H-h]$. We add 1 to deal with the approximation error in $\widehat{V}^\pi_{h+1}$.

Based on $\widehat{\mathbb{V}}_h \widehat{V}^\pi_{h+1}$, the final variance estimator $\widehat{\sigma}_h(\cdot, \cdot)$ (Line 6) is defined as

$$\widehat{\sigma}_h(\cdot, \cdot) = \sqrt{\max\{1, \widehat{\mathbb{V}}_h \widehat{V}^\pi_{h+1}(\cdot, \cdot)\} + 1}.$$

In order to deal with the situation where $\widehat{\mathbb{V}}_h \widehat{V}^\pi_{h+1} < 0$ or is very close to 0, we take maximum between $\widehat{\mathbb{V}}_h \widehat{V}^\pi_{h+1}$ and 1. Also, to account for the noise in the observed rewards, we add an extra 1 which is an upper bound of the noise variance by Assumption 2.1.

**Final estimator.** Recursively repeat the above procedure for $h = H, H-1, \ldots, 1$, and we obtain $\widehat{V}_1$. Then the final estimator for $v^\pi_1$ (Line 12) is defined as $\widehat{v}^\pi_1 = \int_{\mathcal{S}} \widehat{V}^\pi_1(s) \, \mathrm{d}\xi_1(s)$.

**Intuition behind $\Lambda_h$.** To illustrate the intuition behind the weighted covariance matrix $\Lambda_h$, here we provide some brief heuristics. Let $\{(s_{k,h}, a_{k,h}, s'_{k,h})\}_{k \in [K]}$ be i.i.d. samples such that $(s_{k,h}, a_{k,h}) \sim \nu$ for some distribution $\nu$ over $\mathcal{S} \times \mathcal{A}$ and $s'_{k,h} \sim \mathbb{P}_h(\cdot | s_{k,h}, a_{k,h})$. Define

$$\mathbf{e}_k = \boldsymbol{\phi}(s_{k,h}, a_{k,h}) \left( [\mathbb{P}_h V^\pi_{h+1}](s_{k,h}, a_{k,h}) - V^\pi_{h+1}(s'_{k,h}) \right) / [\mathbb{V}_h V^\pi_{h+1}](s_{k,h}, a_{k,h})^2$$

for all $k \in [K]$. Note that $\mathbf{e}_k$'s are i.i.d zero-mean random vectors and a simple calculation yields

$$\operatorname{Cov}(\mathbf{e}_k) = \mathbb{E} \left[ [\mathbb{V}_h V^\pi_{h+1}](s_{k,h}, a_{k,h})^{-2} \boldsymbol{\phi}(s_{k,h}, a_{k,h}) \boldsymbol{\phi}(s_{k,h}, a_{k,h})^\top \right].$$

This coincides with (2.6). Suppose $\operatorname{Cov}(\mathbf{e}_k) \succ 0$, then by the central limit theorem, it holds that

$$\frac{1}{\sqrt{K}} \sum_{k=1}^K \mathbf{e}_k \xrightarrow{d} \mathcal{N}(0, \operatorname{Cov}(\mathbf{e}_k)).$$

Therefore, $\operatorname{Cov}(\mathbf{e}_k)^{-1}$, or equivalently $\boldsymbol{\Lambda}^{-1}_h$, can be seen as the Fisher information matrix associated with the weighted product of the Bellman error and the feature vectors. This is a tighter characterization of the convergence rate than bounding $\mathbb{V}_h V^\pi_{h+1}$ by its naive upper bound $(H-h)^2$.

# 4 Theoretical Results

In this section, we introduce our main theoretical results and give an overview of the proof technique.

## 4.1 OPE Error Bound

Our main result is a refined average-case OPE analysis that yields a tighter error bound in Theorem 4.1. The proof is in Appendix D. To simplify the notation, we define:

$$C_{h,2} = \sum_{i=h}^{H} \frac{H-h+1}{\sqrt{2\iota_h}}, \quad C_{h,3} = \frac{(H-h+1)^2}{2}, \quad C_{h,4} = \left(\|\boldsymbol{\Lambda}_h\| \cdot \|\boldsymbol{\Lambda}_h^{-1}\|\right)^{1/2}.$$

**Theorem 4.1.** Set $\lambda = 1$. Under Assumptions 2.1, 2.3 and 2.5, if $K$ satisfies

$$K \geq C \cdot C_3 \cdot d^2 \left[\log\left(\frac{dH^2K}{\kappa\delta}\right)\right]^2, \tag{4.1}$$

where $C$ is some problem-independent universal constant and

$$C_3 := \max\left\{ \max_{h\in[H]} \frac{C_{h,3} \cdot C_{h,2}^2}{8\iota_h^2}, \frac{H^4}{\kappa^2}, \frac{H^2}{\kappa^2} \cdot \max_{h\in[H]} \frac{C_{h,3}}{2} \cdot \max_{h\in[H]} \frac{C_{h,3}}{\iota_h} \right\},$$

then with probability at least $1 - \delta$, the output of Algorithm 1 satisfies

$$|v_1^\pi - \widehat{v}_1^\pi| \leq C \cdot \left[\sum_{h=1}^{H} \|\mathbf{v}_h^\pi\|_{\boldsymbol{\Lambda}_h^{-1}}\right] \cdot \sqrt{\frac{\log(16H/\delta)}{K}} + C \cdot C_4 \cdot \log\left(\frac{16H}{\delta}\right) \cdot \left(\frac{1}{K^{3/4}} + \frac{1}{K}\right),$$

where $\mathbf{v}_h^\pi := \mathbb{E}_{\pi,h}[\phi(s_h, a_h)]$ and $C_4 := \sum_{h=1}^{H} \sqrt{C_{h,4} \cdot C_{h,2} \cdot \frac{(H-h+1)d}{4\iota_h} \cdot \log\left(\frac{dH^2K}{\kappa\delta}\right)} \cdot \|\mathbf{v}_h^\pi\|_{\boldsymbol{\Lambda}_h^{-1}}$.

Theorem 4.1 suggests that Algorithm 1 provably achieves a tighter instance-dependent error bound for OPE than that in [10]. In detail, the dominant term in our bound is $\widetilde{\mathcal{O}}(\sum_{h=1}^{H} \|\mathbf{v}_h^\pi\|_{\boldsymbol{\Lambda}_h^{-1}}/\sqrt{K})$, as compared to the $\widetilde{\mathcal{O}}(\sum_{h=1}^{H}(H-h+1)\|\mathbf{v}_h^\pi\|_{\boldsymbol{\Sigma}_h^{-1}}/\sqrt{K})$ term in [10]. By (2.5) and (2.6), our bound is at least as good as the latter since $\boldsymbol{\Sigma}_h \preceq [(H-h+1)^2 + 1]\boldsymbol{\Lambda}_h$. More importantly, it is instance-dependent and tight for the general class of linear MDPs: for those where $\mathbb{V}_h V_{h+1}^\pi$ is close to its crude upper bound $(H-h+1)^2$, our bound recovers the prior result. When $\mathbb{V}_h V_{h+1}^\pi$ is small, VA-OPE benefits from incorporating the variance information and our bound gets tightened accordingly.

**Remark 4.2.** Note that we do not require $\mathbb{V}_h V_{h+1}^\pi(s, a)$ to be uniformly small for all $s, a$, and $h$. From the bound and (2.6), as long as the variances are smaller than $(H-h+1)^2$ on average of $(s, a) \in \mathcal{S} \times \mathcal{A}$ and in sum of $h$, the bound is improved. It is also worth noting that the lower bound proved in [10] only holds for a subclass of linear MDPs with $\mathbb{V}_h V_{h+1}^\pi = \Omega((H-h+1)^2)$, and thus their minimax-optimality does not hold for general linear MDPs. For more detailed comparison we refer the reader to Appendix B.

**Remark 4.3.** Conceptually, the term $\|\mathbf{v}_h^\pi\|_{\boldsymbol{\Lambda}_h^{-1}}$ serves as a more precise characterization of the distribution shift between the behavior policy $\bar{\pi}$ and the target policy $\pi$ in a variance-aware manner. This enables our algorithm to utilize the data more effectively. Compared with online RL where one can sample new data, OPE is more 'data-hungry': one cannot decide the overall quality of the data. Thus it is especially beneficial to put more focus on targeted values with less uncertainty. This is also the intuitive reason why our algorithm can achieve a tighter error bound.

## 4.2 Overview of the Proof Technique

Here we provide an overview of the proof for Theorem 4.1. Due to the parallel estimation of the the value functions and their variances, the analysis of VA-OPE is much more challenging compared with that of FQI-OPE. As a result, we need to develop a novel proof technique. First, we have the following error decomposition.

**Lemma 4.4.** For any $h \in [H]$, let $\widehat{V}_h^\pi$ be the output of Algorithm 1. Then it holds that

$$V_h^\pi(s) - \widehat{V}_h^\pi(s) = \int_{\mathcal{A}} [\mathbb{P}_h(V_{h+1}^\pi - \widehat{V}_{h+1}^\pi)](s,a)\mathrm{d}\pi_h(a|s) + \lambda \boldsymbol{\phi}_h^\pi(s)^\top \widehat{\boldsymbol{\Lambda}}_h^{-1} \mathbf{w}_h^\pi \tag{4.2}$$

$$+ \boldsymbol{\phi}_h^\pi(s)^\top \widehat{\boldsymbol{\Lambda}}_h^{-1} \left[ -\lambda \int_{\mathcal{S}} \left( V_{h+1}^\pi(s') - \widehat{V}_{h+1}^\pi(s') \right) \mathrm{d}\boldsymbol{\mu}_h(s') + \sum_{k=1}^{K} \boldsymbol{\phi}_{k,h} \widehat{\sigma}_{k,h}^{-2} \Delta_{k,h} \right],$$

where $\Delta_{k,h} = [\mathbb{P}_h \widehat{V}_{h+1}^\pi](s_{k,h}, a_{k,h}) - \widehat{V}_{h+1}^\pi(s'_{k,h}) - \epsilon_{k,h}$. In particular, recall that $\widehat{v}_1^\pi = \mathbb{E}[\widehat{V}_1^\pi(s_1) \mid s_1 \sim \xi_1]$ and the OPE error can be decomposed as

$$v_1^\pi - \widehat{v}_1^\pi = -\lambda \sum_{h=1}^{H} (\mathbf{v}_h^\pi)^\top \widehat{\boldsymbol{\Lambda}}_h^{-1} \int_{\mathcal{S}} \left( V_{h+1}^\pi(s) - \widehat{V}_{h+1}^\pi(s) \right) \boldsymbol{\mu}_h(s)\mathrm{d}s$$

$$+ \sum_{h=1}^{H} (\mathbf{v}_h^\pi)^\top \widehat{\boldsymbol{\Lambda}}_h^{-1} \sum_{k=1}^{K} \boldsymbol{\phi}_{k,h} \widehat{\sigma}_{k,h}^{-2} \Delta_{k,h} + \lambda \sum_{h=1}^{H} (\mathbf{v}_h^\pi)^\top \widehat{\boldsymbol{\Lambda}}_h^{-1} \mathbf{w}_h^\pi. \tag{4.3}$$

The OPE error bound (Theorem 4.1) is proved by bounding the three terms separately in (4.3). This decomposition is different from [10] in that $\widehat{\boldsymbol{\Sigma}}_h$ is replaced by $\widehat{\boldsymbol{\Lambda}}_h$. This prevents us from adopting a matrix embedding-type proof as used in the prior work.

The key is to show the convergence of $\widehat{\boldsymbol{\Lambda}}_h$ to its population counterpart. However, by definition of $\widehat{\boldsymbol{\Lambda}}_h$, to establish such a result, it first requires the convergence of $\widehat{V}_{h+1}^\pi$ to $V_{h+1}^\pi$ in a uniform manner, i.e., a high probability bound for $\sup_{s \in \mathcal{S}} |\widehat{V}_h^\pi(s) - V_h^\pi(s)|$. To show this, we leverage the decomposition in (4.2) and a backward induction technique, and prove a uniform convergence result which states that with high probability, for all $h \in [H]$, Algorithm 1 can guarantee

$$\sup_{s \in \mathcal{S}} \left| \widehat{V}_h^\pi(s) - V_h^\pi(s) \right| \leq \widetilde{\mathcal{O}} \left( \frac{1}{\sqrt{K}} \right).$$

This result is formalized as Theorem C.2 and proved in Appendix C. To the best of our knowledge, Theorem C.2 is the first to establish the uniform convergence of the estimation error for the value functions in offline RL with linear function approximation. We believe this result is of independent interest and may be broadly useful in OPE.

## 5 Numerical Experiments

In this section, we provide numerical experiments to evaluate our algorithm `VA-OPE`, and compare it with `FQI-OPE`.

We construct a linear MDP instance as follows. The MDP has $|\mathcal{S}| = 2$ states and $|\mathcal{A}| = 100$ actions, with the feature dimension $d = 10$. The behavior policy then chooses action $a = 0$ with probability $p$ and $a \in \{1, \cdots, 99\}$ with probability $1 - p$ and uniformly over $\{1, \cdots, 99\}$. The target policy $\pi$ always chooses $a = 0$ no matter which state it is, making state 0 and 1 absorbing. The parameter $p$ can be used to control the distribution shift between the behavior and target policies. Here $p \to 0$ leads to small distribution shift, and $p \to 1$ leads to large distribution shift. The initial distribution $\xi_1$ is uniform over $|\mathcal{S}|$. For more details about the construction of the linear MDP and parameter configuration, please refer to Appendix A.

We compare the performance of the two algorithms on the synthetic MDP described above under different choices of horizon length $H$. We plot the log-scaled OPE error versus $\sqrt{K}$ in Figure 1. It is clear that `VA-OPE` is at least as good as `FQI-OPE` in all the cases. Specifically, for small $H$ (Figure 1a), their performance is very comparable, which is as expected. As $H$ increases, we can see from Figure 1a, 1b and 1c that `VA-OPE` starts to dominate `FQI-OPE`, and the advantage is more significant for larger $H$, as suggested by Theorem 4.1. Due to space limit, a comprehensive comparison under different parameter settings is deferred to Appendix A.

## 6 Related Work

**Off-policy evaluation.** There is a large body of literature on OPE for tabular MDPs. Since the seminal work by Precup [34], various importance sampling-based estimators have been studied in

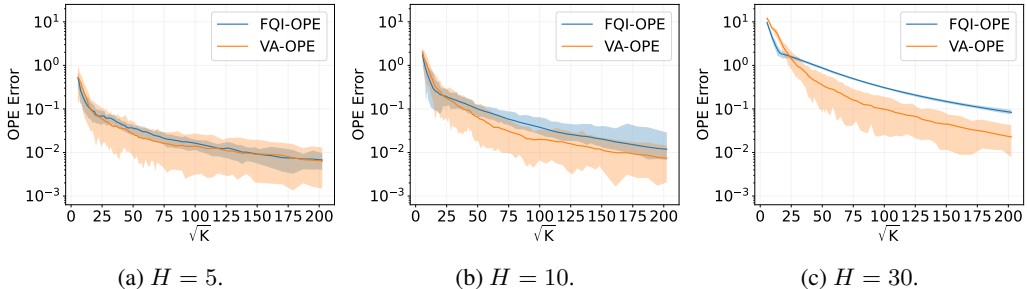

(a) $H = 5$.         (b) $H = 10$.         (c) $H = 30$.

Figure 1: Comparison of VA-OPE and FQI-OPE under different settings of horizon length $H$. VA-OPE's advantage becomes more significant as $H$ increases, matching the theoretical prediction. The results are averaged over 50 trials and the error bars denote an empirical [10%,90%] confidence interval. The y-axis is log-scaled OPE error and x-axis is $\sqrt{K}$. For more details please see Appendix A.

the literature [26, 27, 39]. By using marginalized importance sampling methods [28, 45, 21, 47], one is able to further break the "curse-of-horizon". Moreover, various doubly robust estimators [11, 16, 12, 38, 49] have been developed to achieve variance reduction. Most recently, it is shown by Yin et al. [48] that uniform convergence over all possible policy is also achievable. However, all the aforementioned works are limited to tabular MDPs. There is also a notable line of work on the estimation of the stationary distribution ratio between the target policy and the behavior policy using a primal-dual formulation [30, 51, 8]. However, a theoretical guarantee for the OPE error is not given in the work. More recently, Chen et al. [7] studied OPE in the *infinite-horizon* setting with linear function approximation.

There are many others topics related to OPE, for example, policy gradient [20, 31, 2], conservative policy iteration [19], off-policy temporal-difference learning [35], off-policy Q-learning [23], safe policy iteration [33] and pessimism in RL [22, 18], to mention a few. We refer the reader to the excellent survey by Levine et al. [25] for a more detailed introduction.

**Online RL with linear function approximation.** RL with function approximation has been actively studied as an extension of the tabular setting. Yang and Wang [46] studied discounted linear MDPs with a generative model, and Jin et al. [17] proposed an efficient LSVI-UCB algorithm for linear MDPs without a generative model. It has been shown by Du et al. [9] that MDP with misspecified linear function approximation could be exponentially hard to learn. Linear MDPs under various settings have also been studied by [50, 32, 14, 44].

A parallel line of work studies linear mixture MDPs [15, 3, 5, 53, 29] (a.k.a., linear kernel MDPs [54]) where the transition kernel is a linear function of a ternary feature mapping $\psi : \mathcal{S} \times \mathcal{A} \times \mathcal{S} \to \mathbb{R}^d$. In particular, Zhou et al. [53] achieved a nearly minimax regret bound by carefully utilizing the variance information of the value functions. Zhang et al. [52] constructed a variance-aware confidence set for time-homogeneous linear mixture MDPs. However, both works are focused on online RL rather than offline RL. It requires novel algorithm designs to exploit the variance information for offline tasks like OPE. What's more, the analysis in the offline setting deviates a lot from that for online RL where one can easily apply the law of total variance to obtain tighter bounds.

## 7 Conclusion and Future Work

In this paper, we incorporate the variance information into OPE and propose VA-OPE, an algorithm that provably achieves tighter error bound. Our $\widetilde{O}(\sum_h (\mathbf{v}_h^\top \mathbf{\Lambda}_h^{-1} \mathbf{v}_h)^{1/2}/\sqrt{K})$ error bound has a sharper dependence on the distribution shift between the behavior policy and the target policy.

Our work suggests several promising future directions. Theoretically, it remains open to provide an instance-dependent lower bound for the OPE error. Also, beyond the linear function approximation, it is interesting to establish similar results under more general function approximation schemes. Empirically, can we exploit the algorithmic insight of our algorithm to develop practically more

data-effective OPE algorithms for complex real-world RL tasks? We wish to explore these directions in the future.

## Acknowledgments and Disclosure of Funding

We thank Mengdi Wang and Yaqi Duan for helpful discussions during the preparation of the paper. We also thank the anonymous reviewers for their helpful comments. DZ and QG are partially supported by the National Science Foundation CAREER Award 1906169, IIS-1904183 and AWS Machine Learning Research Award. The views and conclusions contained in this paper are those of the authors and should not be interpreted as representing any funding agencies.

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
