# A Details of the Experiments

## A.1 A Synthetic Linear MDP Example

We construct a synthetic linear MDP example based on a hard example proposed in Section 5 in [10], which was used to illustrate their lower bound. However, the feature dimension $d = 2$ in their illustrative example is too small to show discrepancy between our algorithm and theirs. Therefore, we construct an example sharing a similar structure but of much larger feature dimension and size of action space.

**MDP instance.** In specific, our MDP instance contains $|\mathcal{S}| = 2$ states and $|\mathcal{A}| = 100$ actions, and the feature dimension is $d = 10$. We denote $\mathcal{S} = \{0, 1\}$ and $\mathcal{A} = \{0, 1, \ldots, 99\}$ respectively. For each action $a \in [99]$, we represent it by a binary encoding vector $\mathbf{a} \in \mathbb{R}^8$ with each entry being either $1$ or $-1$. With a slight abuse of notation, we interchangebly use $a$ and and its vector representation $\mathbf{a}$.

We define

$$\delta(s, a) = \begin{cases} 1 & \text{if } \mathbb{1}\{s = 0\} = \mathbb{1}\{a = 0\}, \\ 0 & \text{otherwise.} \end{cases}$$

Then the feature mapping is given by

$$\phi(s, a) = (\mathbf{a}^\top, \delta(s, a), 1 - \delta(s, a))^\top \in \mathbb{R}^{10}.$$

Let $\{\alpha_h\}_{h \in [H]}$ be a sequence of integers taking values in $\{0, 1\}$. For each $s \in \mathcal{S}$, the vector-valued measures are defined as

$$\boldsymbol{\mu}_h(s) = (0, \ldots, 0, (1 - s) \oplus \alpha_h, s \oplus \alpha_h)$$

for all $h \in [H]$, where $\oplus$ denotes the 'XOR' sign. Finally, we define $\boldsymbol{\gamma}_h \equiv \boldsymbol{\gamma} = (0, \ldots, 0, 1, 0) \in \mathbb{R}^{10}$. Thus the transition is $\mathbb{P}_h(s' \mid s, a) = \langle \phi(s, a), \boldsymbol{\mu}_h(s') \rangle$ and the expected reward is $r_h(s, a) = \langle \phi(s, a), \boldsymbol{\gamma} \rangle$. It is straightforward to verify that this is a valid time-inhomogeneous linear MDP.

**Behavior and target policy.** The target policy is given by $\pi(s) = 0$ for both $s = 0, 1$. The behavior policy is determined by a parameter $p \in (0, 1)$: with probability $1 - p$, the behavior policy chooses $a = 0$, and with probability $(1 - p)/99$ it chooses $a = i$ for each $i \in [99]$. This $p$ can be used to control the distribution shift between the behavior and target policies. Note that $p$ close to $0$ induces small distribution shift, while larger $p$ leads to large distribution shift. Moreover, we set the initial distribution $\xi_1$ to be uniform over $\mathcal{S}$.

We remark that in our implementation of VA-OPE we do *not* apply data splitting, i.e., $\mathcal{D} = \check{\mathcal{D}}$ and therefore no data is wasted. As is mentioned in the main text, the only purpose of the data splitting is to avoid an otherwise lengthy theoretical analysis. Therefore, for each fixed $K$, both algorithms use a dataset of size $K$ sampled under the behavior policy.

## A.2 Impact of the Planning Horizon

We first study the impact of the planning horizon $H$ on the performance. We run our algorithm VA-OPE and the baseline method FQI-OPE with $\lambda = 1$ on the linear MDP instance constructed in the previous subsection under different values of $H$. We fix the initial distribution to be $\xi_1 = [1/2, 1/2]$ and $p$ to be $0.6$. The results are reported in Figure 2.

To explain the results, let us first recall the dominant term in our error bound and that in [10] (ignoring the logarithmic and constant factors):

$$D_{\texttt{VA}} = \frac{\sum_{h=1}^{H} \|\mathbf{v}_h\|_{\boldsymbol{\Lambda}_h^{-1}}}{\sqrt{K}} \qquad \text{vs} \qquad D_{\texttt{FQI}} = \frac{\sum_{h=1}^{H} (H - h + 1)\|\mathbf{v}_h^\pi\|_{\boldsymbol{\Sigma}_h^{-1}}}{\sqrt{K}}. \tag{A.1}$$

As mentioned in the discussion following Theorem 4.1, it holds that $D_{\texttt{VA}} \leq D_{\texttt{FQI}}$. Indeed, this is reflected by the error plots where the error of VA-OPE is smaller than that of FQI-OPE except for very small $K$.

Moreover, as careful readers may have already observed, the discrepancy between $D_{\texttt{VA}}$ and $D_{\texttt{FQI}}$ would be amplified as the value of $H$ increases. Again, our simulation results confirm this theoretical

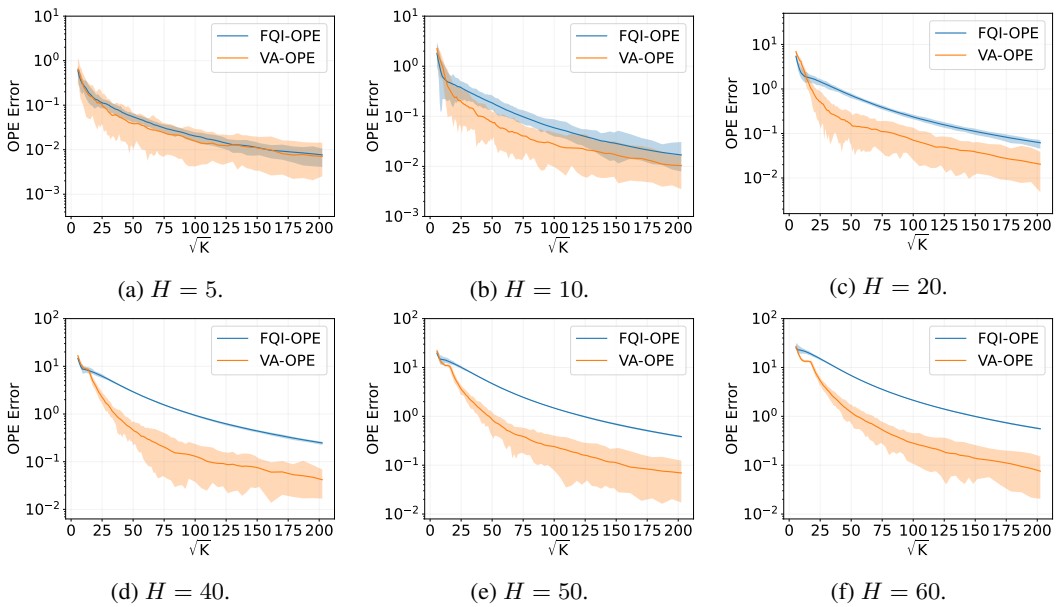

Figure 2: OPE error vs. $\sqrt{K}$. The results are averaged over 50 trials and the error bars are chosen to be the empirical [10%, 90%] confidence intervals. For a proper comparison, each sub-plot corresponds to a different setting of $H$ by keeping everything else the same: $|\mathcal{S}| = 2$, $|\mathcal{A}| = 100$, $p = 0.6$.

observation as we can see by comparing the subplots of Figure 2. For larger values of $H$, VA-OPE tends to enjoy a much faster convergence rate. We would like to emphasize that this performance gain is especially beneficial for long-horizon tasks.

These findings also shed light on the minimax optimality of the OPE problem. The previous FQI-OPE algorithm is nearly minimax optimal only for a subclass of linear MDPs where $\mathbb{V}_h V_{h+1}^\pi = \Theta((H - h)^2)$. As suggested by our theory and confirmed by the numerical experiments, our algorithm VA-OPE achieves a tighter instance-dependent error for general linear MDPs. We would like to establish the universal minimax lower bound in the future work, and we believe that VA-OPE is a promising candidate for achieving minimax optimality.

We would also like to remark that the width of the error bars of VA-OPE is similar to that of FQI-OPE. It only appears wider on the plots since the y-axis is $\log_{10}$-scaled.

### A.3 Impact of Distribution Shift

We also illustrate the impact of distribution shift between the behavior policy and the target policy on the performance, which can be controlled by the value of $p$. In Figure 3[3], we compare the performance of VA-OPE and FQI-OPE under different values of $p$.

The subplots in the same row share the same value of $H$. It is clear that for larger distribution shift, the performance of VA-OPE is superior. The reason behind this is that for fixed $H$, the ratio $D_{\texttt{FQI}}/D_{\texttt{VA}}$ increases as $p$ increases. We further investigate this in the next subsection.

### A.4 Comparison of the Dominant Terms

Finally we compare the dominant terms in the error upper bound of VA-OPE and FQI-OPE as defined in (A.1). Since both $D_{\texttt{VA}}$ and $D_{\texttt{FQI}}$ are theoretical values as the expectation over the occupancy measure induced by the transition kernel and the behavior/target policy, we simply estimate them by averaging over 1,000,000 independent trajectories. As presented in Figure 4, our characterization

---

[3]Note that the range of the y-axis differs among different rows.

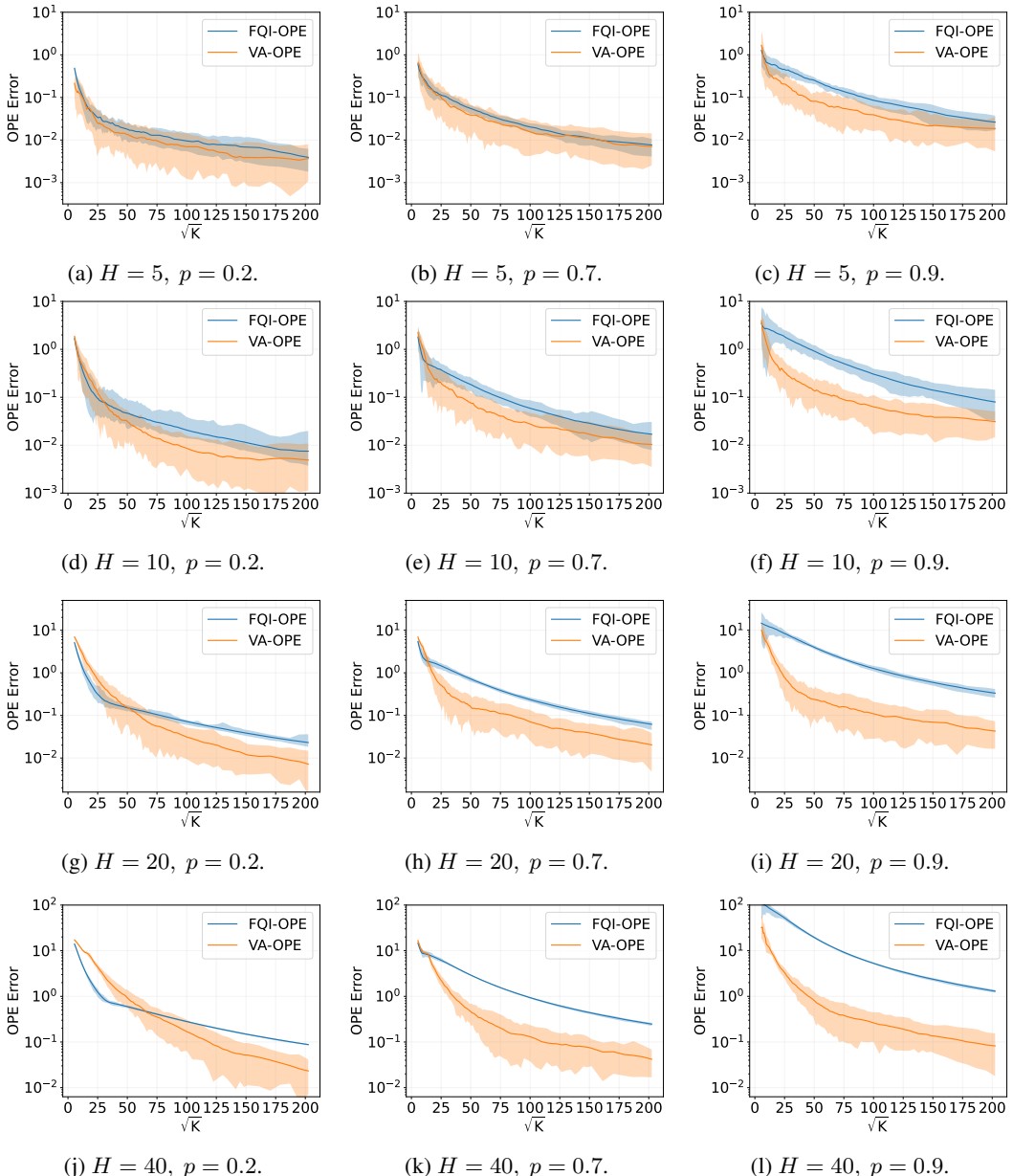

Figure 3: Log-scaled OPE error vs. $\sqrt{K}$ under different levels of distribution shift and horizon $H$. The level of distribution shift is controlled by the parameter $p$, where larger $p$ corresponds to larger distribution shift.

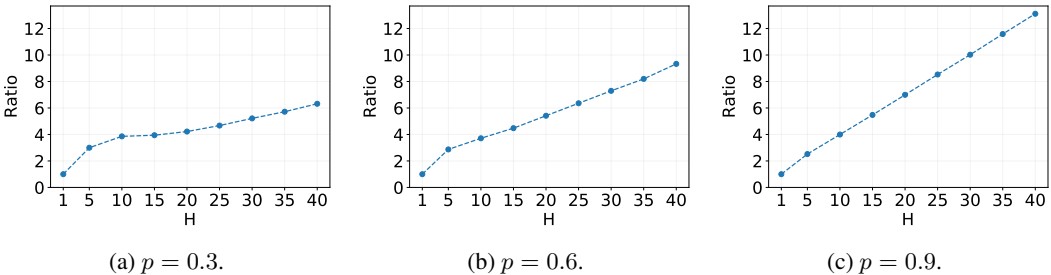

(a) $p = 0.3$.         (b) $p = 0.6$.         (c) $p = 0.9$.

Figure 4: Ratio between dominant terms vs. $H$. The results are generated by averaging over 1,000,000 trajectories.

### A.5 Hardware Details

All experiments are performed on an internal cluster with CPU and 30 GB of memory.

## B Further Comparison with Duan et al. [10]

Consider the dominant term in the OPE error (omiting the logarithmic coefficients) given by Theorem 2 in Duan et al. [10], which was shown to be $\sum_{h=1}^{H}(H - h + 1)\|\mathbf{v}_h^\pi\|_{\Sigma_h^{-1}}/\sqrt{K}$ from their proof. As comparison, recall that our dominant term is about $\sum_{h=1}^{H}\|\mathbf{v}_h^\pi\|_{\Lambda_h^{-1}}/\sqrt{K}$. The definition of $\Sigma_h$ in (2.5) and that of $\Lambda_h$ in (2.6) immediately imply $\Sigma_h \preceq [(H - h + 1)^2 + 1] \cdot \Lambda_h$ as $\sigma_h^2$ is bounded above by $(H - h + 1)^2 + 1$. Therefore, it holds that

$$\sum_{h=1}^{H}(H - h + 1)\|\mathbf{v}_h^\pi\|_{\Sigma_h^{-1}} \geq \sum_{h=1}^{H}\frac{(H - h + 1)\|\mathbf{v}_h^\pi\|_{\Lambda_h^{-1}}}{\sqrt{(H - h + 1)^2 + 1}}. \tag{B.1}$$

The RHS of (B.1) is close to $\sum_{h=1}^{H}\|\mathbf{v}_h^\pi\|_{\Lambda_h^{-1}}$ if $H$ is large. Moreover, when $\mathbb{V}_h V_{h+1}^\pi$ is small, the RHS of (B.1) can be much smaller than the LHS with appropriate choice of $\eta_h$. In other words, our bound is tighter than that of Duan et al. [10] in all scenarios, especially when $\mathbb{V}_h V_{h+1}^\pi$ is small.

Consider, for example, a scenario where the conditional variance of $V_h^\pi$, $h \in [H]$ is less than $H - h + 1$, which is smaller than the crude upper bound of $(H - h + 1)^2$ by a factor of $(H - h + 1)$. Then by choosing $\eta_h = 1$ and $\sigma_r = 1$, we would have $\sigma_h^2 \equiv H - h + 2$, and

$$\frac{(H - h + 1)\|\mathbf{v}_h^\pi\|_{\Sigma_h^{-1}}}{\|\mathbf{v}_h^\pi\|_{\Lambda_h^{-1}}} = \frac{H - h + 1}{\sqrt{H - h + 2}},$$

which suggests that $\|\mathbf{v}_h^\pi\|_{\Lambda_h^{-1}}$ is smaller than its counterpart by a factor of $(H - h + 1)/\sqrt{H - h + 2}$. Also, as mentioned in the main text, the conditional variance of $V_h^\pi$ does **not** need to be uniformly smaller than $H - h + 1$ for all $(s, a) \in \mathcal{S} \times \mathcal{A}$. It only needs to be small on average.

Regarding their lower bound (Theorem 3), it only holds for a subclass of all MDP instances where the conditional variance $\mathbb{V}_h V_{h+1}^\pi$ is on the order of $(H - h + 1)^2$. Indeed, the theorem assumes there exists a high-value subset of states $\overline{\mathcal{S}}$ and a low-value subset of states $\underline{\mathcal{S}}$ under the target policy $\pi$ such that $V_{h+1}^\pi(s) \geq \frac{3}{4}(H - h + 1)$ if $s \in \overline{\mathcal{S}}$ and $V_{h+1}^\pi(s) \leq \frac{1}{4}(H - h + 1)$ if $s \in \underline{\mathcal{S}}$. They also require there is non-zero probability $\overline{p} \geq c > 0$ and $\underline{p} \geq c > 0$ of transiting into $\overline{\mathcal{S}}$ and $\underline{\mathcal{S}}$ respectively. These assumptions immediately imply $\mathbb{V}_h V_{h+1}^\pi = \Omega((H - h + 1)^2)$. Therefore, the prior result is only (nearly) minimax for a very small class of MDPs. This is confirmed by our numerical experiments in Appendix A where we compare the OPE error of `VA-OPE` and `FQI-OPE` under different settings of $H$. The results show that `VA-OPE`'s advantage over `FQI-OPE` increases as $H$ becomes larger. It thus remains open to derive an instance-dependent lower bound that matches our upper bound.

## C The Uniform Convergence Result

### C.1 Important Remark

Throughout the appendix, we consider and analyze a slightly more general form of Algorithm 1. We now explain.

Recall that in (2.7), we define $\sigma_h(\cdot,\cdot)$ as

$$\sigma_h(s,a) = \sqrt{\max\{1, \mathbb{V}_h V_{h+1}^\pi(s,a)\} + 1},$$

and the corresponding estimator is given by

$$\widehat{\sigma}_h(\cdot,\cdot) \leftarrow \sqrt{\max\{1, \widehat{\mathbb{V}}_h \widehat{V}_{h+1}^\pi(\cdot,\cdot)\} + 1}.$$

Here taking maximum with 1 is to deal with the situation where $\widehat{\mathbb{V}}_h \widehat{V}_{h+1}^\pi(\cdot,\cdot)$ is close to zero or negative, and the second 1 is to account for the variance of the rewards. Now as a more general scheme, we replace both with adjustable parameters: $\eta_h$ and $\sigma_r^2$ such that $\eta_h \geq 1$ and $0 \leq \sigma_r \leq 1$. Thereby, for each $h \in [H]$, we have

$$\widehat{\sigma}_h(\cdot,\cdot) \leftarrow \sqrt{\max\{\eta_h, \widehat{\mathbb{V}}_h \widehat{V}_{h+1}^\pi(\cdot,\cdot)\} + \sigma_r^2}.$$

We allow the flexibility of the choices of $\{\eta_h\}_{h\in[H]}$ and $\sigma_r$ in part for generality and theoretical interests. These parameters will appear in the final results for the uniform convergence and the OPE error bound. The general algorithm is then presented as in Algorithm 2.

---

**Algorithm 2** VA-OPE (general form)

1: **Input:** target policy $\pi = \{\pi_h\}_{h\in[H]}$, datasets $\mathcal{D} = \{\{(s_{k,h}, a_{k,h}, r_{k,h}, s'_{k,h})\}_{h\in[H]}\}_{k\in[K]}$ and $\check{\mathcal{D}} = \{\{(\check{s}_{k,h}, \check{a}_{k,h}, \check{r}_{k,h}, \check{s}'_{k,h})\}_{h\in[H]}\}_{k\in[K]}$, initial distribution $\xi_1$, $\widehat{\mathbf{w}}_{H+1}^\pi = 0$, $\lambda$, $\sigma_r$, $\{\eta_h\}_{h\in[H]}$

2: **for** $h = H, H-1, \ldots, 1$ **do**
3:     $\widehat{\boldsymbol{\Sigma}}_h \leftarrow \sum_{k=1}^K \check{\boldsymbol{\phi}}_{k,h} \check{\boldsymbol{\phi}}_{k,h}^\top + \lambda \mathbf{I}_d$
4:     $\widehat{\boldsymbol{\beta}}_h \leftarrow \widehat{\boldsymbol{\Sigma}}_h^{-1} \sum_{k=1}^K \check{\boldsymbol{\phi}}_{k,h} \widehat{V}_{h+1}^\pi(\check{s}'_{k,h})^2$
5:     $\widehat{\boldsymbol{\theta}}_h \leftarrow \widehat{\boldsymbol{\Sigma}}_h^{-1} \sum_{k=1}^K \check{\boldsymbol{\phi}}_{k,h} \widehat{V}_{h+1}^\pi(\check{s}'_{k,h})$
6:     $\widehat{\sigma}_h(\cdot,\cdot) \leftarrow \sqrt{\max\{\eta_h, \widehat{\mathbb{V}}_h \widehat{V}_{h+1}^\pi(\cdot,\cdot)\} + \sigma_r^2}$
7:     $\widehat{\boldsymbol{\Lambda}}_h \leftarrow \sum_{k=1}^K \boldsymbol{\phi}_{k,h} \boldsymbol{\phi}_{k,h}^\top / \widehat{\sigma}_{k,h}^2 + \lambda \mathbf{I}_d$
8:     $Y_{k,h} \leftarrow r_{k,h} + \langle \boldsymbol{\phi}_h^\pi(s'_{k,h}), \widehat{\mathbf{w}}_{h+1}^\pi \rangle$
9:     $\widehat{\mathbf{w}}_h^\pi \leftarrow \widehat{\boldsymbol{\Lambda}}_h^{-1} \sum_{k=1}^K \boldsymbol{\phi}_{k,h} Y_{k,h} / \widehat{\sigma}_{k,h}^2$
10:    $\widehat{Q}_h^\pi(\cdot,\cdot) \leftarrow \langle \boldsymbol{\phi}(\cdot,\cdot), \widehat{\mathbf{w}}_h^\pi \rangle$,   $\widehat{V}_h^\pi(\cdot) \leftarrow \langle \boldsymbol{\phi}_h^\pi(\cdot), \widehat{\mathbf{w}}_h^\pi \rangle$
11: **end for**
12: **Output:** $\widehat{v}_1^\pi \leftarrow \int_{\mathcal{S}} \widehat{V}_1^\pi(s) \, \mathrm{d}\xi_1(s)$

---

Correspondingly, throughout the appendix we redefine for each $h \in [H]$:

$$\sigma_h(s,a) = \sqrt{\max\{\eta_h, \mathbb{V}_h V_{h+1}^\pi(s,a)\} + \sigma_r^2}, \tag{C.1}$$

and thus $\boldsymbol{\Lambda}_h$ defined in (2.6) also becomes $(\eta_h, \sigma_r^2)$-related.

Besides generality, this is actually also meaningful, because let's consider, for example, a situation where the agent actually knows that the reward is deterministic (i.e. there is no noise in the observed reward). Then the agent can choose $\sigma_r = 0$ (though this will not give a huge boost to the OPE error bound since the determinant factor in $\|\mathbf{v}_h^\pi\|_{\boldsymbol{\Lambda}_h^{-1}}$ is the variance $\mathbb{V}_h V_{h+1}^\pi$).

### C.2 Recap of Notations

Before presenting the theorems and proof, let's walk through the algorithm and remind the readers of the notations.

**Variance estimation** Recall the dataset $\check{\mathcal{D}} = \{\check{\mathcal{D}}_h\}_{h\in[H]}$, where $\check{\mathcal{D}}_h = \{(\check{s}_{k,h}, \check{a}_{k,h}, \check{r}_{k,h}, \check{s}'_{k,h})\}_{k\in[K]}$. For each $h$, the dataset $\check{\mathcal{D}}_h$ is used to compute the function $\hat{\sigma}_h(\cdot,\cdot)$, which is an estimator for the conditional variance of $V_{h+1}^\pi$. To be more clear, let go through the inner loop of Algorithm 2.

In the main text, due to the space limit, we use the abbreviation:
$$\check{\phi}_{k,h} = \phi(\check{s}_{k,h}, \check{a}_{k,h}).$$

For each $h$, the (biased and un-normalized) sample covariance matrix $\widehat{\boldsymbol{\Sigma}}_h$ (line 3) is given as
$$\widehat{\boldsymbol{\Sigma}}_h = \sum_{k=1}^K \check{\phi}_{k,h}\check{\phi}_{k,h}^\top + \lambda \mathbf{I}_d,$$

and its normalized population counterpart $\boldsymbol{\Sigma}_h$ is defined by (2.5) as
$$\boldsymbol{\Sigma}_h = \mathbb{E}_{\bar{\pi},h}\left[\phi(s,a)\phi(s,a)^\top\right].$$

Then the Algorithm computes $\widehat{\mathbb{V}}_h \widehat{V}_{h+1}^\pi$, which is an estimator of $\mathbb{V}_h V_{h+1}^\pi$, as the following:
$$[\widehat{\mathbb{V}}_h \widehat{V}_{h+1}^\pi](\cdot,\cdot) = \langle \phi(\cdot,\cdot), \widehat{\boldsymbol{\beta}}_h^\pi \rangle_{[0,(H-h+1)^2]} - \left[\langle \phi(\cdot,\cdot), \widehat{\boldsymbol{\theta}}_h^\pi \rangle_{[0,H-h+1]}\right]^2,$$

where $\widehat{\boldsymbol{\beta}}_h^\pi$ and $\widehat{\boldsymbol{\theta}}_h^\pi$ are computed in Algorithm 2 based on the estimated value function $\widehat{V}_{h+1}^\pi$ from last iteration, and the dataset $\check{D}_h$. Finally, the function $\hat{\sigma}_h$ is computed.

**Value function estimation** Once we have the variance estimator $\hat{\sigma}_h$, we can apply weighted regression to estimate the value function $V_h^\pi$ using the dataset $\mathcal{D}_h = \{(s_{k,h}, a_{k,h}, r_{k,h}, s'_{k,h})\}_{k\in[K]}$. This is described by line 7 to 10 in Algorithm 2. Please note that we have adopt the abbreviation:
$$\phi_{k,h} = \phi(s_{k,h}, a_{k,h}), \ \hat{\sigma}_{k,h} = \hat{\sigma}_h(s_{k,h}, a_{k,h}).$$

Note that the weighted sample covariance matrix $\widehat{\boldsymbol{\Lambda}}_h$ in Algorithm 2 is given as
$$\widehat{\boldsymbol{\Lambda}}_h = \sum_{k=1}^K \phi_{k,h}\phi_{k,h}^\top / \hat{\sigma}_{k,h}^2 + \lambda \mathbf{I}_d,$$

with its normalized population counterpart $\boldsymbol{\Lambda}_h$ defined by (2.6) as
$$\boldsymbol{\Lambda}_h = \mathbb{E}_{\bar{\pi},h}\left[\sigma_h(s,a)^{-2}\phi(s,a)\phi(s,a)^\top\right].$$

Also note that in the offline dataset $\mathcal{D}$, for each $\mathcal{D}_h$ and the data point $(s_{k,h}, a_{k,h}, r_{k,h}, s'_{k,h})$ in $\mathcal{D}_h$, the reward $r_{k,h}$ is the random reward given by $r_{k,h} = r_h(s_{k,h}, a_{k,h}) + \epsilon_{k,h}$, where $r_h(\cdot,\cdot)$ is an unknown deterministic function representing the (conditional) mean and $\epsilon_{k,h}$ is some independent random noise. We only observe $r_{k,h}$ and not $\epsilon_{k,h}$.

**Function classes** Based on this characterization of the value functions, we define the following function class for each $h \in [H]$ and $L > 0$:
$$\mathcal{V}_h(L) := \left\{ V(s) = \langle \phi_h^\pi(s), \mathbf{w} \rangle \middle| \mathbf{w} \in \mathbb{R}^d, \|\mathbf{w}\|_2 \leq L, \sup_{s\in\mathcal{S}} |V(s)| \leq H - h + 2 \right\}. \tag{C.2}$$

One can see that functions in $\mathcal{V}_h(L)$ are parametrized by vectors $\mathbf{w} \in \mathbb{R}^d$. From Proposition 2.2, it is clear that $V_h^\pi \in \mathcal{V}_h(2H\sqrt{d})$ for all $h \in [H]$.

We define the following function class for each $h \in [H]$ and $L_1, L_2 > 0$:
$$\mathcal{T}_h(L_1, L_2)$$
$$:= \left\{ \sigma(\cdot,\cdot) = \sqrt{\max\left\{\eta_h, \langle \phi(\cdot,\cdot), \boldsymbol{\beta} \rangle_{[0,(H-h+1)^2]} + \left[\langle \phi(\cdot,\cdot), \boldsymbol{\theta} \rangle_{[0,H-h+1]}\right]^2\right\} + \sigma_r^2} \ \middle| \ \|\boldsymbol{\beta}\| \leq L_1, \|\boldsymbol{\theta}\| \leq L_2 \right\},$$
$$\tag{C.3}$$

which is parametrized by $\boldsymbol{\beta}, \boldsymbol{\theta} \in \mathbb{R}^d$. Later we will see that, with high probability, for all $h \in [H]$, we have $\hat{\sigma}_h \in T_h(L_1, L_2)$ with above choice of $L_1 = H^2 \sqrt{\frac{Kd}{\lambda}}$ and $L_2 = H\sqrt{\frac{Kd}{\lambda}}$, which is an immediate result by Theorem C.2 and Lemma H.15. Also note that $\sigma_h \in \mathcal{T}_h(L_1, L_2)$, which is clear from (3.4).

## C.3 Formal Statement of Uniform Convergence Theorem

**A weaker data sampling assumption.** Recall Assumption 2.5 on the data sampling process introduced in the main text. It turns out that the uniform convergence result (Theorem C.2) holds under a weaker assumption which is the following.

**Assumption C.1** (Trajectory-sampling Data). We have two offline datasets $\mathcal{D}$ and $\check{\mathcal{D}}$ where each dataset consists of $K$ trajectories with horizon length equal to $H$. Each trajectory is independently generated by the behavior policy $\bar{\pi}$. That is, $\mathcal{D} = \{\mathcal{D}_k\}_{k \in [K]}$, where each $\mathcal{D}_k$ is given by $\mathcal{D}_k = \{(s_{k,h}, a_{k,h}, r_{k,h})\}_{h \in [H]}$ such that $a_{k,h} \sim \pi_h(\cdot|s_{k,h})$ and $s_{k,h+1} \sim \mathbb{P}_h(\cdot|s_{k,h}, a_{k,h})$. For each $(k, h) \in [K] \times [H]$, the random reward $r_{k,h} = r_h(s_{k,h}, a_{k,h}) + \epsilon_{k,h}$, where $r_h(s_{k,h}, a_{k,h})$ is the (unknown) expected reward and $\epsilon_{k,h}$ is the noise. Similarly, we have $\check{\mathcal{D}} = \{\check{\mathcal{D}}_k\}_{k \in [K]}$, where $\check{\mathcal{D}}_k = \{(\check{s}_{k,h}, \check{a}_{k,h}, \check{r}_{k,h})\}_{h \in [H]}$. Here we denote $s'_{k,h} = s_{k,h+1}$ for simplicity.

Note that Assumption 2.5 is stronger than Assumption C.1 in the sense that Assumption 2.5 assumes an extra independence between the data points sampled at different stages. Therefore, as will be clear from the proof, since Theorem C.2 is established under Assumption C.1, it automatically holds under the stronger Assumption 2.5.

We now introduce the uniform convergence theorem. To simplify the notation, we define:

$$C_{h,1} = \sum_{i=h}^{H} \frac{1}{\iota_h}, \quad C_{h,2} = \sum_{i=h}^{H} \sqrt{\frac{C_{h,3}}{\iota_h}}, \quad C_{h,3} = \frac{(H - h + 1)^2}{\eta_h + \sigma_r^2}, \quad C_{h,4} = \left(\|\mathbf{\Lambda}_h\| \cdot \|\mathbf{\Lambda}_h^{-1}\|\right)^{1/2}.$$

Note that by setting $\eta_h = \sigma_r = 1$ we recover the same $C_{h,2}, C_{h,3}$ as in the main text.

**Theorem C.2** (Uniform Convergence). Set $\lambda = 1$ and $\eta_h \in (0, (H - h + 1)^2]$ for all $h \in [H]$ in Algorithm 1. Under Assumption 2.1, 2.3 and C.1, there exists some universal constant $C$ such that if $K$ satisfies

$$K \geq C \cdot \frac{H^2 d^2}{\kappa^2} \log\left(\frac{dHK}{\kappa\delta}\right) \cdot \max_{h \in [H]} \frac{(H - h + 1)^2}{(\eta_h + \sigma_r^2)^2} \cdot \max_{h \in [H]} \frac{(H - h + 1)^2}{\iota_h(\eta_h + \sigma_r^2)}, \tag{C.4}$$

then with probability at least $1 - \delta$, it holds for all $h \in [H]$ that $\sup_{s \in \mathcal{S}} \left|\widehat{V}_h^\pi(s)\right| \leq H - h + 2$, and

$$\sup_{s \in \mathcal{S}} \left|\widehat{V}_h^\pi(s) - V_h^\pi(s)\right| \leq C \cdot \frac{C_{h,2} d}{\sqrt{K}} \log\left(\frac{dH^2 K}{\kappa\delta}\right) + C \cdot \frac{C_{h,1} H \sqrt{d}}{K}.$$

We now present the proof of Theorem C.2. The proof relies on a backward induction argument, i.e., we will show $|\widehat{V}_h^\pi(s) - V_h^\pi(s)|$ is uniformly small for $h = H, H - 1, \cdots, 1$. For this purpose, we need to use the first form of error decomposition (4.2) in Lemma 4.4.

## C.4 Step 1: Base Case at Stage $h = H$

We first bound the approximation error at the last stage $h = H$. From the algorithm we have $\widehat{V}_{H+1}^\pi \equiv V_{H+1}^\pi \equiv 0$. Therefore, we have $\widehat{\boldsymbol{\theta}}_H^\pi = \widehat{\boldsymbol{\beta}}_H = 0, \widehat{\sigma}_H \equiv \sqrt{\eta_H + \sigma_r^2}$, and

$$\widehat{\mathbf{\Lambda}}_H = \frac{1}{\eta_H + \sigma_r^2} \sum_{k=1}^{K} \boldsymbol{\phi}(s_{k,H}, a_{k,H}) \boldsymbol{\phi}(s_{k,H}, a_{k,H})^\top + \lambda \mathbf{I}_d.$$

By the error decomposition in (4.2), we have

$$V_H^\pi(s) - \widehat{V}_H^\pi(s) = \underbrace{-\boldsymbol{\phi}_H^\pi(s)^\top \widehat{\mathbf{\Lambda}}_H^{-1} \sum_{k=1}^{K} \frac{\boldsymbol{\phi}(s_{k,H}, a_{k,H})}{\widehat{\sigma}_H(s_{k,H}, a_{k,H})^2} \epsilon_{k,H}}_{\Delta_1} + \underbrace{\lambda \boldsymbol{\phi}_H^\pi(s)^\top \widehat{\mathbf{\Lambda}}_H^{-1} \mathbf{w}_H^\pi}_{\Delta_2}. \tag{C.5}$$

We will bound the two terms separately.

To bound $|\Delta_1|$, we first apply Cauchy-Schwartz inequality to obtain that

$$|\Delta_1| \leq \|\boldsymbol{\phi}_H^\pi(s)\|_{\widehat{\mathbf{\Lambda}}_H^{-1}} \cdot \left\|\sum_{k=1}^{K} \frac{\boldsymbol{\phi}(s_{k,H}, a_{k,H})}{\widehat{\sigma}_H(s_{k,H}, a_{k,H})^2} \epsilon_{k,H}\right\|_{\widehat{\mathbf{\Lambda}}_H^{-1}} \tag{C.6}$$

By Lemma H.5, with probability at least $1 - \delta$, we have

$$\|\phi_H^\pi(s)\|_{\widehat{\mathbf{\Lambda}}_H^{-1}} \leq \frac{2}{\sqrt{K}} \cdot \|\phi_H^\pi(s)\|_{\mathbf{\Lambda}_H^{-1}} \tag{C.7}$$

for all $s \in \mathcal{S}$, as long as $K$ satisfies that

$$K \geq \max\left\{ \frac{512\|\mathbf{\Lambda}_H^{-1}\|^2}{(\eta_H + \sigma_r^2)^2} \log\left(\frac{2d}{\delta}\right), 2\lambda\|\mathbf{\Lambda}_H^{-1}\| \right\} = \frac{512\|\mathbf{\Lambda}_H^{-1}\|^2}{(\eta_H + \sigma_r^2)^2} \log\left(\frac{2d}{\delta}\right).$$

Note that $\mathrm{Var}(\epsilon_{k,h}) = \sigma_r^2 \leq \widehat{\sigma}_H(s_{k,H}, a_{k,H})^2$ for all $k \in [K]$. Then by Theorem H.10 we have

$$\left\| \sum_{k=1}^K \widehat{\sigma}_H(s_{k,H}, a_{k,H})^{-2} \phi(s_{k,H}, a_{k,H})\epsilon_{k,H} \right\|_{\widehat{\mathbf{\Lambda}}_H^{-1}}$$

$$\leq 8\sqrt{d \log\left(1 + \frac{K}{\lambda d(\eta_H + \sigma_r^2)}\right) \cdot \log\left(\frac{4K^2}{\delta}\right)} + 4\sqrt{\frac{1}{\eta_H + \sigma_r^2} \log\left(\frac{4K^2}{\delta}\right)}$$

$$\leq 12\sqrt{d} \log\left(\frac{4K^2}{\delta}\right) \tag{C.8}$$

with probability at least $1 - \delta$.

Then by (C.6), it suffices to take a union bound over (C.7) and (C.8) to conclude that if $K \geq 128\|\mathbf{\Lambda}_H^{-1}\|^2 \log(2d/\delta)/(\eta_H + \sigma_r^2)$ then

$$|\Delta_1| \leq \frac{12\sqrt{d}}{\sqrt{K}} \|\phi_H^\pi(s)\|_{\mathbf{\Lambda}_H^{-1}} \cdot \log\left(\frac{4K^2}{\delta}\right) \tag{C.9}$$

with probability at least $1 - \delta$.

At the same time, we can bound $|\Delta_2|$ using the same argument.

$$|\Delta_2| \leq \lambda\|\phi_H^\pi(s)\|_{\widehat{\mathbf{\Lambda}}_H^{-1}} \cdot \|\mathbf{w}_H^\pi\|_{\widehat{\mathbf{\Lambda}}_H^{-1}} \leq \frac{4\lambda}{K} \cdot \|\phi_H^\pi(s)\|_{\mathbf{\Lambda}_H^{-1}} \cdot \|\mathbf{w}_H^\pi\|_{\mathbf{\Lambda}_H^{-1}}, \tag{C.10}$$

where the second inequality holds on the same event as does (C.9).

Finally, we combine (C.5), (C.9) and (C.10), and obtain that if $K \geq 512\|\mathbf{\Lambda}_H^{-1}\|^2/(\eta_H + \sigma_r^2)^2 \log(4d/\delta)$ then

$$\sup_{s \in \mathcal{S}} \left|V_H^\pi(s) - \widehat{V}_H^\pi(s)\right| \leq \frac{12\sqrt{d}}{\sqrt{K}} \log\left(\frac{4K^2}{\delta}\right) \cdot \sup_{s \in \mathcal{S}} \|\phi_H^\pi(s)\|_{\mathbf{\Lambda}_H^{-1}} + \frac{4\lambda}{K} \cdot \sup_{s \in \mathcal{S}} \|\phi_H^\pi(s)\|_{\mathbf{\Lambda}_H^{-1}} \cdot \|\mathbf{w}_H^\pi\|_{\mathbf{\Lambda}_H^{-1}}$$

$$\leq \frac{12\sqrt{d\|\mathbf{\Lambda}_H^{-1}\|}}{\sqrt{K}} \log\left(\frac{4K^2}{\delta}\right) + \frac{8\lambda H\sqrt{d}\|\mathbf{\Lambda}_H^{-1}\|}{K}$$

with probability at least $1 - \delta$, where the last inequality follows from Assumption 2.1, Proposition 2.2 and the choice that $\lambda = 1$. Note that since $\widehat{\sigma}_H(\cdot, \cdot) \leq 1 + \sigma_r^2$, we have $\mathbf{\Lambda}_H \succeq \mathbf{\Sigma}_H/(1 + \sigma_r^2)$, which implies that $\|\mathbf{\Lambda}_H^{-1}\| \leq 2\|\mathbf{\Sigma}_H^{-1}\|$ as $\sigma_r^2 \leq 1$. Then we further have

$$\sup_{s \in \mathcal{S}} \left|V_H^\pi(s) - \widehat{V}_H^\pi(s)\right| \leq \frac{12\sqrt{2d}}{\sqrt{K}\kappa_H} + \frac{16\lambda H^3\sqrt{d}}{K\kappa_H}$$

Meanwhile, we can bound $\sup_{s \in \mathcal{S}} |\widehat{V}_H^\pi(s)|$ as follows

$$\sup_{s \in \mathcal{S}} |\widehat{V}_H^\pi(s)| \leq \sup_{s \in \mathcal{S}} V_H^\pi(s) + \frac{12\sqrt{2d}}{\sqrt{K}\kappa_H} + \frac{16\lambda H\sqrt{d}}{K\kappa_H} \leq 2,$$

when $K$ satisfies that $K \geq 600(\lambda + 1)(d + H\sqrt{d})/\kappa_H$.

In conclusion, we have

$$\sup_{s \in \mathcal{S}} |\widehat{V}_H^\pi(s)| \leq 2,$$

and

$$\sup_{s\in\mathcal{S}}\left|V_H^\pi(s)-\widehat{V}_H^\pi(s)\right|\leq\frac{12\sqrt{2d}}{\sqrt{K}\kappa_H}+\frac{16\lambda H\sqrt{d}}{K\kappa_H},$$

given that $K$ satisfies

$$K\geq\max\left\{\frac{2048}{\kappa_H^2(\eta_H+\sigma_r^2)^2}\log\left(\frac{2d}{\delta}\right),600(\lambda+1)\frac{d+H\sqrt{d}}{\kappa_H}\right\}\tag{C.11}$$

### C.5 Step 2: Induction Hypothesis

For the induction hypothesis, we assume that if for all sufficiently large $K$, with probability at least $1-(H-h)\delta$, the following event (denoted as $\mathcal{E}_{h+1}$) holds:

$$\sup_{s\in\mathcal{S}}|\widehat{V}_{h+2}^\pi(s)|\leq H-h,\ \sup_{s\in\mathcal{S}}|\widehat{V}_{h+1}^\pi(s)|\leq H-h+1,\ \sup_s|\widehat{V}_{h+1}^\pi(s)-V_{h+1}^\pi(s)|\leq\alpha_{H-h},$$

where $\alpha_{H-h}\leq(\eta_h+\sigma_r^2)/[8(H-h+1)]$.

We claim that if $K$ satisfies

$$K\geq\frac{3600(H-h+1)^4d^2}{\kappa_h^2(\eta_h+\sigma_r^2)^2}\cdot\log\left(\frac{dHK}{\kappa_h\delta}\right)\tag{C.12}$$

then with probability at least $1-(H-h+1)\delta$, the following event (denoted by $\mathcal{E}_h$) holds:

$$|\widehat{V}_{h+1}^\pi|<H-h+1,$$
$$|\widehat{V}_h^\pi|<H-h+2,$$
$$\sup_s|\widehat{V}_h^\pi(s)-V_h^\pi(s)|\leq\left(1+\frac{8\lambda}{\iota_hK}\right)\alpha_{H-h}+\frac{2\lambda H\sqrt{d}}{\iota_hK}$$
$$+\frac{20}{\sqrt{K}}\cdot\left(\frac{d}{\sqrt{\iota_h}}+\frac{d(H-h+1)}{\sqrt{\iota_h(\eta_h+\sigma_r^2)}}\right)\cdot\log\left(\frac{d(H-h+1)^2K}{\kappa_h(\eta_h+\sigma_r^2)\delta}\right)$$

We again bound the three terms in the error decomposition (4.2) simultaneously. Let $\widetilde{\mathcal{E}}_h$ be the event given by Lemma F.7 for $h$ such that $\mathbb{P}(\widetilde{\mathcal{E}}_h)\geq1-\delta$, where we have $L=(1+1/H)d\sqrt{K/\lambda}$.

Let's consider the event $\widetilde{\mathcal{E}}_h\cap\mathcal{E}_{h+1}$, which satisfies $\mathbb{P}\{\widetilde{\mathcal{E}}_h\cap\mathcal{E}_{h+1}\}\geq1-(H-h+1)\delta$ by a union bound. Note that on $\mathcal{E}_{h+1}$, we have $|\widehat{V}_{h+1}^\pi|\leq H-h+1$. Furthermore, since $|\widehat{V}_{h+2}^\pi|\leq H-h$ on $\mathcal{E}_{h+1}$, again by Lemma H.15 with $B=H$, we see that $\widehat{V}_{h+1}^\pi\in\mathcal{V}_{h+1}(L)$. Therefore, by Lemma F.7 it holds on $\widetilde{\mathcal{E}}_h\cap\mathcal{E}_{h+1}$ that

$$\left\|\left(\frac{\widehat{\mathbf{\Lambda}}_h}{K}\right)^{-1}\right\|\leq\frac{8}{\iota_h},\tag{C.13}$$

and

$$\left|\phi(s,a)^\top\widehat{\mathbf{\Lambda}}_h^{-1}\sum_{k=1}^K\widehat{\sigma}_h(s_{k,h},a_{k,h})^{-2}\phi(s_{k,h},a_{k,h})\left(\mathbb{P}_hV(s_{k,h},a_{k,h})-V(s_{k,h}')-\epsilon_{k,h}\right)\right|$$
$$\leq\frac{20}{\sqrt{K}}\cdot\left(\frac{d}{\sqrt{\iota_h}}+\frac{d(H-h+1)}{\sqrt{\iota_h(\eta_h+\sigma_r^2)}}\right)\cdot\log\left(\frac{d(H-h+1)^2K}{\kappa_h(\eta_h+\sigma_r^2)\delta}\right),\tag{C.14}$$

for all $(s,a)\in\mathcal{S}\times\mathcal{A}$.

Since it holds on $\mathcal{E}_{h+1}$ that $\sup_{s\in\mathcal{S}}|\widehat{V}_{h+1}^\pi(s)-V_{h+1}^\pi(s)|\leq\alpha_{H-h}$, we have

$$\sup_{s\in\mathcal{S}}\left|[\mathbb{J}_h\mathbb{P}_h(V_{h+1}^\pi-\widehat{V}_{h+1}^\pi)](s)\right|\leq\alpha_{H-h}.\tag{C.15}$$

Also by (C.13) and $\sup_{s \in \mathcal{S}} |\widehat{V}_{h+1}^\pi(s) - V_{h+1}^\pi(s)| \leq \alpha_{H-h}$, it follows from Cauchy-Schwartz inequality that

$$\sup_{s \in \mathcal{S}} \left| \lambda \boldsymbol{\phi}_h^\pi(s)^\top \widehat{\boldsymbol{\Lambda}}_h^{-1} \int_{\mathcal{S}} \left( V_{h+1}^\pi(s) - \widehat{V}_{h+1}^\pi(s) \right) \boldsymbol{\mu}_h(s) \mathrm{d}s \right| \leq \frac{8\lambda \alpha_{H-h}}{\iota_h K}. \tag{C.16}$$

Similarly, for the last term in (4.2) we have by Cauchy-Schwartz inequality that

$$\sup_{s \in \mathcal{S}} \left| \lambda \boldsymbol{\phi}_h^\pi(s)^\top \widehat{\boldsymbol{\Lambda}}_h^{-1} \mathbf{w}_h^\pi \right| \leq \lambda \sup_{s \in \mathcal{S}} \|\boldsymbol{\phi}_h^\pi(s)\|_2 \cdot \|\widehat{\boldsymbol{\Lambda}}_h^{-1}\| \cdot \|\mathbf{w}_h^\pi\|_2 \leq \frac{2\lambda H \sqrt{d}}{\iota_h K}, \tag{C.17}$$

where the second inequality follows from Proposition 2.2.

Finally, combining (C.14), (C.15), (C.16) and (C.17), we obtain by the error decomposition (4.2) that

$$\sup_{s \in \mathcal{S}} \left| V_h^\pi(s) - \widehat{V}_h^\pi(s) \right| \leq \frac{20}{\sqrt{K}} \cdot \left( \frac{d}{\sqrt{\iota_h}} + \frac{d(H-h+1)}{\sqrt{\iota_h(\eta_h + \sigma_r^2)}} \right) \cdot \log \left( \frac{d(H-h+1)^2 K}{\kappa_h(\eta_h + \sigma_r^2)\delta} \right)$$
$$+ \left( 1 + \frac{8\lambda}{\iota_h K} \right) \alpha_{H-h} + \frac{2\lambda H \sqrt{d}}{\iota_h K} \tag{C.18}$$

Note that when $K$ satisfies (C.12), we would have $\sup_{s \in \mathcal{S}} |V_h^\pi(s) - \widehat{V}_h^\pi(s)| \leq 1$, and thus

$$\sup_{s \in \mathcal{S}} |\widehat{V}_h^\pi(s)| \leq \sup_{s \in \mathcal{S}} |V_h^\pi(s)| + \sup_{s \in \mathcal{S}} \left| V_h^\pi(s) - \widehat{V}_h^\pi(s) \right| \leq H - h + 2. \tag{C.19}$$

Therefore, by (C.18) and (C.19), we conclude that $\widetilde{\mathcal{E}}_h \cap \mathcal{E}_{h+1} \subseteq \mathcal{E}_h$, which implies that

$$\mathbb{P}\{\mathcal{E}_h\} \geq \mathbb{P}\{\widetilde{\mathcal{E}}_h \cap \mathcal{E}_{h+1}\} \geq 1 - (H-h+1)\delta.$$

## C.6 Step 3: Recursion

Let $\kappa = \min_{h \in [H]} \kappa_h$. Suppose $K$ satisfies that

$$K \geq \frac{3600 H^2 d^2}{\kappa^2} \log \left( \frac{dHK}{\kappa\delta} \right) \cdot \max_{h \in [H]} \frac{(H-h+1)^2}{(\eta_h + \sigma_r^2)^2} \cdot \max_{h \in [H]} \frac{(H-h+1)^2}{\iota_h(\eta_h + \sigma_r^2)} \tag{C.20}$$

We also define the following quantity

$$\xi_{H-h} = \frac{2\lambda H \sqrt{d}}{\iota_h K} + \frac{20}{\sqrt{K}} \cdot \left( \frac{d}{\sqrt{\iota_h}} + \frac{d(H-h+1)}{\sqrt{\iota_h(\eta_h + \sigma_r^2)}} \right) \cdot \log \left( \frac{d(H-h+1)^2 K}{\kappa_h(\eta_h + \sigma_r^2)\delta} \right) \tag{C.21}$$

for all $h \in [H]$. With the choice of $K$ in (C.20), the following holds

$$\xi_{H-h} \leq \min \left\{ \frac{1}{2He}, \frac{1}{16H} \cdot \min_{h \in [H]} \frac{\eta_h + \sigma_r^2}{H-h+1} \right\} \tag{C.22}$$

for all $h \in [H]$.

First by the base case at stage $h = H$ from subsection C.4, we have

$$\sup_{s \in \mathcal{S}} \left| V_H^\pi(s) - \widehat{V}_H^\pi(s) \right| \leq \frac{12\sqrt{2d}}{\sqrt{K\kappa_H}} + \frac{16\lambda H \sqrt{d}}{K\kappa_H} := \alpha_0. \tag{C.23}$$

Also by the choice of $K$ in (C.20), we have

$$\alpha_0 \leq \min \left\{ \frac{1}{2e}, \frac{1}{16e} \cdot \min_{h \in [H]} \frac{\eta_h + \sigma_r^2}{H-h+1} \right\}. \tag{C.24}$$

Then by the induction step at stage $h = H - 1$ from subsection C.5, we have

$$\sup_{s \in \mathcal{S}} \left| V_{H-1}^\pi(s) - \widehat{V}_{H-1}^\pi(s) \right| \leq \alpha_1,$$

with

$$\alpha_1 \le \left(1 + \frac{\lambda C_{H-1} H}{K}\right)\alpha_0 + \xi_1 \le \left(1 + \frac{1}{H}\right)\alpha_0 + \xi_1 \le \min\left\{1, \frac{\eta_H + \sigma_r^2}{16}\right\}$$

We then define $\alpha_{H-h} = (1 + 1/H)\alpha_{H-h+1} + \xi_{H-h}$ recursively for all $h \in [H-1]$. Note that for all $i \in [H-h]$ we have

$$\alpha_i \le \left(1 + \frac{1}{H}\right)^i \alpha_0 + \sum_{j=0}^{i}\left(1 + \frac{1}{H}\right)^{i-j}\xi_j \le e \cdot \alpha_0 + e \cdot \sum_{h=0}^{i}\xi_j \le \min\left\{1, \frac{\eta_{H-i} + \sigma_r^2}{8(i+1)}\right\},$$

where the first inequality follows from the fact that $(1 + 1/n)^n \le e$ for all positive integer $n$, and the second inequality is due to (C.22) and (C.24).

Therefore, we may apply the induction step from the previous subsection to all $h \in [H-1]$ and obtain that

$$\sup_{s \in \mathcal{S}}\left|\widehat{V}_h^\pi\right| \le H - h + 2$$

and

$$\begin{aligned}
\sup_{s \in \mathcal{S}}\left|\widehat{V}_h^\pi(s) - V_h^\pi(s)\right| &\le \left(1 + \frac{1}{H}\right)\alpha_{H-h} + \iota_{H-h} \\
&\le \left(1 + \frac{1}{H}\right)^{H-h}\alpha_0 + \sum_{i=0}^{H-h}\left(1 + \frac{1}{H}\right)^{H-h-i}\xi_i \\
&\le e \cdot \alpha_0 + e \cdot \sum_{i=0}^{H-h}\xi_i
\end{aligned} \tag{C.25}$$

with probability at least $1 - H\delta$ simultaneously for all $h \in [H]$.

Therefore, replacing $\delta$ by $\delta/H$ and plugging (C.21) and (C.23) into (C.25), we obtain that

$$\begin{aligned}
&\sup_{s \in \mathcal{S}}\left|\widehat{V}_h^\pi(s) - V_h^\pi(s)\right| \\
&\le \frac{12e\sqrt{2d}}{\sqrt{K}\kappa_H} + \frac{16e\lambda H\sqrt{d}}{K\kappa_H} + \frac{2e\lambda H\sqrt{d}}{K}\sum_{i=h}^{H-1}\frac{1}{\iota_h} + \frac{40ed}{\sqrt{K}}\log\left(\frac{dH^2K}{\kappa\delta}\right)\sum_{i=h}^{H-1}\frac{H-h+1}{\sqrt{\iota_h(\eta_h + \sigma_r^2)}}.
\end{aligned} \tag{C.26}$$

We further define

$$C_{h,1} = \sum_{i=h}^{H}\frac{1}{\iota_h}, \quad C_{h,2} = \sum_{i=h}^{H}\frac{H-h+1}{\sqrt{\iota_h(\eta_h + \sigma_r^2)}},$$

then we can simplify and rearrange (C.26) to

$$\sup_{s \in \mathcal{S}}\left|\widehat{V}_h^\pi(s) - V_h^\pi(s)\right| \le C \cdot \frac{C_{h,2}d}{\sqrt{K}}\log\left(\frac{dH^2K}{\kappa\delta}\right) + C \cdot \frac{C_{h,1}H\sqrt{d}}{K}.$$

This completes the proof of Theorem C.2.

## D  Proof of OPE Convergence

As stated in Appendix C.1, we consider the general form of Theorem 4.1. Recall the following notation:

$$C_{h,1} = \sum_{i=h}^{H}\frac{1}{\iota_h}, \quad C_{h,2} = \sum_{i=h}^{H}\sqrt{\frac{C_{h,3}}{\iota_h}}, \quad C_{h,3} = \frac{(H-h+1)^2}{\eta_h + \sigma_r^2}, \quad C_{h,4} = \left(\|\mathbf{\Lambda}_h\| \cdot \|\mathbf{\Lambda}_h^{-1}\|\right)^{1/2}.$$

**Theorem D.1** (General form of Theorem 4.1). Set $\lambda = 1$, $\eta_h \in (0, (H - h + 1)^2]$ for all $h \in [H]$ and $\sigma_r^2 \leq 1$. Under Assumptions 2.1, 2.3 and 2.5, if $K$ satisfies

$$K \geq C \cdot C_3 \cdot d^2 \left[ \log \left( \frac{dH^2K}{\kappa\delta} \right) \right]^2, \tag{D.1}$$

where $C$ is some problem-independent universal constant and

$$C_3 := \max \left\{ \max_{h \in [H]} \frac{C_{h,3} \cdot C_{h,2}^2}{\iota_h^2 (\eta_h + \sigma_r^2)^3}, \frac{H^4}{\sigma_r^4 \kappa^2}, \frac{H^2}{\sigma_r^4 \kappa^2} \cdot \max_{h \in [H]} \frac{C_{h,3}}{\eta_h + \sigma_r^2} \cdot \max_{h \in [H]} \frac{C_{h,3}}{\iota_h} \right\}.$$

Then with probability at least $1 - \delta$, it holds that

$$|v_1^\pi - \widehat{v}_1^\pi| \leq C \cdot \left[ \sum_{h=1}^H \|\mathbf{v}_h^\pi\|_{\mathbf{\Lambda}_h^{-1}} \right] \cdot \sqrt{\frac{\log(16H/\delta)}{K}} + C \cdot C_4 \cdot \log \left( \frac{16H}{\delta} \right) \cdot \left( \frac{1}{K^{3/4}} + \frac{1}{K} \right),$$

where $C_4 := \sum_{h=1}^H \left\{ \sqrt{C_{h,4} \cdot C_{h,2} \cdot \frac{(H-h+1)d}{\iota_h(\eta_h+\sigma_r^2)^2} \cdot \log \left( \frac{dH^2K}{\kappa\delta} \right)} \cdot \|\mathbf{v}_h^\pi\|_{\mathbf{\Lambda}_h^{-1}} \right\}.$

Note that by setting $\eta_h = \sigma_r = 1$ we recover Theorem 4.1.

The proof is based on the recursive error decomposition given by (4.3) and the prerequisite result on uniform convergence. We will show the OPE convergence conditioned on the high probability event of uniform convergence established by Theorem C.2.

Recall the error decomposition for the OPE problem given by (4.3) (proof in Section E):

$$\begin{aligned}
v_1^\pi - \widehat{v}_1^\pi = &-\lambda \sum_{h=1}^H (\mathbf{v}_h^\pi)^\top \widehat{\mathbf{\Lambda}}_h^{-1} \int_{\mathcal{S}} \left( V_{h+1}^\pi(s) - \widehat{V}_{h+1}^\pi(s) \right) \boldsymbol{\mu}_h(s) \mathrm{d}s \\
&+ \sum_{h=1}^H (\mathbf{v}_h^\pi)^\top \widehat{\mathbf{\Lambda}}_h^{-1} \sum_{k=1}^K \frac{\boldsymbol{\phi}(s_{k,h}, a_{k,h})}{\widehat{\sigma}_h(s_{k,h}, a_{k,h})^2} \left( [\mathbb{P}_h \widehat{V}_{h+1}^\pi](s_{k,h}, a_{k,h}) - \widehat{V}_{h+1}^\pi(s'_{k,h}) - \epsilon_{k,h} \right) \\
&+ \lambda \sum_{h=1}^H (\mathbf{v}_h^\pi)^\top \widehat{\mathbf{\Lambda}}_h^{-1} \mathbf{w}_h^\pi \\
:= &\ E_1 + E_2 + E_3. \tag{D.2}
\end{aligned}$$

It suffices to prove that each term can be bounded with high probability and then we can take a union bound. By the result of Theorem C.2, we can condition on the event where both $\widehat{V}_{h+1}^\pi$ and $\widehat{\sigma}_h$ are good estimators of their population counterparts.

**Remark D.2.** All the lemmas in the remaining of this Section D will be proved under Assumptions 2.1 and 2.5. So we do not explicitly add these two assumptions into the description of the lemmas.

Also, recall the function classes $\mathcal{V}_h(L)$ and $\mathcal{T}_h(L_1, L_2)$ defined by (C.2) and (C.3). In the remaining of this section, we will assume $L$, $L_1$ and $L_2$ to be

$$L = \frac{H + 1}{\sqrt{\eta + \sigma_r^2}} \sqrt{\frac{Kd}{\lambda}}, \ L_1 = H^2 \sqrt{\frac{Kd}{\lambda}}, \ L_2 = H \sqrt{\frac{Kd}{\lambda}}.$$

The reason that we can make the above assumption is that, conditioning on the high probability event of uniform convergence (Theorem C.2), it follows immediately from Lemma H.15 that we have $\widehat{\sigma}_h \in \mathcal{T}_h(L_1, L_2)$, and $\widehat{V}_h^\pi \in \mathcal{V}_h(L)$ for all $h \in [H]$ with the above choice of $L$, $L_1$ and $L_2$.

## D.1 Bounding the $E_2$ Term in the OPE Decomposition

We consider the term $E_2$ first. Decompose $E_2$ into $E_2 = \sum_{h=1}^H E_{2,h}$ where for each $h \in [H]$, $E_{2,h}$ is given as

$$E_{2,h} := (\mathbf{v}_h^\pi)^\top \widehat{\mathbf{\Lambda}}_h^{-1} \sum_{k=1}^K \frac{\boldsymbol{\phi}(s_{k,h}, a_{k,h})}{\widehat{\sigma}_h(s_{k,h}, a_{k,h})^2} \left( [\mathbb{P}_h \widehat{V}_{h+1}^\pi](s_{k,h}, a_{k,h}) - \widehat{V}_{h+1}^\pi(s'_{k,h}) - \epsilon_{k,h} \right).$$

Further decompose $E_{2,h}$ into $E_{2,h} = \sum_{k=1}^{K} e_{h,k}$ where

$$e_{h,k} = (\mathbf{v}_h^\pi)^\top \widehat{\mathbf{\Lambda}}_h^{-1} \frac{\phi(s_{k,h}, a_{k,h})}{\widehat{\sigma}_h(s_{k,h}, a_{k,h})^2} \left( [\mathbb{P}_h \widehat{V}_{h+1}^\pi](s_{k,h}, a_{k,h}) - \widehat{V}_{h+1}^\pi(s'_{k,h}) - \epsilon_{k,h} \right).$$

In the following lemma, we consider the term $E_{2,h}$ for arbitrarily fixed $h \in [H]$. To simplify the notation, we omit the subscript $h$ and take $e_k = e_{h,k}$ once $h$ is fixed.

**Lemma D.3.** For any $h \in [H]$, condition on $\widehat{V}_{h+1}^\pi \in \mathcal{V}_{h+1}(L)$ and the induced $\widehat{\sigma}_h(\cdot, \cdot) \in \mathcal{T}_h(L_1, L_2)$ being fixed, such that $\widehat{\sigma}_h$ satisfies for all $(s, a)$

$$\left| \widehat{\sigma}_h^2(s, a) - \sigma_r^2 - \max\left\{ \eta_h, \mathbb{V}_h \widehat{V}_{h+1}^\pi(s, a) \right\} \right| \leq \frac{C(H - h + 1)^2 \sqrt{d}}{\sqrt{K}}, \tag{D.3}$$

for some $C > 0$. If $K$ satisfies (D.7), then with conditional probability at least $1 - \delta$, we have

$$|E_{2,h}| \leq 2\sqrt{2\log\left(\frac{4}{\delta}\right)} B_h \cdot \frac{1}{\sqrt{K}} \cdot \|\mathbf{v}_h^\pi\|_{\mathbf{G}_h^{-1}} + \frac{8}{3} \log\left(\frac{4}{\delta}\right) \cdot \frac{2(H - h + 1) + 1}{\eta_h + \sigma_r^2} \cdot \|\mathbf{v}_h^\pi\|_{\mathbf{G}_h^{-1}} \cdot \left\|\mathbf{G}_h^{-1}\right\|^{1/2} \cdot \frac{1}{K},$$

where $B_h$ is a $\widehat{V}_{h+1}^\pi$-dependent constant and $\mathbf{G}_h$ is a $\widehat{\sigma}_h$-dependent matrix given by

$$B_h = \max_{(s,a) \sim \nu_h} \frac{\mathbb{V}_h \widehat{V}_{h+1}^\pi(s, a) + \sigma_r^2}{\max\left\{ \eta_h, \mathbb{V}_h \widehat{V}_{h+1}^\pi(s, a) \right\} + \sigma_r^2 - \frac{C(H-h+1)^2 \sqrt{d}}{\sqrt{K}}}$$

$$\sim 1 + \widetilde{\mathcal{O}}(1/\sqrt{K}),$$

$$\mathbf{G}_h := \mathbb{E}_h \left[ \frac{\phi(s, a)\phi(s, a)^\top}{\widehat{\sigma}_h^2(s, a)} \middle| \widehat{\sigma}_h \right],$$

**Remark D.4.** Lemma D.3 will be combined with Lemma F.2, which gives an explicit formula for the constant $C$ with high probability, as will be shown in Lemma D.5.

*Proof of Lemma D.3.* By definition,

$$e_k = (\mathbf{v}_h^\pi)^\top \widehat{\mathbf{\Lambda}}_h^{-1} \frac{\phi(s_{k,h}, a_{k,h})}{\widehat{\sigma}_h(s_{k,h}, a_{k,h})^2} \left( [\mathbb{P}_h \widehat{V}_{h+1}^\pi](s_{k,h}, a_{k,h}) - \widehat{V}_{h+1}^\pi(s'_{k,h}) - \epsilon_{k,h} \right)$$

for all $k \in [K]$. From Algorithm 1, it is clear that the function $\widehat{V}_{h+1}^\pi(\cdot)$ depends on the dataset $\check{\mathcal{D}}_i, \mathcal{D}_i$ for $i \geq h + 1$, and the function $\widehat{\sigma}_h(\cdot, \cdot)$ depends on $\widehat{V}_{h+1}^\pi$ and the dataset $\check{\mathcal{D}}_h$, which are all independent of the dataset $\mathcal{D}_h$ under Assumption C.1. Therefore, conditioning on $\widehat{V}_{h+1}$ and $\widehat{\sigma}_h$ will not change the distribution of $\mathcal{D}_h$.

Define $F_h = \{(s_{k,h}, a_{k,h}), k \in [K]\}$, and for now we further condition on $F_h$ being fixed. Then $\widehat{\mathbf{\Lambda}}_h$ and $\widehat{\sigma}_{k,h} := \widehat{\sigma}_h(s_{k,h}, a_{k,h})$, $k \in [K]$ are both fixed. Define the filtration $\{\mathcal{F}_k\}_{k \in [K]}$ conditioned on $F_h$ as $\mathcal{F}_k = \sigma\{s'_{1,h}, \epsilon_{1,h}, \cdots, s'_{k-1,h}, \epsilon_{k-1,h} | F_h\}$ for $1 < k \leq K$, and $\mathcal{F}_1$ as the empty $\sigma$-field. Then $\mathbb{E}[e_k \mid \mathcal{F}_k] = 0$ implies that $\{e_k\}_{k \in [K]}$ is a martingale difference sequence. Since $\widehat{V}_{h+1}^\pi \in \mathcal{V}_{h+1}(L)$ and $\widehat{\sigma}_h \in \mathcal{T}_h(L_1, L_2)$, we have $\widehat{\sigma}_h(s, a)^2 \geq \eta_h + \sigma_r^2$ for all $(s, a)$ and $|\widehat{V}_{h+1}^\pi(s)| \leq H - h + 1$ for all $s$. Also by Assumption 2.1 we have $|\epsilon_{k,h}| \leq 1$ almost surely. This then implies

$$|e_k| \leq \underbrace{\frac{2(H - h + 1) + 1}{\eta_h + \sigma_r^2} \cdot \|\mathbf{v}_h^\pi\|_{\widehat{\mathbf{\Lambda}}_h^{-1}} \cdot \|\phi(s_{k,h}, a_{k,h})\|_{\widehat{\mathbf{\Lambda}}_h^{-1}}}_{c_{h,k}},$$

and

$$\text{Var}(e_k | F_h, \mathcal{F}_k) = \left[ (\mathbf{v}_h^\pi)^\top \widehat{\mathbf{\Lambda}}_h^{-1} \frac{\phi(s_{k,h}, a_{k,h})}{\widehat{\sigma}_h(s_{k,h}, a_{k,h})} \right]^2 \cdot \mathbb{E}\left[ \left( \frac{\mathbb{P}_h \widehat{V}_{h+1}^\pi(s_{k,h}, a_{k,h}) - \widehat{V}_{h+1}^\pi(s'_{k,h}) - \epsilon_{k,h}}{\widehat{\sigma}_h(s_{k,h}, a_{k,h})} \right)^2 \middle| F_h, \mathcal{F}_k \right]$$

$$\leq \left[ (\mathbf{v}_h^\pi)^\top \widehat{\mathbf{\Lambda}}_h^{-1} \frac{\phi(s_{k,h}, a_{k,h})\phi(s_{k,h}, a_{k,h})^\top}{\widehat{\sigma}_h(s_{k,h}, a_{k,h})^2} \widehat{\mathbf{\Lambda}}_h^{-1} \mathbf{v}_h^\pi \right]$$

$$\cdot \left( \frac{\mathbb{V}_h \widehat{V}_{h+1}^\pi(s_{k,h}, a_{k,h}) + \sigma_r^2}{\max\left\{ \eta_h, \mathbb{V}_h \widehat{V}_{h+1}^\pi(s_{k,h}, a_{k,h}) \right\} + \sigma_r^2 - \frac{C(H-h+1)^2 \sqrt{d}}{\sqrt{K}}} \right),$$

where the last step is from Assumption 2.1 that $\epsilon_{k,h}$ is independent random noise satisfying $\mathrm{Var}[\epsilon_{k,h} \mid s_{k,h}, a_{k,k}] \leq \sigma_r^2$, and (D.3). Denote $c_h = \max_{k \in [K]}\{c_{h,k}\}$. We then have

$$c_h \leq \frac{2(H-h+1)+1}{\eta_h + \sigma_r^2} \cdot \|\mathbf{v}_h^\pi\|_{\widehat{\boldsymbol{\Lambda}}_h^{-1}} \cdot \|\widehat{\boldsymbol{\Lambda}}_h^{-1}\|^{1/2}. \tag{D.4}$$

For simplicity, denote

$$b_{h,k} := \frac{\mathbb{V}_h \widehat{V}_{h+1}^\pi(s_{k,h}, a_{k,h}) + \sigma_r^2}{\max\left\{\eta_h, \mathbb{V}_h \widehat{V}_{h+1}^\pi(s_{k,h}, a_{k,h})\right\} + \sigma_r^2 - \frac{C(H-h+1)^2\sqrt{d}}{\sqrt{K}}}$$

$$\leq 1 + \frac{\frac{C(H-h+1)^2\sqrt{d}}{\sqrt{K}}}{\eta_h + \sigma_r^2 - \frac{C(H-h+1)^2\sqrt{d}}{\sqrt{K}}}$$

$$\sim 1 + \widetilde{\mathcal{O}}(1/\sqrt{K}),$$

and $b_h := \max_k\{b_{h,k}\}$. Therefore, we further have

$$\sum_{k=1}^K \mathrm{Var}(e_k | F_h, \mathcal{F}_k) \leq (\mathbf{v}_h^\pi)^\top \widehat{\boldsymbol{\Lambda}}_h^{-1} \left( \sum_{k=1}^K \frac{\phi(s_{k,h}, a_{k,h})\phi(s_{k,h}, a_{k,h})^\top}{\widehat{\sigma}_h(s_{k,h}, a_{k,h})^2} \right) \widehat{\boldsymbol{\Lambda}}_h^{-1} \mathbf{v}_h^\pi \cdot b_h$$

$$= (\mathbf{v}_h^\pi)^\top \widehat{\boldsymbol{\Lambda}}_h^{-1} \left( \widehat{\boldsymbol{\Lambda}}_h - \lambda \mathbf{I}_d \right) \widehat{\boldsymbol{\Lambda}}_h^{-1} \mathbf{v}_h^\pi \cdot b_h$$

$$\leq b_h \cdot \|\mathbf{v}_h^\pi\|_{\widehat{\boldsymbol{\Lambda}}_h^{-1}}^2,$$

since $(\widehat{\boldsymbol{\Lambda}}_h)^{-1/2}(\widehat{\boldsymbol{\Lambda}}_h - \lambda \mathbf{I}_d)(\widehat{\boldsymbol{\Lambda}}_h)^{-1/2}$ is a contraction. Then by Freedman's inequality H.2, we have

$$\mathbb{P}\left( \left| \sum_{k=1}^K e_k \right| \geq \epsilon \Big| F_h \right) \leq 2\exp\left( -\frac{\epsilon^2/2}{b_h \|\mathbf{v}_h^\pi\|_{\widehat{\boldsymbol{\Lambda}}_h^{-1}}^2 + c_h \epsilon/3} \right),$$

since $b_h$ and $c_h$ are fixed once we condition on $\widehat{V}_{h+1}^\pi$, $\widehat{\sigma}_h$ and $F_h$. It follows that with conditional (on $\widehat{V}_{h+1}^\pi, \widehat{\sigma}_h, F_h$) probability at least $1 - \delta$,

$$\left| \sum_{k=1}^K e_k \right| \leq \sqrt{2\log\left(\frac{2}{\delta}\right) b_h} \cdot \|\mathbf{v}_h^\pi\|_{\widehat{\boldsymbol{\Lambda}}_h^{-1}} + \frac{2}{3}\log\frac{2}{\delta} \cdot c_h. \tag{D.5}$$

Define the matrix $\mathbf{G}_h$ as the conditional expectation given as

$$\mathbf{G}_h := \mathbb{E}_h \left[ \frac{\phi(s,a)\phi(s,a)^\top}{\widehat{\sigma}_h(s,a)^2} \Big| \widehat{\sigma}_h \right], \tag{D.6}$$

by recalling the notation $\mathbb{E}_h[f(s,a)] = \int_{\mathcal{S} \times \mathcal{A}} f(s,a)\mathrm{d}\nu_h(s,a)$ for any function $f$ on $\mathcal{S} \times \mathcal{A}$, with $\nu_h(\cdot, \cdot)$ being the occupancy measure of the MDP for stage $h$ induced by the behavior policy $\bar{\pi}$. Now, since conditioning on $\widehat{V}_{h+1}^\pi$ and $\widehat{\sigma}_h$ does not change the distribution of $F_h$, by Lemma H.5, if $K$ satisfies

$$K \geq \max\left\{ 512(\eta_h + \sigma_r^2)^{-2}\|\mathbf{G}_h^{-1}\|^2 \log\left(\frac{2d}{\delta}\right), 4\lambda\|\mathbf{G}_h^{-1}\| \right\}, \tag{D.7}$$

then over the space of $F_h$, there exists an event $\mathcal{E}_h$ such that $\mathbb{P}(\mathcal{E}_h) \geq 1 - \delta$ and for all $F_h \in \mathcal{E}_h$ we have

$$\|\mathbf{u}\|_{\widehat{\boldsymbol{\Lambda}}_h^{-1}} \leq \frac{2}{\sqrt{K}} \cdot \|\mathbf{u}\|_{\mathbf{G}_h^{-1}} \tag{D.8}$$

for all $\mathbf{u} \in \mathbb{R}^d$. Combining (D.4), (D.5) and (D.8), we conclude that, with conditional probability (on $\widehat{V}_{h+1}^\pi, \widehat{\sigma}_h$ only) at least $1 - 2\delta$,

$$\left| \sum_{k=1}^K e_k \right| \leq \sqrt{2\log\left(\frac{2}{\delta}\right) B_h} \cdot \frac{2}{\sqrt{K}} \cdot \|\mathbf{v}_h^\pi\|_{\mathbf{G}_h^{-1}} + \frac{2}{3}\log\frac{2}{\delta} \cdot \frac{2(H-h+1)+1}{\eta_h + \sigma_r^2} \cdot \|\mathbf{v}_h^\pi\|_{\mathbf{G}_h^{-1}} \cdot \|\mathbf{G}_h^{-1}\|^{1/2} \cdot \frac{4}{K},$$

where $B_h$ is a $\widehat{V}_{h+1}^\pi$-dependent constants given by

$$B_h = \max_{(s,a)\sim\nu_h} \frac{\mathbb{V}_h\widehat{V}_{h+1}^\pi(s,a) + \sigma_r^2}{\max\left\{\eta_h, \mathbb{V}_h\widehat{V}_{h+1}^\pi(s,a)\right\} + \sigma_r^2 - \frac{C(H-h+1)^2\sqrt{d}}{\sqrt{K}}}$$

$$\sim 1 + \widetilde{\mathcal{O}}(1/\sqrt{K}),$$

and $\mathbf{G}_h$ is a $\widehat{\sigma}_h$-dependent matrix given by (D.6). Replacing $\delta$ with $\delta/2$ finishes the proof. $\qquad\square$

In the next lemma, we relax the conditioning on $\widehat{\sigma}_h$ and condition on $\widehat{V}_{h+1}^\pi$ only.

**Lemma D.5.** For any $h \in [H]$, condition on $\widehat{V}_{h+1}^\pi \in \mathcal{V}_{h+1}(L)$ being fixed and satisfying $\sup_s |\widehat{V}_{h+1}^\pi(s) - V_{h+1}^\pi(s)| \le \rho$ for some $\rho \ge 0$, if $K$ satisfies (D.14) and

$$K \ge \max\left\{ \frac{911}{(\eta_h + \sigma_r^2)^2\iota_h^2} \cdot \log\left(\frac{4d}{\delta}\right) \,,\, \frac{6\lambda}{\iota_h} \right\}, \tag{D.9}$$

then with conditional probability at least $1 - \delta$, we have

$$\begin{aligned}
|E_{2,h}| \le\, & 2\sqrt{2\log\left(\frac{8}{\delta}\right)}\, B_h \cdot \|\mathbf{v}_h^\pi\|_{\mathbf{\Lambda}_h^{-1}} \cdot \frac{1}{\sqrt{K}} \\
& + 2\sqrt{2\log\left(\frac{8}{\delta}\right)}\, B_h \cdot \sqrt{C_0 C_1 \cdot \frac{1}{\iota_h}} \cdot \|\mathbf{v}_h^\pi\|_{\mathbf{\Lambda}_h^{-1}} \cdot (K^{1/4}\sqrt{\widetilde{\rho}}) \cdot \frac{1}{K^{3/4}} \\
& + \frac{8}{3}\log\left(\frac{8}{\delta}\right) \cdot \frac{2(H-h+1)+1}{\eta_h + \sigma_r^2}\sqrt{C_1} \cdot \frac{1}{\sqrt{\iota_h}} \cdot \|\mathbf{v}_h^\pi\|_{\mathbf{\Lambda}_h^{-1}} \cdot \frac{1}{K} \\
& + \frac{8}{3}\log\left(\frac{8}{\delta}\right) \cdot \frac{2(H-h+1)+1}{\eta_h + \sigma_r^2} \cdot \sqrt{C_0} \cdot C_1 \cdot \frac{1}{\iota_h} \cdot \|\mathbf{v}_h^\pi\|_{\mathbf{\Lambda}_h^{-1}} \cdot \sqrt{\widetilde{\rho}} \cdot \frac{1}{K}
\end{aligned}$$

where

$$C_0 = \left(\frac{\|\mathbf{\Lambda}_h\|}{\iota_h}\right)^{1/2},$$

$$C_1 = \frac{1}{1 - \widetilde{\rho}/\iota_h},$$

$$\widetilde{\rho} = \frac{1}{(\eta_h + \sigma_r^2)^2} \cdot \left(\frac{C_{K,h,\delta}(H-h+1)^2\sqrt{d}}{\sqrt{K}} + 4(H-h+1)\cdot\rho\right),$$

$$C_{K,h,\delta} = 12\sqrt{2} \cdot \frac{1}{\sqrt{\kappa_h}} \cdot \left[\frac{1}{2}\log\left(\frac{\lambda + K}{\lambda}\right) + \frac{1}{d}\log\frac{8}{\delta}\right]^{1/2} + 12\lambda \cdot \frac{1}{\kappa_h}.$$

and $B_h$ is a $\widehat{V}_{h+1}^\pi$-dependent constant given by

$$B_h = \max_{(s,a)\sim\nu_h} \frac{\mathbb{V}_h\widehat{V}_{h+1}^\pi(s,a) + \sigma_r^2}{\max\left\{\eta_h, \mathbb{V}_h\widehat{V}_{h+1}^\pi(s,a)\right\} + \sigma_r^2 - \frac{C_{K,h,\delta}(H-h+1)^2\sqrt{d}}{\sqrt{K}}}.$$

**Remark D.6.** Conditioning on the event in Theorem C.2, we have $\rho \sim \widetilde{\mathcal{O}}(1/\sqrt{K})$ and thus $\widetilde{\rho} \sim \mathcal{O}(1/\sqrt{K})$, which means the term $\sqrt{K}\widetilde{\rho}$ is a constant up to a logarithmic factor. This indicates that in the upper bound of $|E_{2,h}|$, only the first term is of order $\widetilde{\mathcal{O}}(1/\sqrt{K})$.

*Proof of Lemma D.5.* For simplicity, denote the function $\sigma_V(\cdot,\cdot)$ as

$$\sigma_V(\cdot,\cdot) := \sqrt{\max\left\{\eta_h, \mathbb{V}_h\widehat{V}_{h+1}^\pi(\cdot,\cdot)\right\} + \sigma_r^2},$$

and recall that $\widehat{\sigma}_h(\cdot, \cdot)$ is an estimator for $\sigma_V(\cdot, \cdot)$ generated using the dataset $\check{\mathcal{D}}_h$. Also recall the definition

$$\sigma_h(\cdot, \cdot) := \sqrt{\max\{\eta_h, \, \mathbb{V}_h V_{h+1}^\pi(\cdot, \cdot)\} + \sigma_r^2}.$$

First of all, by Lemma F.2, with probability at least $1 - \delta$ over the space of $\check{\mathcal{D}}_h$, the following event happens:

$$\sup_{s,a} |\widehat{\sigma}_h^2(s, a) - \sigma_V^2(s, a)| \leq \frac{C_{K,h,\delta}(H - h + 1)^2 \sqrt{d}}{\sqrt{K}}, \tag{D.10}$$

where $C_{K,h,\delta}$ is given by

$$C_{K,h,\delta} = 12\sqrt{2} \cdot \frac{1}{\sqrt{\kappa_h}} \cdot \left[\frac{1}{2}\log\left(\frac{\lambda + K}{\lambda}\right) + \frac{1}{d}\log\frac{4}{\delta}\right]^{1/2} + 12\lambda \cdot \frac{1}{\kappa_h}.$$

Denote the above event of $\widehat{\sigma}_h$ by $\mathcal{E}_{\widehat{\sigma}}$. For each fixed $\widehat{\sigma}_h \in \mathcal{E}_{\widehat{\sigma}}$, we can then apply Lemma D.3 with $C$ replace by $C_{K,h,\delta}$. This gives that, for any $h$, condition on $\widehat{V}_{h+1}^\pi$ and $\widehat{\sigma}_h$, with probability at least $1 - \delta$,

$$|E_{2,h}|$$
$$\leq 2\sqrt{2\log\left(\frac{4}{\delta}\right) B_h} \cdot \frac{1}{\sqrt{K}} \cdot \|\mathbf{v}_h^\pi\|_{\mathbf{G}_h^{-1}} + \frac{8}{3}\log\left(\frac{4}{\delta}\right) \cdot \frac{2(H - h + 1) + 1}{\eta_h + \sigma_r^2} \cdot \|\mathbf{v}_h^\pi\|_{\mathbf{G}_h^{-1}} \cdot \|\mathbf{G}_h^{-1}\|^{1/2} \cdot \frac{1}{K}, \tag{D.11}$$

where

$$B_h := \max_{(s,a) \sim \nu_h} \frac{\mathbb{V}_h \widehat{V}_{h+1}^\pi(s, a) + \sigma_r^2}{\max\{\eta_h, \mathbb{V}_h \widehat{V}_{h+1}^\pi(s, a)\} + \sigma_r^2 - \frac{C_{K,h,\delta}(H-h+1)^2\sqrt{d}}{\sqrt{K}}} \sim 1 + \widetilde{\mathcal{O}}(1/\sqrt{K}),$$

$$\mathbf{G}_h := \mathbb{E}_h\left[\frac{\phi(s, a)\phi(s, a)^\top}{\widehat{\sigma}_h^2(s, a)}\bigg|\widehat{\sigma}_h\right].$$

However, note that in the upper bound of $|E_{2,h}|$, the $\|\mathbf{v}_h^\pi\|_{\mathbf{G}_h^{-1}}$ and $\|\mathbf{G}_h^{-1}\|^{1/2}$ term are $\widehat{\sigma}_h$-dependent, and so is the lower bound of the sample complexity given by (D.7). Therefore, it remains to derive a uniform upper bound of $E_{2,h}$ for all $\widehat{\sigma}_h \in \mathcal{E}_{\widehat{\sigma}}$, and a uniform lower bound of $K$.

To get this, first note that since $\widehat{V}_{h+1}^\pi, V_{h+1}^\pi \in \mathcal{V}_{h+1}(L)$ and $\sup_s |\widehat{V}_{h+1}^\pi(s) - V_{h+1}^\pi(s)| \leq \rho$, we have

$$\sup_{s,a} |\sigma_V^2(s, a) - \sigma_h^2(s, a)| \leq 4(H - h + 1)\rho.$$

Using triangular inequality and (D.10) gives

$$\sup_{s,a} |\widehat{\sigma}_h^2(s, a) - \sigma_h^2(s, a)| \leq \frac{C_{K,h,\delta}(H - h + 1)^2\sqrt{d}}{\sqrt{K}} + 4(H - h + 1) \cdot \rho,$$

for all $\widehat{\sigma}_h \in \mathcal{E}_{\widehat{\sigma}}$.

Note that by definition,

$$\|\mathbf{G}_h - \mathbf{\Lambda}_h\| = \left\|\mathbb{E}_h\left[\frac{\phi(s, a)\phi(s, a)^\top}{\widehat{\sigma}_h^2(s, a)}\right] - \mathbb{E}_h\left[\frac{\phi(s, a)\phi(s, a)^\top}{\sigma_h^2(s, a)}\right]\right\|$$

$$= \left\|\mathbb{E}_h\left[\phi(s, a)\phi(s, a)^\top \frac{\widehat{\sigma}_h^2(s, a) - \sigma_h^2(s, a)}{\widehat{\sigma}_h^2(s, a) \cdot \sigma_h^2(s, a)}\right]\right\|$$

$$\leq \frac{1}{(\eta_h + \sigma_r^2)^2} \cdot \left(\frac{C_{K,h,\delta}(H - h + 1)^2\sqrt{d}}{\sqrt{K}} + 4(H - h + 1) \cdot \rho\right)$$

$$:= \widetilde{\rho}$$

$$\sim \widetilde{\mathcal{O}}(1/\sqrt{K}),$$

where the inequality is from $\|\phi(\cdot,\cdot)\| \le 1$ and $|\sigma(\cdot,\cdot)| \ge \sqrt{\eta_h + \sigma_r^2}$ for all $\sigma(\cdot,\cdot) \in \mathcal{T}_h$. Combine the above inequality with Lemma H.3, and we have

$$\left\|\mathbf{G}_h^{-1}\right\| \le \frac{\left\|\boldsymbol{\Lambda}_h^{-1}\right\|}{1 - \left\|\boldsymbol{\Lambda}_h^{-1}\right\| \cdot \left\|\boldsymbol{\Lambda}_h - \mathbf{G}_h\right\|} \le \frac{\left\|\boldsymbol{\Lambda}_h^{-1}\right\|}{1 - \left\|\boldsymbol{\Lambda}_h^{-1}\right\| \cdot \widetilde{\rho}}, \tag{D.12}$$

and by Lemma H.3 again,

$$\begin{aligned}
\|\mathbf{v}_h^\pi\|_{\mathbf{G}_h^{-1}} &\le \left[1 + \sqrt{\left(\left\|\boldsymbol{\Lambda}_h^{-1}\right\| \cdot \left\|\boldsymbol{\Lambda}_h\right\|\right)^{1/2} \cdot \left\|\mathbf{G}_h^{-1}\right\| \cdot \widetilde{\rho}}\right] \cdot \|\mathbf{v}_h^\pi\|_{\boldsymbol{\Lambda}_h^{-1}} \\
&\le \left[1 + \sqrt{\left(\left\|\boldsymbol{\Lambda}_h^{-1}\right\| \cdot \left\|\boldsymbol{\Lambda}_h\right\|\right)^{1/2} \cdot \frac{\left\|\boldsymbol{\Lambda}_h^{-1}\right\|}{1 - \left\|\boldsymbol{\Lambda}_h^{-1}\right\| \cdot \widetilde{\rho}} \cdot \widetilde{\rho}}\right] \cdot \|\mathbf{v}_h^\pi\|_{\boldsymbol{\Lambda}_h^{-1}} \\
&= \left[1 + \sqrt{\left(\left\|\boldsymbol{\Lambda}_h^{-1}\right\|\right)^{3/2} \cdot \left(\left\|\boldsymbol{\Lambda}_h\right\|\right)^{1/2} \cdot \frac{1}{1 - \left\|\boldsymbol{\Lambda}_h^{-1}\right\| \cdot \widetilde{\rho}} \cdot \widetilde{\rho}}\right] \cdot \|\mathbf{v}_h^\pi\|_{\boldsymbol{\Lambda}_h^{-1}} \\
&= \|\mathbf{v}_h^\pi\|_{\boldsymbol{\Lambda}_h^{-1}} + \sqrt{\left(\left\|\boldsymbol{\Lambda}_h^{-1}\right\|\right)^{3/2} \cdot \left(\left\|\boldsymbol{\Lambda}_h\right\|\right)^{1/2} \cdot \frac{1}{1 - \left\|\boldsymbol{\Lambda}_h^{-1}\right\| \cdot \widetilde{\rho}}} \cdot \sqrt{\widetilde{\rho}} \cdot \|\mathbf{v}_h^\pi\|_{\boldsymbol{\Lambda}_h^{-1}}. \tag{D.13}
\end{aligned}$$

Note that the above holds when $K$ is sufficiently large such that $\left\|\boldsymbol{\Lambda}_h^{-1}\right\| \cdot \widetilde{\rho}$ is less than, for example,

$$\left\|\boldsymbol{\Lambda}_h^{-1}\right\| \cdot \widetilde{\rho} \le 1/4. \tag{D.14}$$

We are now ready to derive an upper bound independent of $\widehat{\sigma}_h$. First define

$$C_0 = \left(\left\|\boldsymbol{\Lambda}_h^{-1}\right\|\right)^{1/2} \cdot \left(\left\|\boldsymbol{\Lambda}_h\right\|\right)^{1/2}, \quad C_1 = \frac{1}{1 - \left\|\boldsymbol{\Lambda}_h^{-1}\right\| \cdot \widetilde{\rho}}.$$

Then we have $\|\mathbf{v}_h^\pi\|_{\mathbf{G}_h^{-1}} \le \|\mathbf{v}_h^\pi\|_{\boldsymbol{\Lambda}_h^{-1}} + \sqrt{C_0 C_1 \left\|\boldsymbol{\Lambda}_h^{-1}\right\| \cdot \widetilde{\rho}} \cdot \|\mathbf{v}_h^\pi\|_{\boldsymbol{\Lambda}_h^{-1}}$, and $\left\|\mathbf{G}_h^{-1}\right\|^{1/2} \le \sqrt{C_1} \cdot \left\|\boldsymbol{\Lambda}_h^{-1}\right\|^{1/2}$. It follows that

$$\|\mathbf{v}_h^\pi\|_{\mathbf{G}_h^{-1}} \cdot \left\|\mathbf{G}_h^{-1}\right\|^{1/2} \le \sqrt{C_1} \cdot \left\|\boldsymbol{\Lambda}_h^{-1}\right\|^{1/2} \cdot \|\mathbf{v}_h^\pi\|_{\boldsymbol{\Lambda}_h^{-1}} + C_1 \cdot \left\|\boldsymbol{\Lambda}_h^{-1}\right\| \cdot \sqrt{C_0} \cdot \|\mathbf{v}_h^\pi\|_{\boldsymbol{\Lambda}_h^{-1}} \cdot \sqrt{\widetilde{\rho}}.$$

Plug into (D.11), and we have that, condition on $\widehat{V}_{h+1}^\pi$, with probability at least $1 - 2\delta$,

$$\begin{aligned}
|E_{2,h}| \le\ & 2\sqrt{2\log\left(\frac{4}{\delta}\right) B_h} \cdot \|\mathbf{v}_h^\pi\|_{\boldsymbol{\Lambda}_h^{-1}} \cdot \frac{1}{\sqrt{K}} \\
& + 2\sqrt{2\log\left(\frac{4}{\delta}\right) B_h} \cdot \sqrt{C_0 C_1 \left\|\boldsymbol{\Lambda}_h^{-1}\right\|} \cdot \|\mathbf{v}_h^\pi\|_{\boldsymbol{\Lambda}_h^{-1}} \cdot \sqrt{\widetilde{\rho}} \cdot \frac{1}{\sqrt{K}} \\
& + \frac{8}{3}\log\left(\frac{4}{\delta}\right) \cdot \frac{2(H - h + 1) + 1}{\eta_h + \sigma_r^2} \cdot \sqrt{C_1} \cdot \left\|\boldsymbol{\Lambda}_h^{-1}\right\|^{1/2} \cdot \|\mathbf{v}_h^\pi\|_{\boldsymbol{\Lambda}_h^{-1}} \cdot \frac{1}{K} \\
& + \frac{8}{3}\log\left(\frac{4}{\delta}\right) \cdot \frac{2(H - h + 1) + 1}{\eta_h + \sigma_r^2} \cdot C_1 \cdot \left\|\boldsymbol{\Lambda}_h^{-1}\right\| \cdot \sqrt{C_0} \cdot \|\mathbf{v}_h^\pi\|_{\boldsymbol{\Lambda}_h^{-1}} \cdot \sqrt{\widetilde{\rho}} \cdot \frac{1}{K}
\end{aligned}$$

Replacing $\delta$ by $\delta/2$ and using $1/\iota_h = \left\|\boldsymbol{\Lambda}_h^{-1}\right\|$ gives the desired upper bound. It remains to show the lower bound. By (D.7), (D.12), and (D.14), a uniform version of D.7 is given by

$$\begin{aligned}
K &\ge \max\left\{512(\eta_h + \sigma_r^2)^{-2} \cdot \frac{16}{9}\|\boldsymbol{\Lambda}_h^{-1}\|^2 \log\left(\frac{4d}{\delta}\right), 4\lambda \cdot \frac{4}{3}\|\boldsymbol{\Lambda}_h^{-1}\|\right\} \\
&> \max\left\{\frac{911}{(\eta_h + \sigma_r^2)^2 \iota_h^2} \cdot \log\left(\frac{4d}{\delta}\right), \frac{6\lambda}{\iota_h}\right\}. \tag{D.15}
\end{aligned}$$

$\square$

**Lemma D.7.** If $K$ satisfies (D.17), (D.20), (D.21) and

$$K \ge \max_{h \in [H]} \max\left\{\frac{911}{(\eta_h + \sigma_r^2)^2 \iota_h^2} \log\left(\frac{8Hd}{\delta}\right), \frac{6\lambda}{\iota_h}\right\}, \tag{D.16}$$

then with probability at least $1 - \delta$, we have

$$
|E_2| \leq 2\sqrt{2\log\left(\frac{16H}{\delta}\right)} B \cdot \left[\sum_{h=1}^{H} \|\mathbf{v}_h^\pi\|_{\mathbf{\Lambda}_h^{-1}}\right] \cdot \frac{1}{\sqrt{K}}
$$

$$
+ \frac{16\sqrt{2}}{3}\log\left(\frac{16H}{\delta}\right) \cdot \sqrt{B} \cdot \left[\sum_{h=1}^{H} A_1(h)\right] \cdot \frac{1}{K^{3/4}}
$$

$$
+ \frac{16\sqrt{2}}{3}\log\left(\frac{16H}{\delta}\right) \cdot \sqrt{B} \cdot \left[\sum_{h=1}^{H} (A_2(h) + A_3(h))\right] \cdot \frac{1}{K},
$$

where

$$
A_1(h) = \sqrt{C_0(h) \cdot \frac{1}{\iota_h}} \cdot \left(K^{1/4}\sqrt{\widetilde{\rho}(h)}\right) \cdot \|\mathbf{v}_h^\pi\|_{\mathbf{\Lambda}_h^{-1}},
$$

$$
A_2(h) = \frac{2(H-h+1)+1}{\eta_h + \sigma_r^2} \cdot \frac{1}{\sqrt{\iota_h}} \cdot \|\mathbf{v}_h^\pi\|_{\mathbf{\Lambda}_h^{-1}},
$$

$$
A_3(h) = \frac{2(H-h+1)+1}{\eta_h + \sigma_r^2} \cdot \sqrt{C_0(h)} \cdot \sqrt{\widetilde{\rho}} \cdot \frac{1}{\iota_h} \cdot \|\mathbf{v}_h^\pi\|_{\mathbf{\Lambda}_h^{-1}},
$$

$$
C_0(h) = \left(\frac{\|\mathbf{\Lambda}_h\|}{\iota_h}\right)^{1/2},
$$

$$
\widetilde{\rho}(h) = \frac{1}{(\eta_h + \sigma_r^2)^2} \cdot \left(\frac{C_{K,h,\delta}(H-h+1)^2\sqrt{d}}{\sqrt{K}} + 4(H-h+1) \cdot \widetilde{C}(h) \cdot \frac{d}{\sqrt{K}}\right),
$$

$$
C_{K,h,\delta} = 12\sqrt{2} \cdot \frac{1}{\sqrt{\kappa_h}} \cdot \left[\frac{1}{2}\log\left(\frac{\lambda + K}{\lambda}\right) + \frac{1}{d}\log\frac{16H}{\delta}\right]^{1/2} + 12\lambda \cdot \frac{1}{\kappa_h},
$$

$$
\widetilde{C}(h) = C \cdot C_{h,2} \cdot \log\left(\frac{dH^2K}{\kappa\delta}\right),
$$

and $C$ is some universal constant, $B$ is a problem-dependent constant given by

$$
B = \max_{h\in[H]} \max_{V\in\mathcal{V}_{h+1}(L)} \max_{(s,a)\sim\nu_h} \frac{\mathbb{V}_h V(s,a) + \sigma_r^2}{\max\{\eta_h, \mathbb{V}_h V(s,a)\} + \sigma_r^2 - \frac{C_{K,h,\delta}(H-h+1)^2\sqrt{d}}{\sqrt{K}}},
$$

and $C_{h,2}$ are the same constants as in Theorem C.2.

*Proof of Lemma D.7.* First, by Theorem C.2, if $K$ satisfies

$$
K \geq C \cdot \frac{H^2 d^2}{\kappa^2}\log\left(\frac{dHK}{\kappa\delta}\right) \cdot \max_{h\in[H]} \frac{(H-h+1)^2}{(\eta_h + \sigma_r^2)^2} \cdot \max_{h\in[H]} \frac{(H-h+1)^2}{\iota_h(\eta_h + \sigma_r^2)}, \tag{D.17}
$$

for some problem-independent constant $C$, then with probability at least $1 - \delta/2$, for all $h \in [H]$, we have

$$
\sup_s \left|\widehat{V}_{h+1}^\pi(s) - V_{h+1}^\pi(s)\right| \leq \widetilde{C} \cdot \frac{d}{\sqrt{K}},
$$

where

$$
\widetilde{C} := C \cdot C_{h,2} \cdot \log\left(\frac{dH^2K}{\kappa\delta}\right) + C \cdot C_{h,1} \cdot \frac{H}{\sqrt{dK}}, \tag{D.18}
$$

and

$$
C_{h,1} = \sum_{i=h}^{H} \frac{1}{\iota_h}, \qquad C_{h,2} = \sum_{i=h}^{H} \frac{H-h+1}{\sqrt{\iota_h(\eta_h + \sigma_r^2)}}.
$$

For simplicity, we define

$$\widetilde{C}(h) = C \cdot C_{h,2} \cdot \log\left(\frac{dH^2 K}{\kappa \delta}\right),$$

for some different constant $C$, since the first term on the RHS of (D.18) is much larger than the second one by using $\eta_h \leq (H - h + 1)^2$.

Now we can combine Theorem C.2 and Lemma D.5 with the parameter $\rho$ replaced by $\widetilde{C} \cdot \frac{d}{\sqrt{K}}$, take a union bound over all $H$ terms, and conclude that, with probability at least $1 - \delta$, the result of Lemma D.5 holds for all $h \in [H]$ :

$$|E_{2,h}| \leq 2\sqrt{2\log\left(\frac{16H}{\delta}\right) B_h} \cdot \|\mathbf{v}_h^\pi\|_{\mathbf{\Lambda}_h^{-1}} \cdot \frac{1}{\sqrt{K}}$$

$$+ 2\sqrt{2\log\left(\frac{16H}{\delta}\right) B_h} \cdot \sqrt{C_0 C_1 \cdot \frac{1}{\iota_h}} \cdot \|\mathbf{v}_h^\pi\|_{\mathbf{\Lambda}_h^{-1}} \cdot (K^{1/4}\sqrt{\widetilde{\rho}(h)}) \cdot \frac{1}{K^{3/4}}$$

$$+ \frac{8}{3}\log\left(\frac{16H}{\delta}\right) \cdot \frac{2(H - h + 1) + 1}{\eta_h + \sigma_r^2} \cdot \sqrt{C_1} \cdot \frac{1}{\sqrt{\iota_h}} \cdot \|\mathbf{v}_h^\pi\|_{\mathbf{\Lambda}_h^{-1}} \cdot \frac{1}{K}$$

$$+ \frac{8}{3}\log\left(\frac{16H}{\delta}\right) \cdot \frac{2(H - h + 1) + 1}{\eta_h + \sigma_r^2} \cdot \sqrt{C_0} \cdot C_1 \cdot \frac{1}{\iota_h} \cdot \|\mathbf{v}_h^\pi\|_{\mathbf{\Lambda}_h^{-1}} \cdot \sqrt{\widetilde{\rho}(h)} \cdot \frac{1}{K} \quad \text{(D.19)}$$

where for each $h \in [H]$, $\widetilde{\rho}(h)$ is given as

$$\widetilde{\rho}(h) = \frac{1}{(\eta_h + \sigma_r^2)^2} \cdot \left(\frac{C_{K,h,\delta}(H - h + 1)^2\sqrt{d}}{\sqrt{K}} + 4(H - h + 1) \cdot \widetilde{C}(h) \cdot \frac{d}{\sqrt{K}}\right),$$

$$C_{K,h,\delta} = 12\sqrt{2} \cdot \frac{1}{\sqrt{\kappa_h}} \cdot \left[\frac{1}{2}\log\left(\frac{\lambda + K}{\lambda}\right) + \frac{1}{d}\log\frac{16H}{\delta}\right]^{1/2} + 12\lambda \cdot \frac{1}{\kappa_h}.$$

Now, by the expression of $B_h$, if $K$ satisfies

$$K \geq \frac{4C_{K,h,\delta}^2 H^4 d}{\sigma_r^4} \geq 1152 \cdot \max_{h \in [H]} \frac{1}{\kappa_h^2} \cdot \left[\frac{1}{2}\log\left(\frac{\lambda + K}{\lambda}\right) + \frac{1}{d}\log\frac{16H}{\delta}\right] \cdot \frac{H^4 d}{\sigma_r^4}, \quad \text{(D.20)}$$

then we have $B_h \leq 2$ for all $h \in [H]$. Also, by (D.14), $K$ also needs to be large enough so that

$$\max_{h \in [H]} \left\{\|\mathbf{\Lambda}_h^{-1}\| \cdot \widetilde{\rho}(h)\right\} = \max_{h \in [H]} \{\widetilde{\rho}(h)/\iota_h\} \leq 1/4, \quad \text{(D.21)}$$

which implies $C_1 \leq 4/3$ for all $h$. We can then simplify (D.19) into

$$|E_{2,h}|$$

$$\leq 2\sqrt{2\log\left(\frac{16H}{\delta}\right) B_h} \cdot \|\mathbf{v}_h^\pi\|_{\mathbf{\Lambda}_h^{-1}} \cdot \frac{1}{\sqrt{K}}$$

$$+ \frac{16\sqrt{2}}{3}\log\left(\frac{16H}{\delta}\right) \cdot \sqrt{B_h} \cdot A_1(h) \cdot \frac{1}{K^{3/4}}$$

$$+ \frac{16\sqrt{2}}{3}\log\left(\frac{16H}{\delta}\right) \cdot \sqrt{B_h} \cdot [A_2(h) + A_3(h)] \cdot \frac{1}{K},$$

where

$$A_1(h) = \sqrt{C_0(h) \cdot \frac{1}{\iota_h}} \cdot \left(K^{1/4}\sqrt{\widetilde{\rho}(h)}\right) \cdot \|\mathbf{v}_h^\pi\|_{\mathbf{\Lambda}_h^{-1}},$$

$$A_2(h) = \frac{2(H - h + 1) + 1}{\eta_h + \sigma_r^2} \cdot \frac{1}{\sqrt{\iota_h}} \cdot \|\mathbf{v}_h^\pi\|_{\mathbf{\Lambda}_h^{-1}},$$

$$A_3(h) = \frac{2(H - h + 1) + 1}{\eta_h + \sigma_r^2} \cdot \sqrt{C_0(h)} \cdot \sqrt{\widetilde{\rho}} \cdot \frac{1}{\iota_h} \cdot \|\mathbf{v}_h^\pi\|_{\mathbf{\Lambda}_h^{-1}}.$$

By denoting

$$B := \max_{h \in [H]} \max_{V \in \mathcal{V}_{h+1}(L)} \max_{(s,a) \sim \nu_h} \frac{\mathbb{V}_h V(s,a) + \sigma_r^2}{\max\left\{\eta_h, \mathbb{V}_h V(s,a)\right\} + \sigma_r^2 - \frac{C_{K,h,\delta}(H-h+1)^2\sqrt{d}}{\sqrt{K}}},$$

which is less than 2 by (D.20),and using $|E_2| \le \sum_{h=1}^{K} |E_{2,h}|$, we prove the upper bound. The lower bound of $K$ comes from (D.17), (D.20), (D.21) and (D.15) .

$\square$

## D.2 Bounding the $E_1$ Term in the OPE Decomposition

Consider the term $E_1$ in (D.2):

$$|E_1| \le \lambda \sum_{h=1}^{H} \left| (\mathbf{v}_h^\pi)^\top \widehat{\mathbf{\Lambda}}_h^{-1} \int_{\mathcal{S}} \left( V_{h+1}^\pi(s) - \widehat{V}_{h+1}^\pi(s) \right) \boldsymbol{\mu}_h(s) \mathrm{d}s \right| := \sum_{h=1}^{H} |E_{1,h}|,$$

where for each $h \in [H]$,

$$E_{1,h} := \lambda(\mathbf{v}_h^\pi)^\top \widehat{\mathbf{\Lambda}}_h^{-1} \int_{\mathcal{S}} \left( V_{h+1}^\pi(s) - \widehat{V}_{h+1}^\pi(s) \right) \boldsymbol{\mu}_h(s) \mathrm{d}s.$$

**Lemma D.8.** Under the same event where the result of Lemma D.7 holds, if $K$ satisfies (D.16), (D.17), (D.20) and (D.21), we have

$$|E_1| \le 4\sqrt{2}\lambda \left[ \sum_{h=1}^{H} A_4(h) \right] \cdot \frac{H\sqrt{d}}{K},$$

where for each $h$,

$$A_4(h) = \frac{H - h + 1}{H} \cdot \frac{1}{\sqrt{\iota_h}} \cdot \|\mathbf{v}_h^\pi\|_{\mathbf{\Lambda}_h^{-1}} \cdot \left[ 1 + \sqrt{2C_0(h) \cdot \frac{1}{\iota_h} \cdot \widetilde{\rho}(h)} \right],$$

and $C_0(h)$ and $\widetilde{\rho}(h)$ are same constants as in Lemma D.7.

*Proof of Lemma D.8.* By (D.8) and (D.13) , we have that

$$\|\mathbf{u}\|_{\widehat{\mathbf{\Lambda}}_h^{-1}} \le \frac{2}{\sqrt{K}} \cdot \left\{ \|\mathbf{u}\|_{\mathbf{\Lambda}_h^{-1}} + \sqrt{\left(\|\mathbf{\Lambda}_h^{-1}\|\right)^{3/2} \cdot \left(\|\mathbf{\Lambda}_h\|\right)^{1/2} \cdot \frac{1}{1 - \|\mathbf{\Lambda}_h^{-1}\| \cdot \widetilde{\rho}(h)} \cdot \sqrt{\widetilde{\rho}(h)} \cdot \|\mathbf{u}\|_{\mathbf{\Lambda}_h^{-1}}} \right\}$$

$$= \frac{2}{\sqrt{K}} \cdot \left\{ \|\mathbf{u}\|_{\mathbf{\Lambda}_h^{-1}} + \sqrt{C_0(h) \cdot \frac{1}{\iota_h} \cdot \frac{1}{1 - \|\mathbf{\Lambda}_h^{-1}\| \cdot \widetilde{\rho}(h)} \cdot \sqrt{\widetilde{\rho}(h)} \cdot \|\mathbf{u}\|_{\mathbf{\Lambda}_h^{-1}}} \right\}, \qquad \text{(D.22)}$$

for all $\mathbf{u} \in \mathbb{R}^d$, where the constants take the same values as given in Lemma D.7, i.e.,

$$\widetilde{\rho}(h) = \frac{1}{(\eta_h + \sigma_r^2)^2} \cdot \left( \frac{C_{K,h,\delta}(H - h + 1)^2\sqrt{d}}{\sqrt{K}} + 4(H - h + 1) \cdot \widetilde{C}(h) \cdot \frac{d}{\sqrt{K}} \right),$$

$$C_{K,h,\delta} = 12\sqrt{2} \cdot \frac{1}{\sqrt{\kappa_h}} \cdot \left[ \frac{1}{2} \log\left( \frac{\lambda + K}{\lambda} \right) + \frac{1}{d} \log \frac{16H}{\delta} \right]^{1/2} + 12\lambda \cdot \frac{1}{\kappa_h},$$

$$\widetilde{C}(h) = C \cdot C_{h,2} \cdot \log\left( \frac{dH^2 K}{\kappa\delta} \right),$$

with $C_{h,2}$ being the same constant as in Theorem C.2. Also, since the result of Lemma G.4 holds, we have

$$\left\| \frac{\widehat{\mathbf{\Lambda}}_h}{K} - \mathbf{\Lambda}_h \right\| \le \frac{4\sqrt{2}}{(\eta_h + \sigma_r^2)\sqrt{K}} \cdot \left( \log \frac{16Hd}{\delta} \right)^{1/2} + \frac{\lambda}{K} + \widetilde{\rho}(h) \le 2\widetilde{\rho}(h),$$

where the first step is by replacing $\delta$ with $\delta/4H$ and the second step is by the choice of $K$ in Lemma D.7. Then It follows from Lemma H.3 that

$$\left\|\widehat{\mathbf{\Lambda}}_h^{-1}\right\| \leq \frac{\left\|(K\mathbf{\Lambda}_h)^{-1}\right\|}{1 - \|(K\mathbf{\Lambda}_h)^{-1}\| \cdot \left\|\widehat{\mathbf{\Lambda}}_h - K\mathbf{\Lambda}_h\right\|} \leq \frac{1}{K} \cdot \frac{\left\|\mathbf{\Lambda}_h^{-1}\right\|}{1 - 2\widetilde{\rho}(h) \cdot \left\|\mathbf{\Lambda}_h^{-1}\right\|} \leq \frac{2\left\|\mathbf{\Lambda}_h^{-1}\right\|}{K}, \quad \text{(D.23)}$$

since $2\widetilde{\rho}(h) \cdot \left\|\mathbf{\Lambda}_h^{-1}\right\| \leq 1/2$ by (D.21). Also, since on the event of Lemma D.7, we have $|V_{h+1}^\pi(s) - \widehat{V}_{h+1}^\pi(s)| \leq 2(H - h + 1)$, Assumption 2.1 then implies

$$\left\|\int_{\mathcal{S}} \left(V_{h+1}^\pi(s) - \widehat{V}_{h+1}^\pi(s)\right) \boldsymbol{\mu}_h(s)\mathrm{d}s\right\|_2 \leq 2(H - h + 1)\sqrt{d}.$$

Together with (D.22) and (D.23) and Cauchy-Schwartz inequality, we conclude that

$$|E_{1,h}| = \left|\lambda(\mathbf{v}_h^\pi)^\top \widehat{\mathbf{\Lambda}}_h^{-1} \int_{\mathcal{S}} \left(V_{h+1}^\pi(s) - \widehat{V}_{h+1}^\pi(s)\right) \boldsymbol{\mu}_h(s)\mathrm{d}s\right|$$

$$\leq \lambda \cdot \frac{4\sqrt{2}(H - h + 1)\sqrt{d}\left\|\mathbf{\Lambda}_h^{-1}\right\|^{1/2}}{K} \cdot \|\mathbf{v}_h^\pi\|_{\mathbf{\Lambda}_h^{-1}} \cdot \left\{1 + \sqrt{2C_0(h) \cdot \frac{1}{\iota_h} \cdot \widetilde{\rho}(h)}\right\},$$

and thus

$$|E_1| \leq 4\sqrt{2}\lambda \frac{H\sqrt{d}}{K} \left[\sum_{h=1}^H A_4(h)\right],$$

where for each $h$,

$$A_4(h) := \frac{H - h + 1}{H} \cdot \frac{1}{\sqrt{\iota_h}} \cdot \|\mathbf{v}_h^\pi\|_{\mathbf{\Lambda}_h^{-1}} \cdot \left[1 + \sqrt{2C_0(h) \cdot \frac{1}{\iota_h} \cdot \widetilde{\rho}(h)}\right],$$

and $C_0(h)$ and $\widetilde{\rho}(h)$ are same constants as in Lemma D.7.

$\square$

### D.3 Bounding the $E_3$ Term in the OPE Decomposition

It remains to bound the term $E_3$ in (E.3) given by:

$$E_3 := \lambda \sum_{h=1}^H (\mathbf{v}_h^\pi)^\top \widehat{\mathbf{\Lambda}}_h^{-1} \mathbf{w}_h^\pi = \sum_{h=1}^H E_{3,h},$$

where $E_{3,h} = \lambda(\mathbf{v}_h^\pi)^\top \widehat{\mathbf{\Lambda}}_h^{-1} \mathbf{w}_h^\pi$. Similar to Lemma D.8, we have the following lemma.

**Lemma D.9.** Under the same event where the result of Lemma D.7 and Lemma D.8 holds, if $K$ satisfies (D.16), (D.17), (D.20) and (D.21), we have

$$|E_3| \leq 4\sqrt{2}\lambda \left(\sum_{h=1}^H A_5(h)\right) \cdot \frac{H\sqrt{d}}{K},$$

where

$$A_5(h) = \frac{1}{\sqrt{\iota_h}} \cdot \|\mathbf{v}_h^\pi\|_{\mathbf{\Lambda}_h^{-1}} \cdot \left\{1 + \sqrt{2C_0(h) \cdot \frac{1}{\iota_h} \cdot \widetilde{\rho}(h)}\right\},$$

and $C_0(h)$ and $\widetilde{\rho}(h)$ are same constants as in Lemma D.7.

*Proof of Lemma D.9.* First note that

$$|E_{3,h}| \leq \lambda \cdot \|\mathbf{v}_h^\pi\|_{\widehat{\mathbf{\Lambda}}_h^{-1}} \cdot \|\mathbf{w}_h^\pi\|_{\widehat{\mathbf{\Lambda}}_h^{-1}}$$

$$\leq \lambda \cdot \|\mathbf{v}_h^\pi\|_{\widehat{\mathbf{\Lambda}}_h^{-1}} \cdot \|\mathbf{w}_h^\pi\|_2 \cdot \left\|\widehat{\mathbf{\Lambda}}_h^{-1}\right\|^{1/2}$$

$$\leq \lambda \cdot \frac{2}{\sqrt{K}} \cdot \left\{\|\mathbf{v}_h^\pi\|_{\mathbf{\Lambda}_h^{-1}} + \sqrt{2C_0(h) \cdot \left\|\mathbf{\Lambda}_h^{-1}\right\| \cdot \widetilde{\rho}(h)} \cdot \|\mathbf{v}_h^\pi\|_{\mathbf{\Lambda}_h^{-1}}\right\} \cdot \frac{\sqrt{2}}{\sqrt{K}} \left\|\mathbf{\Lambda}_h^{-1}\right\|^{1/2} \cdot 2H\sqrt{d}$$

$$= 4\sqrt{2}\lambda \cdot \frac{1}{\sqrt{\iota_h}} \cdot \|\mathbf{v}_h^\pi\|_{\mathbf{\Lambda}_h^{-1}} \cdot \left\{1 + \sqrt{2C_0(h) \cdot \frac{1}{\iota_h} \cdot \widetilde{\rho}(h)}\right\} \cdot \frac{H\sqrt{d}}{K},$$

where the third step is by (D.22), (D.23) and Proposition 2.2. We then conclude that

$$|E_3| \le 4\sqrt{2}\lambda \left(\sum_{h=1}^{H} A_5(h)\right) \cdot \frac{H\sqrt{d}}{K},$$

where for each $h \in [H]$,

$$A_5(h) = \frac{1}{\sqrt{\iota_h}} \cdot \|\mathbf{v}_h^{\pi}\|_{\mathbf{\Lambda}_h^{-1}} \cdot \left\{1 + \sqrt{2C_0(h) \cdot \frac{1}{\iota_h} \cdot \widetilde{\rho}(h)}\right\}.$$

$\square$

## D.4 Proof of Theorem D.1

*Proof of Theorem D.1.* By (4.3), and Lemmas D.7, D.8 and D.9, we have that with probability at least $1 - \delta$,

$$\begin{aligned}
|v_1^{\pi} - \widehat{v}_1^{\pi}| \le &\sqrt{2\log\left(\frac{16H}{\delta}\right)B} \cdot \left[\sum_{h=1}^{H} \|\mathbf{v}_h^{\pi}\|_{\mathbf{\Lambda}_h^{-1}}\right] \cdot \frac{1}{\sqrt{K}} \\
&+ \frac{16\sqrt{2}}{3}\log\left(\frac{16H}{\delta}\right) \cdot \sqrt{B} \cdot \left[\sum_{h=1}^{H} A_1(h)\right] \cdot \frac{1}{K^{3/4}} \\
&+ \frac{16\sqrt{2}}{3}\log\left(\frac{16H}{\delta}\right) \cdot \sqrt{B} \cdot \left[\sum_{h=1}^{H} (A_2(h) + A_3(h))\right] \cdot \frac{1}{K} \\
&+ 4\sqrt{2}\lambda \left[\sum_{h=1}^{H} (A_4(h) + A_5(h))\right] \cdot \frac{H\sqrt{d}}{K}.
\end{aligned} \tag{D.24}$$

We now compute a lower bound for $K$. This comes from the lower bound of $K$ required by Theorem C.2, Lemma D.7, Lemma D.8 and Lemma D.9. Recall (D.21), (D.16), (D.17) and (D.20):

$$\begin{aligned}
K &\ge \max_{h \in [H]} \frac{16\|\mathbf{\Lambda}_h^{-1}\|^2}{(\eta_h + \sigma_r^2)^4} \cdot \left(C_{K,h,\delta}(H - h + 1)^2\sqrt{d} + 4(H - h + 1) \cdot \widetilde{C}(h) \cdot d\right)^2, \\
K &\ge \max_{h \in [H]} \max\left\{\frac{911}{(\eta_h + \sigma_r^2)^2\iota_h^2}\log\left(\frac{8Hd}{\delta}\right), \frac{6\lambda}{\iota_h}\right\}, \\
K &\ge C \cdot \frac{H^2 d^2}{\kappa^2}\log\left(\frac{dHK}{\kappa\delta}\right) \cdot \max_{h \in [H]} \frac{(H - h + 1)^2}{(\eta_h + \sigma_r^2)^2} \cdot \max_{h \in [H]} \frac{(H - h + 1)^2}{\iota_h(\eta_h + \sigma_r^2)}, \\
K &\ge 1152 \cdot \left[\frac{1}{2}\log\left(\frac{\lambda + K}{\lambda}\right) + \frac{1}{d}\log\frac{16H}{\delta}\right] \cdot \frac{H^4 d}{\kappa^2\sigma_r^4}.
\end{aligned} \tag{D.25}$$

It remains to simplify the expression. For the first lower bound in (D.25), note that $\widetilde{C}(h) \ge C_{K,h,\delta} \cdot H$, and thus

$$C_{K,h,\delta}(H - h + 1)^2\sqrt{d} + 4(H - h + 1) \cdot \widetilde{C}(h) \cdot d < 8(H - h + 1) \cdot \widetilde{C}(h) \cdot d. \tag{D.26}$$

Therefore, it suffices to let $K$ satisfy

$$K \ge \max_{h \in [H]} C^2 \cdot \frac{1024}{\iota_h^2(\eta_h + \sigma^2)^4} \cdot (H - h + 1)^2 d^2 \cdot \left[\sum_{i=h}^{H} \frac{H - h + 1}{\sqrt{\iota_h(\eta_h + \sigma_r^2)}}\right]^2 \cdot \left[\log\left(\frac{dH^2 K}{\kappa\delta}\right)\right]^2, \tag{D.27}$$

where $C$ is the problem-independent universal constant from the proof of Theorem C.2. The second lower bound in (D.25) is much smaller than (D.27) and thus can be omitted. We then consider the third and the fourth lower bound together. They can be combined into

$$K \ge C \cdot \frac{H^2 d^2}{\sigma_r^4\kappa^2} \cdot \max\left\{\max_{h \in [H]} \frac{(H - h + 1)^2}{(\eta_h + \sigma_r^2)^2} \cdot \max_{h \in [H]} \frac{(H - h + 1)^2}{\iota_h(\eta_h + \sigma_r^2)}, H^2\right\} \cdot \log\left(\frac{dHK}{\kappa\delta}\right). \tag{D.28}$$

Denote
$$C_{h,3} := \frac{(H-h+1)^2}{\eta_h + \sigma_r^2}.$$

Then (D.27) is simplified to

$$K \geq C \max_{h \in [H]} \frac{C_{h,3} d^2}{\iota_h^2 (\eta_h + \sigma_r^2)^3} \cdot \left[ \sum_{i=h}^{H} \sqrt{\frac{C_{h,3}}{\iota_h}} \right]^2 \cdot \left[ \log \left( \frac{dH^2 K}{\kappa \delta} \right) \right]^2, \tag{D.29}$$

and (D.28) can be simplified

$$K \geq C \cdot \frac{H^2 d^2}{\sigma_r^4 \kappa^2} \cdot \max \left\{ \max_{h \in [H]} \frac{C_{h,3}}{\eta_h + \sigma_r^2} \cdot \max_{h \in [H]} \frac{C_{h,3}}{\iota_h}, H^2 \right\} \cdot \log \left( \frac{dHK}{\kappa \delta} \right). \tag{D.30}$$

We then combine (D.29) and (D.30) and get that

$$K \geq C \cdot C_3(h) \cdot d^2 \left[ \log \left( \frac{dH^2 K}{\kappa \delta} \right) \right]^2,$$

where $C$ is some problem-independent universal constant and

$$C_3(h) := \max \left\{ \max_{h \in [H]} \frac{C_{h,3}}{\iota_h^2 (\eta_h + \sigma_r^2)^3} \cdot \left[ \sum_{i=h}^{H} \sqrt{\frac{C_{h,3}}{\iota_h}} \right]^2, \frac{H^2}{\sigma_r^4 \kappa^2} \cdot \left( \max_{h \in [H]} \frac{C_{h,3}}{\eta_h + \sigma_r^2} \cdot \max_{h \in [H]} \frac{C_{h,3}}{\iota_h} \right), \frac{H^4}{\sigma_r^4 \kappa^2} \right\}.$$

To simplify the upper bound given by (D.24), first note that by the choice of $K$, we have $B < 2$. By (D.26), we have that $\tilde{\rho}(h)$ satisfies

$$\tilde{\rho}(h) \leq 8C \frac{1}{\sqrt{K}} \cdot \frac{(H-h+1)d}{(\eta_h + \sigma_r^2)^2} \cdot \left[ \sum_{i=h}^{H} \frac{H-h+1}{\sqrt{\iota_h (\eta_h + \sigma_r^2)}} \right] \cdot \log \left( \frac{dH^2 K}{\kappa \delta} \right). \tag{D.31}$$

It follows that

$$A_1(h) \leq C \cdot \sqrt{C_0(h) \cdot \frac{(H-h+1)d}{(\eta_h + \sigma_r^2)^2} \cdot \left[ \sum_{i=h}^{H} \frac{H-h+1}{\sqrt{\iota_h (\eta_h + \sigma_r^2)}} \right] \cdot \frac{1}{\iota_h} \cdot \log \left( \frac{dH^2 K}{\kappa \delta} \right)} \cdot \| \mathbf{v}_h^\pi \|_{\mathbf{\Lambda}_h^{-1}}, \tag{D.32}$$

for some (different) universal constant $C$. Also, it is not hard to see $A_2(h)$ and $A_3(h)$ are less than the RHS of (D.32) up to a constant factor by our choice of $K$, which gives

$$\sum_{h=1}^{H} A_2(h) + A_3(h)$$

$$\leq C \sum_{h=1}^{H} \sqrt{C_0(h) \cdot \frac{(H-h+1)d}{(\eta_h + \sigma_r^2)^2} \cdot \left[ \sum_{i=h}^{H} \frac{H-h+1}{\sqrt{\iota_h (\eta_h + \sigma_r^2)}} \right] \cdot \frac{1}{\iota_h} \cdot \log \left( \frac{dH^2 K}{\kappa \delta} \right)} \cdot \| \mathbf{v}_h^\pi \|_{\mathbf{\Lambda}_h^{-1}}, \tag{D.33}$$

for some universal constant $C$. To bound $A_4(h) + A_5(h)$, note that $A_4(h) \leq A_5(h)$ and thus

$$A_4(h) + A_5(h) \leq \frac{2}{\sqrt{\iota_h}} \cdot \| \mathbf{v}_h^\pi \|_{\mathbf{\Lambda}_h^{-1}} \cdot \left\{ 1 + \sqrt{2C_0(h) \cdot \frac{1}{\iota_h} \cdot \tilde{\rho}(h)} \right\}, \tag{D.34}$$

where

$$\frac{2}{\sqrt{\iota_h}} \cdot \| \mathbf{v}_h^\pi \|_{\mathbf{\Lambda}_h^{-1}} \cdot \sqrt{2C_0(h) \cdot \frac{1}{\iota_h} \cdot \tilde{\rho}(h)}$$

$$\leq \frac{C}{K^{1/4}} \sqrt{C_0(h) \frac{(H-h+1)d}{(\eta_h + \sigma_r^2)^2} \cdot \left[ \sum_{i=h}^{H} \frac{H-h+1}{\sqrt{\iota_h (\eta_h + \sigma_r^2)}} \right] \cdot \frac{1}{\iota_h^2} \cdot \log \left( \frac{dH^2 K}{\kappa \delta} \right)} \cdot \| \mathbf{v}_h^\pi \|_{\mathbf{\Lambda}_h^{-1}}.$$

Recall (D.24). By our choice of $K$, it is clear that

$$H\sqrt{d}\cdot\frac{2}{\sqrt{\iota_h}}\cdot\|\mathbf{v}_h^\pi\|_{\mathbf{\Lambda}_h^{-1}}\cdot\sqrt{2C_0(h)\cdot\frac{1}{\iota_h}\cdot\widetilde{\rho}(h)}$$

$$\leq C\cdot\sqrt{C_0(h)\cdot\frac{(H-h+1)d}{(\eta_h+\sigma_r^2)^2}\cdot\left[\sum_{i=h}^H\frac{H-h+1}{\sqrt{\iota_h(\eta_h+\sigma_r^2)}}\right]\cdot\frac{1}{\iota_h}\cdot\log\left(\frac{dH^2K}{\kappa\delta}\right)}\cdot\|\mathbf{v}_h^\pi\|_{\mathbf{\Lambda}_h^{-1}},$$

where the RHS of the above is exactly the RHS of (D.32) up to a constant factor. Therefore, we can combine $[\sum_{h=1}^H A_4(h)+A_5(h)]H\sqrt{d}$ with (D.33), and together with (D.34), the last two terms on the RHS of (D.24) can be upper bounded by

$$\frac{16\sqrt{2}}{3}\log\left(\frac{16H}{\delta}\right)\cdot\sqrt{B}\cdot\left[\sum_{h=1}^H(A_2(h)+A_3(h))\right]\cdot\frac{1}{K}+4\sqrt{2}\lambda\left[\sum_{h=1}^H(A_4(h)+A_5(h))\right]\cdot\frac{H\sqrt{d}}{K}$$

$$\leq C\cdot C_4\cdot\log\left(\frac{16H}{\delta}\right)\cdot\frac{1}{K},\tag{D.35}$$

where $C$ is some universal constant and $C_4$ is given by

$$C_4:=\sum_{h=1}^H\left\{\sqrt{C_0(h)\cdot\frac{(H-h+1)d}{(\eta_h+\sigma_r^2)^2}\cdot\left[\sum_{i=h}^H\frac{H-h+1}{\sqrt{\iota_h(\eta_h+\sigma_r^2)}}\right]\cdot\frac{1}{\iota_h}\cdot\log\left(\frac{dH^2K}{\kappa\delta}\right)}\cdot\|\mathbf{v}_h^\pi\|_{\mathbf{\Lambda}_h^{-1}}\right\}$$

$$=\sum_{h=1}^H\left\{\sqrt{C_0(h)\cdot C_{h,2}\cdot\frac{(H-h+1)d}{\iota_h(\eta_h+\sigma_r^2)^2}\cdot\log\left(\frac{dH^2K}{\kappa\delta}\right)}\cdot\|\mathbf{v}_h^\pi\|_{\mathbf{\Lambda}_h^{-1}}\right\}.$$

Plugging in the formula for $C_0(h)$ given in Lemma D.7 finishes the proof. Note that in Theorem C.2, the notation $C_0(h)$ is changed to $C_{h,4}$.

$\square$

## E   Proof of Error Decomposition

*Proof.* Since $Q_h^\pi(s,a)=\phi(s,a)^\top\mathbf{w}_h^\pi=r_h(s,a)+[\mathbb{P}_hV_{h+1}^\pi](s,a)$ for some vector $\mathbf{w}_h^\pi\in\mathbb{R}^d$, we further have

$$Q_h^\pi(s,a)=\phi(s,a)^\top\widehat{\mathbf{\Lambda}}_h^{-1}\left(\sum_{k=1}^K\frac{\phi(s_{k,h},a_{k,h})\phi(s_{k,h},a_{k,h})^\top}{\widehat{\sigma}_h(s_{k,h},a_{k,h})^2}+\lambda\mathbf{I}_d\right)\mathbf{w}_h^\pi$$

$$=\phi(s,a)^\top\widehat{\mathbf{\Lambda}}_h^{-1}\sum_{k=1}^K\frac{\phi(s_{k,h},a_{k,h})}{\widehat{\sigma}_h(s_{k,h},a_{k,h})^2}Q_h^\pi(s_{k,h},a_{k,h})+\lambda\phi(s,a)^\top\widehat{\mathbf{\Lambda}}_h^{-1}\mathbf{w}_h^\pi$$

$$=\phi(s,a)^\top\widehat{\mathbf{\Lambda}}_h^{-1}\sum_{k=1}^K\frac{\phi(s_{k,h},a_{k,h})}{\widehat{\sigma}_h(s_{k,h},a_{k,h})^2}\left(r_h(s_{k,h},a_{k,h})+[\mathbb{P}_hV_{h+1}^\pi](s_{k,h},a_{k,h})\right)$$

$$+\lambda\phi(s,a)^\top\widehat{\mathbf{\Lambda}}_h^{-1}\mathbf{w}_h^\pi.$$

It follows that

$$Q_h^\pi(s,a)-\widehat{Q}_h^\pi(s,a)=\phi(s_h,a_h)^\top\widehat{\mathbf{\Lambda}}_h^{-1}\sum_{k=1}^K\frac{\phi(s_{k,h},a_{k,h})}{\widehat{\sigma}_h(s_{k,h},a_{k,h})^2}\left([\mathbb{P}_hV_{h+1}^\pi](s_{k,h},a_{k,h})-\widehat{V}_{h+1}^\pi(s_{k,h}')-\epsilon_{k,h}\right)$$

$$+\lambda\phi(s_h,a_h)^\top\widehat{\mathbf{\Lambda}}_h^{-1}\mathbf{w}_h^\pi$$

$$=\phi(s,a)^\top\widehat{\mathbf{\Lambda}}_h^{-1}\sum_{k=1}^K\frac{\phi(s_{k,h},a_{k,h})}{\widehat{\sigma}_h(s_{k,h},a_{k,h})^2}\left([\mathbb{P}_hV_{h+1}^\pi](s_{k,h},a_{k,h})-[\mathbb{P}_h\widehat{V}_{h+1}^\pi](s_{k,h},a_{k,h})\right)$$

$$+\phi(s,a)^\top\widehat{\mathbf{\Lambda}}_h^{-1}\sum_{k=1}^K\frac{\phi(s_{k,h},a_{k,h})}{\widehat{\sigma}_h(s_{k,h},a_{k,h})^2}\left([\mathbb{P}_h\widehat{V}_{h+1}^\pi](s_{k,h},a_{k,h})-\widehat{V}_{h+1}^\pi(s_{k,h}')-\epsilon_{k,h}\right)$$

$$+\lambda\phi(s,a)^\top\widehat{\mathbf{\Lambda}}_h^{-1}\mathbf{w}_h^\pi.$$

where $\epsilon_{k,h}$ is the noise in reward. Note that

$$[\mathbb{P}_h V_{h+1}^\pi](s_{k,h}, a_{k,h}) - [\mathbb{P}_h \widehat{V}_{h+1}^\pi](s_{k,h}, a_{k,h}) = \int_{\mathcal{S}} \left( V_{h+1}^\pi(s) - \widehat{V}_{h+1}^\pi(s) \right) \langle \phi(s_{k,h}, a_{k,h}), \boldsymbol{\mu}_h(s) \rangle \mathrm{d}s$$

$$= \phi(s_{k,h}, a_{k,h})^\top \int_{\mathcal{S}} \left( V_{h+1}^\pi(s) - \widehat{V}_{h+1}^\pi(s) \right) \boldsymbol{\mu}_h(s) \mathrm{d}s,$$

and thus

$$Q_h^\pi(s,a) - \widehat{Q}_h^\pi(s,a)$$

$$= \phi(s,a)^\top \widehat{\boldsymbol{\Lambda}}_h^{-1} \sum_{k=1}^K \frac{\phi(s_{k,h}, a_{k,h})\phi(s_{k,h}, a_{k,h})^\top}{\widehat{\sigma}(s_{k,h}, a_{k,h})^2} \int_{\mathcal{S}} \left( V_{h+1}^\pi(s') - \widehat{V}_{h+1}^\pi(s') \right) \boldsymbol{\mu}_h(s') \mathrm{d}s'$$

$$+ \phi(s,a)^\top \widehat{\boldsymbol{\Lambda}}_h^{-1} \sum_{k=1}^K \frac{\phi(s_{k,h}, a_{k,h})}{\widehat{\sigma}_h(s_{k,h}, a_{k,h})^2} \left( [\mathbb{P}_h \widehat{V}_{h+1}^\pi](s_{k,h}, a_{k,h}) - \widehat{V}_{h+1}^\pi(s_{k,h}') - \epsilon_{k,h} \right)$$

$$+ \lambda \phi(s_h, a_h)^\top \widehat{\boldsymbol{\Lambda}}_h^{-1} \mathbf{w}_h^\pi$$

$$= [\mathbb{P}_h(V_{h+1}^\pi - \widehat{V}_{h+1}^\pi)](s,a) - \lambda \phi(s,a)^\top \widehat{\boldsymbol{\Lambda}}_h^{-1} \int_{\mathcal{S}} \left( V_{h+1}^\pi(s) - \widehat{V}_{h+1}^\pi(s) \right) \boldsymbol{\mu}_h(s) \mathrm{d}s$$

$$+ \phi(s,a)^\top \widehat{\boldsymbol{\Lambda}}_h^{-1} \sum_{k=1}^K \frac{\phi(s_{k,h}, a_{k,h})}{\widehat{\sigma}_h(s_{k,h}, a_{k,h})^2} \left( [\mathbb{P}_h \widehat{V}_{h+1}^\pi](s_{k,h}, a_{k,h}) - \widehat{V}_{h+1}^\pi(s_{k,h}') - \epsilon_{k,h} \right)$$

$$+ \lambda \phi(s,a)^\top \widehat{\boldsymbol{\Lambda}}_h^{-1} \mathbf{w}_h^\pi. \tag{E.1}$$

Then by the Bellman equation, we have

$$V_h^\pi(s) - \widehat{V}_h^\pi(s) = \mathbb{J}_h(Q_h^\pi - \widehat{Q}_h^\pi)(s)$$

$$= \mathbb{J}_h \mathbb{P}_h(V_{h+1}^\pi - \widehat{V}_{h+1}^\pi)(s) - \lambda \mathbb{J}_h \phi(s)^\top \widehat{\boldsymbol{\Lambda}}_h^{-1} \int_{\mathcal{S}} \left( V_{h+1}^\pi(s') - \widehat{V}_{h+1}^\pi(s') \right) \boldsymbol{\mu}_h(s') \mathrm{d}s'$$

$$+ \mathbb{J}_h \phi(s)^\top \widehat{\boldsymbol{\Lambda}}_h^{-1} \sum_{k=1}^K \frac{\phi(s_{k,h}, a_{k,h})}{\widehat{\sigma}_h(s_{k,h}, a_{k,h})^2} \left( [\mathbb{P}_h \widehat{V}_{h+1}^\pi](s_{k,h}, a_{k,h}) - \widehat{V}_{h+1}^\pi(s_{k,h}') - \epsilon_{k,h} \right)$$

$$+ \lambda \mathbb{J}_h \phi(s)^\top \widehat{\boldsymbol{\Lambda}}_h^{-1} \mathbf{w}_h^\pi, \tag{E.2}$$

where $\mathbb{J}_h f(\cdot) = \int_{\mathcal{A}} f(\cdot, a)\pi_h(a|\cdot)\mathrm{d}a$ for any function $f : \mathcal{S} \times \mathcal{A} \to \mathbb{R}$. Recursively expanding the above equation, we obtain

$$V_1^\pi(s) - \widehat{V}_1^\pi(s)$$

$$= -\lambda \sum_{h=1}^H \left( \prod_{i=1}^{h-1} \mathbb{J}_i \mathbb{P}_i \right) \mathbb{J}_h \phi(s)^\top \widehat{\boldsymbol{\Lambda}}_h^{-1} \int_{\mathcal{S}} \left( V_{h+1}^\pi(s') - \widehat{V}_{h+1}^\pi(s') \right) \boldsymbol{\mu}_h(s') \mathrm{d}s'$$

$$+ \sum_{h=1}^H \left( \prod_{i=1}^{h-1} \mathbb{J}_i \mathbb{P}_i \right) \mathbb{J}_h \phi(s)^\top \widehat{\boldsymbol{\Lambda}}_h^{-1} \sum_{k=1}^K \frac{\phi(s_{k,h}, a_{k,h})}{\widehat{\sigma}_h(s_{k,h}, a_{k,h})^2} \left( [\mathbb{P}_h \widehat{V}_{h+1}^\pi](s_{k,h}, a_{k,h}) - \widehat{V}_{h+1}^\pi(s_{k,h}') - \epsilon_{k,h} \right)$$

$$+ \lambda \sum_{h=1}^H \left( \prod_{i=1}^{h-1} \mathbb{J}_i \mathbb{P}_i \right) \mathbb{J}_h \phi(s_1)^\top \widehat{\boldsymbol{\Lambda}}_h^{-1} \mathbf{w}_h^\pi. \tag{E.3}$$

Here with a slight abuse of notation we define $\prod_{i=1}^{h-1} \mathbb{J}_i \mathbb{P}_i = 1$ when $h = 1$. We then have

$$v_1^\pi - \widehat{v}_1^\pi = -\lambda \sum_{h=1}^H (\mathbf{v}_h^\pi)^\top \widehat{\boldsymbol{\Lambda}}_h^{-1} \int_{\mathcal{S}} \left( V_{h+1}^\pi(s) - \widehat{V}_{h+1}^\pi(s) \right) \boldsymbol{\mu}_h(s) \mathrm{d}s$$

$$+ \sum_{h=1}^H (\mathbf{v}_h^\pi)^\top \widehat{\boldsymbol{\Lambda}}_h^{-1} \sum_{k=1}^K \frac{\phi(s_{k,h}, a_{k,h})}{\widehat{\sigma}_h(s_{k,h}, a_{k,h})^2} \left( [\mathbb{P}_h \widehat{V}_{h+1}^\pi](s_{k,h}, a_{k,h}) - \widehat{V}_{h+1}^\pi(s_{k,h}') - \epsilon_{k,h} \right)$$

$$+ \lambda \sum_{h=1}^H (\mathbf{v}_h^\pi)^\top \widehat{\boldsymbol{\Lambda}}_h^{-1} \mathbf{w}_h^\pi$$

$$:= E_1 + E_2 + E_3, \tag{E.4}$$

where for simplicity we write $\mathbf{v}_h^\pi = \mathbb{E}_\pi\left[\left(\prod_{i=1}^{h-1}\mathbb{J}_i\mathbb{P}_i\right)\mathbb{J}_h\phi(s_1)\Big|s_1 \sim \xi_1\right] = \mathbb{E}_{\pi,h}[\phi(s_h,a_h)]$ by recalling the definition of $\mathbb{E}_{\pi,h}[\cdot]$ given in the text following (2.4). $\square$

# F   Lemmas for Uniform Convergence

All lemmas in this section are under the Assumption of Theorem C.2.

## F.1   Convergence of $\widehat{\sigma}$

**Lemma F.1.** For any $h \in [H]$ and any $\widehat{V}_{h+1}^\pi \in \mathcal{V}_{h+1}(L)$, with probability at least $1-\delta$, it holds for all $(s,a) \in \mathcal{S} \times \mathcal{A}$ that

$$\left|\langle\phi(s,a),\widehat{\boldsymbol{\beta}}_h\rangle_{[0,(H-h+1)^2]} - \mathbb{P}_h(\widehat{V}_{h+1})^2(s,a)\right| \leq C'_{K,\delta} \cdot \frac{(H-h+1)^2\sqrt{d}}{\sqrt{K}}\left[\frac{1}{2}\log\left(\frac{\lambda+K}{\lambda}\right) + \frac{1}{d}\log\frac{4}{\delta}\right]^{1/2},$$

and

$$\left|\langle\phi(s,a),\widehat{\boldsymbol{\theta}}_h\rangle_{[0,H-h+1]} - \mathbb{P}_h(\widehat{V}_{h+1})(s,a)\right| \leq C'_{K,\delta} \cdot \frac{(H-h+1)\sqrt{d}}{\sqrt{K}}\left[\frac{1}{2}\log\left(\frac{\lambda+K}{\lambda}\right) + \frac{1}{d}\log\frac{4}{\delta}\right]^{1/2},$$

where

$$C'_{K,\delta} := 4\sqrt{2}\frac{1}{\sqrt{\kappa_h}} + 4\lambda \cdot \frac{1}{\kappa_h} \cdot \left[\frac{1}{2}\log\left(\frac{\lambda+K}{\lambda}\right) + \frac{1}{d}\log\frac{4}{\delta}\right]^{-1/2}.$$

*Proof of Lemma F.1.* First we consider $\langle\phi(s,a),\widehat{\boldsymbol{\beta}}_h\rangle_{[0,(H-h+1)^2]}$.    Note that, since $\mathbb{P}_h(\widehat{V}_{h+1}^\pi)^2(s,a) \in [0,(H-h+1)^2]$,

$$|\langle\phi(s,a),\widehat{\boldsymbol{\beta}}_h\rangle_{[0,(H-h+1)^2]} - \mathbb{P}_h(\widehat{V}_{h+1}^\pi)^2(s,a)| \leq |\langle\phi(s,a),\widehat{\boldsymbol{\beta}}_h\rangle - \mathbb{P}_h(\widehat{V}_{h+1}^\pi)^2(s,a)|.$$

It then suffices to bound the RHS.

$$\langle\phi(s,a),\widehat{\boldsymbol{\beta}}_h\rangle - \mathbb{P}_h(\widehat{V}_{h+1})^2(s,a)$$

$$= \phi(s,a)^\top(\widehat{\boldsymbol{\Sigma}}_h)^{-1}\sum_{k=1}^K\phi(\check{s}_{k,h},\check{a}_{k,h})\widehat{V}_{h+1}^\pi(\check{s}'_{k,h})^2 - \mathbb{P}_h(\widehat{V}_{h+1}^\pi)^2(s,a)$$

$$= \phi(s,a)^\top(\widehat{\boldsymbol{\Sigma}}_h)^{-1}\sum_{k=1}^K\phi(\check{s}_{k,h},\check{a}_{k,h})\widehat{V}_{h+1}^\pi(\check{s}'_{k,h})^2 - \phi(s,a)^\top\int_\mathcal{S}(\widehat{V}_{h+1}^\pi)^2(s')\mathrm{d}\boldsymbol{\mu}_h(s').$$

Note that

$$\phi(s,a)^\top\int_\mathcal{S}(\widehat{V}_{h+1}^\pi)^2(s')\mathrm{d}\boldsymbol{\mu}_h(s')$$

$$= \phi(s,a)^\top(\widehat{\boldsymbol{\Sigma}}_h)^{-1}\left(\sum_{k=1}^K\phi(\check{s}_{k,h},\check{a}_{k,h})\phi(\check{s}_{k,h},\check{a}_{k,h})^\top + \lambda\mathbf{I}_d\right)\int_\mathcal{S}(\widehat{V}_{h+1}^\pi)^2(s')\mathrm{d}\boldsymbol{\mu}_h(s')$$

$$= \phi(s,a)^\top(\widehat{\boldsymbol{\Sigma}}_h)^{-1}\sum_{k=1}^K\phi(\check{s}_{k,h},\check{a}_{k,h})\mathbb{P}_h(\widehat{V}_{h+1}^\pi)^2(\check{s}_{k,h},\check{a}_{k,h}) + \lambda\phi(s,a)(\widehat{\boldsymbol{\Sigma}}_h)^{-1}\int_\mathcal{S}(\widehat{V}_{h+1}^\pi)^2(s')\mathrm{d}\boldsymbol{\mu}_h(s'),$$

and it follows that

$$\langle\phi(s,a),\widehat{\boldsymbol{\beta}}_h\rangle - \mathbb{P}_h(\widehat{V}_{h+1})^2(s,a)$$

$$= \underbrace{\phi(s,a)^\top(\widehat{\boldsymbol{\Sigma}}_h)^{-1}\sum_{k=1}^K\phi(\check{s}_{k,h},\check{a}_{k,h})\left[(\widehat{V}_{h+1}^\pi)^2(\check{s}'_{k,h}) - \mathbb{P}_h(\widehat{V}_{h+1}^\pi)^2(\check{s}_{k,h},\check{a}_{k,h})\right]}_{A_1(s,a)}$$

$$\underbrace{-\lambda\phi(s,a)^\top(\widehat{\boldsymbol{\Sigma}}_h)^{-1}\int_\mathcal{S}(\widehat{V}_{h+1}^\pi)^2(s')\mathrm{d}\boldsymbol{\mu}_h(s')}_{A_2(s,a)}.$$

To bound $|A_1|$, we first apply Cauchy-Schwartz inequality to obtain that

$$|E_1(s,a)| \leq \|\phi(s,a)\|_{\widehat{\boldsymbol{\Sigma}}_h^{-1}} \cdot \left\| \sum_{k=1}^{K} \phi(\check{s}_{k,H}, \check{a}_{k,H}) \left[ (\widehat{V}_{h+1}^\pi)^2(\check{s}'_{k,h}) - \mathbb{P}_h(\widehat{V}_{h+1}^\pi)^2(\check{s}_{k,h}, \check{a}_{k,h}) \right] \right\|_{\widehat{\boldsymbol{\Sigma}}_h^{-1}}.$$

By Lemma H.5, if $K$ satisfies

$$K \geq \max \left\{ 512 \|\boldsymbol{\Sigma}_h^{-1}\|^2 \log\left( \frac{4d}{\delta} \right), 4\lambda \|\boldsymbol{\Sigma}_h^{-1}\| \right\}, \tag{F.1}$$

then with probability at least $1 - \delta/2$, for all $(s,a) \in \mathcal{S} \times \mathcal{A}$,

$$\|\phi(s,a)\|_{\widehat{\boldsymbol{\Sigma}}_h^{-1}} \leq \frac{2}{\sqrt{K}} \cdot \|\phi(s,a)\|_{\boldsymbol{\Sigma}_h^{-1}}.$$

By Lemma G.5, for fixed $\widehat{V}_{h+1}^\pi$, with probability at least $1 - \delta/2$, we have

$$\left\| \sum_{k=1}^{K} \phi(\check{s}_{k,h}, \check{a}_{k,h}) \left[ (\widehat{V}_{h+1}^\pi)^2(\check{s}'_{k,h}) - \mathbb{P}_h(\widehat{V}_{h+1}^\pi)^2(\check{s}_{k,h}, \check{a}_{k,h}) \right] \right\|_{\widehat{\boldsymbol{\Sigma}}_h^{-1}}$$
$$\leq 2\sqrt{2}(H-h+1)^2 \left[ \frac{d}{2} \log\left( \frac{\lambda + K}{\lambda} \right) + \log \frac{4}{\delta} \right]^{1/2}.$$

Combining the two inequalities above, we have that, with probability at least $1 - \delta$,

$$|A_1(s,a)| \leq 2\sqrt{2}(H-h+1)^2 \left[ \frac{d}{2} \log\left( \frac{\lambda + K}{\lambda} \right) + \log \frac{4}{\delta} \right]^{1/2} \cdot \frac{2}{\sqrt{K}} \cdot \|\phi(s,a)\|_{\boldsymbol{\Sigma}_h^{-1}}$$
$$\leq 4\sqrt{2} \|\boldsymbol{\Sigma}_h^{-1}\|^{1/2} \left[ \frac{1}{2} \log\left( \frac{\lambda + K}{\lambda} \right) + \frac{1}{d} \log \frac{4}{\delta} \right]^{1/2} \cdot \frac{(H-h+1)^2\sqrt{d}}{\sqrt{K}},$$

for all $(s,a)$. At the same time, we can bound $A_2$ as

$$|A_2(s,a)| \leq \lambda \|\phi(s,a)\|_{\widehat{\boldsymbol{\Sigma}}_h^{-1}} \cdot \left\| \int_{\mathcal{S}} (\widehat{V}_{h+1}^\pi)^2(s') \mathrm{d}\boldsymbol{\mu}_h(s') \right\|_{\widehat{\boldsymbol{\Sigma}}_h^{-1}}$$
$$\leq \lambda \cdot \frac{2}{\sqrt{K}} \|\phi(s,a)\|_{\boldsymbol{\Sigma}_h^{-1}} \cdot \frac{2}{\sqrt{K}} \left\| \int_{\mathcal{S}} (\widehat{V}_{h+1}^\pi)^2(s') \mathrm{d}\boldsymbol{\mu}_h(s') \right\|_{\boldsymbol{\Sigma}_h^{-1}}$$
$$\leq 4\lambda \|\boldsymbol{\Sigma}_h^{-1}\| \cdot \frac{(H-h+1)^2\sqrt{d}}{K},$$

where the last step is by Assumption 2.1. We then conclude that, if $K$ satisfies (F.1), then with probability at least $1 - \delta$, for all $(s,a)$,

$$\left| \langle \phi(s,a), \widehat{\boldsymbol{\beta}}_h \rangle - \mathbb{P}_h(\widehat{V}_{h+1})^2(s,a) \right|$$
$$\leq |A_1(s,a)| + |A_2(s,a)|$$
$$\leq 4\sqrt{2} \|\boldsymbol{\Sigma}_h^{-1}\|^{1/2} \left[ \frac{1}{2} \log\left( \frac{\lambda + K}{\lambda} \right) + \frac{1}{d} \log \frac{4}{\delta} \right]^{1/2} \cdot \frac{(H-h+1)^2\sqrt{d}}{\sqrt{K}} + 4\lambda \|\boldsymbol{\Sigma}_h^{-1}\| \cdot \frac{(H-h+1)^2\sqrt{d}}{K}$$
$$= 4\sqrt{2} \frac{1}{\sqrt{\kappa_h}} \left[ \frac{1}{2} \log\left( \frac{\lambda + K}{\lambda} \right) + \frac{1}{d} \log \frac{4}{\delta} \right]^{1/2} \cdot \frac{(H-h+1)^2\sqrt{d}}{\sqrt{K}} + \frac{4\lambda}{\kappa_h} \cdot \frac{(H-h+1)^2\sqrt{d}}{K},$$

where in the last step we use the definition $\kappa_h := \lambda_{\min}(\boldsymbol{\Sigma}_h)$. Note that by Assumption 2.3, we have $\kappa_h > 0$ for all $h \in [H]$. At the same time, we can bound $\langle \phi(s,a), \widehat{\boldsymbol{\theta}}_h \rangle_{[0, H-h+1]} - \mathbb{P}_h(\widehat{V}_{h+1})(s,a)$ in a similar way as

$$\left| \langle \phi(s,a), \widehat{\boldsymbol{\theta}}_h \rangle_{[0, H-h+1]} - \mathbb{P}_h(\widehat{V}_{h+1})(s,a) \right|$$
$$\leq 4\sqrt{2} \frac{1}{\sqrt{\kappa_h}} \left[ \frac{1}{2} \log\left( \frac{\lambda + K}{\lambda} \right) + \frac{1}{d} \log \frac{4}{\delta} \right]^{1/2} \cdot \frac{(H-h+1)\sqrt{d}}{\sqrt{K}} + \frac{4\lambda}{\kappa_h} \cdot \frac{(H-h+1)\sqrt{d}}{K}.$$

$\square$

**Lemma F.2.** For any $h \in [H]$ and any $\widehat{V}_{h+1}^\pi \in \mathcal{V}_{h+1}(L)$, with probability at least $1 - \delta$, it holds for all $(s, a) \in \mathcal{S} \times \mathcal{A}$ that

$$\left| \widehat{\sigma}_h^2(s, a) - \sigma_r^2 - \max\left\{ \eta_h, \, \mathbb{V}_h \widehat{V}_{h+1}^\pi(s, a) \right\} \right| \leq \frac{C_{K,h,\delta}(H - h + 1)^2 \sqrt{d}}{\sqrt{K}}$$

$$\leq \frac{20(H - h + 1)^2 \sqrt{d}}{\kappa_h \sqrt{K}} \cdot \sqrt{\log\left( \frac{K}{\lambda \delta} \right)},$$

where

$$C_{K,h,\delta} = 12\sqrt{2} \frac{1}{\sqrt{\kappa_h}} \cdot \left[ \frac{1}{2} \log\left( \frac{\lambda + K}{\lambda} \right) + \frac{1}{d} \log \frac{4}{\delta} \right]^{1/2} + 12\lambda \frac{1}{\kappa_h}. \tag{F.2}$$

*Proof of Lemma F.2.* Recall that by definition,

$$\mathbb{V}_h \widehat{V}_{h+1}^\pi(s, a) = \mathbb{P}_h(\widehat{V}_{h+1}^\pi)^2(s, a) - \left( \mathbb{P}_h \widehat{V}_{h+1}^\pi(s, a) \right)^2.$$

We then have

$$\left| \left[ \langle \boldsymbol{\phi}(s, a), \widehat{\boldsymbol{\beta}}_h \rangle_{[0,(H-h+1)^2]} - \left( \langle \boldsymbol{\phi}(s, a), \widehat{\boldsymbol{\theta}}_h \rangle_{[0,H-h+1]} \right)^2 \right] - \mathbb{V}_h \widehat{V}_{h+1}^\pi(s, a) \right|$$

$$\leq \left| \langle \boldsymbol{\phi}(s, a), \widehat{\boldsymbol{\beta}}_h \rangle_{[0,(H-h+1)^2]} - \mathbb{P}_h(\widehat{V}_{h+1}^\pi)^2(s, a) \right| + 2(H - h + 1) \cdot \left| \langle \boldsymbol{\phi}(s, a), \widehat{\boldsymbol{\theta}}_h \rangle_{[0,H-h+1]} - \mathbb{P}_h \widehat{V}_{h+1}^\pi(s, a) \right|,$$

and the rest follows from Lemma F.1 and the fact that $\max\{\eta_h, \cdot\}$ is a contraction mapping. $\square$

**Lemma F.3.** For any $h \in [H - 1]$, let $V \in \mathcal{V}_{h+1}(L) \cap \{V : \sup_{s \in \mathcal{S}} |V(s) - V_{h+1}^\pi(s)| \leq \rho\}$ for some sufficiently small $\rho < (\eta_h + \sigma_r^2)/[12(H - h + 1)]$. Suppose $K$ satisfies that

$$K \geq \frac{3600(H - h + 1)^4 d}{\kappa_h^2 \inf_{s,a} \sigma_h(s, a)^2} \cdot \log\left( \frac{Kd}{\lambda \delta} \right) \tag{F.3}$$

Then for any $\delta \in (0, 1)$, it holds with probability at least $1 - \delta$ that

$$\left\| \left( \frac{\widehat{\boldsymbol{\Lambda}}_h}{K} \right)^{-1} \right\| \leq \frac{4}{\iota_h}.$$

*Proof of Lemma F.3.* By Lemma F.2, there exists an event $\check{\mathcal{E}}$ over $\{(\check{s}_{k,h}, \check{a}_{k,h}), k \in [K]\}$ such that $\mathbb{P}(\check{\mathcal{E}}) \geq 1 - \delta$ and on this event it holds for all $(s, a) \in \mathcal{S} \times \mathcal{A}$ that

$$\left| \widehat{\sigma}_h^2(s, a) - \sigma_r^2 - \max\left\{ \eta_h, \, \mathbb{V}_h V_{h+1}^\pi(s, a) \right\} \right| \leq \frac{20(H - h + 1)^2 \sqrt{d}}{\kappa_h \sqrt{K}} \sqrt{\log\left( \frac{K}{\lambda \delta} \right)} + 4(H - h + 1) \cdot \rho.$$

Then by (F.3) and the assumption on $\rho$, we have

$$\frac{1}{3} \sigma_h(s, a) \leq \widehat{\sigma}_h(s, a) \leq \frac{5}{3} \sigma_h(s, a) \tag{F.4}$$

for all $(s, a) \in \mathcal{S} \times \mathcal{A}$. In the following argument we condition on $\check{\mathcal{E}}$, and this will not affect the distribution of $\{(s_{k,h}, a_{k,h}), k \in [K]\}$ by independence.

Recall that

$$\widehat{\boldsymbol{\Lambda}}_h = \sum_{k=1}^K \widehat{\sigma}_h(s_{k,h}, a_{k,h})^{-2} \boldsymbol{\phi}(s_{k,h}, a_{k,h}) \boldsymbol{\phi}(s_{k,h}, a_{k,h})^\top + \lambda \mathbf{I}_d.$$

Since $\widehat{\sigma}_h \geq \inf_{s,a} \sigma_h(s, a)/3$, it then follows from Lemma H.1 that

$$\left\| \frac{\widehat{\boldsymbol{\Lambda}}_h}{K} - \mathbb{E}\left[ \frac{\widehat{\boldsymbol{\Lambda}}_h}{K} \right] \right\| \leq \frac{12\sqrt{2}}{\sqrt{K} \cdot \inf_{s,a} \sigma_h(s, a)^2} \cdot \sqrt{\log\left( \frac{2d}{\delta} \right)}. \tag{F.5}$$

To bound $\|(\widehat{\boldsymbol{\Lambda}}_h/K)^{-1}\|$, we use the fact that

$$
\left\|\left(\frac{\widehat{\boldsymbol{\Lambda}}_h}{K}\right)^{-1}\right\| \le \left\|\mathbb{E}\left[\frac{\widehat{\boldsymbol{\Lambda}}_h}{K}\right]^{-1}\right\| + \left\|\left(\frac{\widehat{\boldsymbol{\Lambda}}_h}{K}\right)^{-1} - \mathbb{E}\left[\frac{\widehat{\boldsymbol{\Lambda}}_h}{K}\right]^{-1}\right\|
$$

$$
\le \left\|\mathbb{E}\left[\frac{\widehat{\boldsymbol{\Lambda}}_h}{K}\right]^{-1}\right\| + \left\|\left(\frac{\widehat{\boldsymbol{\Lambda}}_h}{K}\right)^{-1}\right\| \cdot \left\|\mathbb{E}\left[\frac{\widehat{\boldsymbol{\Lambda}}_h}{K}\right]^{-1}\right\| \cdot \left\|\frac{\widehat{\boldsymbol{\Lambda}}_h}{K} - \mathbb{E}\left[\frac{\widehat{\boldsymbol{\Lambda}}_h}{K}\right]\right\|,
$$

which implies

$$
\left\|\left(\frac{\widehat{\boldsymbol{\Lambda}}_h}{K}\right)^{-1}\right\| \le \left\|\mathbb{E}\left[\frac{\widehat{\boldsymbol{\Lambda}}_h}{K}\right]^{-1}\right\| \cdot \left(1 - \left\|\mathbb{E}\left[\frac{\widehat{\boldsymbol{\Lambda}}_h}{K}\right]^{-1}\right\| \cdot \left\|\frac{\widehat{\boldsymbol{\Lambda}}_h}{K} - \mathbb{E}\left[\frac{\widehat{\boldsymbol{\Lambda}}_h}{K}\right]\right\|\right)^{-1}
$$

$$
\le \left(\left\|\mathbb{E}\left[\frac{\widehat{\boldsymbol{\Lambda}}_h}{K}\right]^{-1}\right\|^{-1} - \left\|\frac{\widehat{\boldsymbol{\Lambda}}_h}{K} - \mathbb{E}\left[\frac{\widehat{\boldsymbol{\Lambda}}_h}{K}\right]\right\|\right)^{-1} \tag{F.6}
$$

Note that by (F.4), we have

$$
\mathbb{E}\left[\frac{\widehat{\boldsymbol{\Lambda}}_h}{K}\right] = \frac{1}{K}\sum_{k=1}^{K}\mathbb{E}\left[\frac{\phi(s_{k,h},a_{k,h})\phi(s_{k,h},a_{k,h})^\top}{\widehat{\sigma}_h(s_{k,h},a_{k,h})^2}\right] + \frac{\lambda}{K}\mathbf{I}_d
$$

$$
\succeq \frac{1}{2K}\sum_{k=1}^{K}\mathbb{E}\left[\frac{\phi(s_{k,h},a_{k,h})\phi(s_{k,h},a_{k,h})^\top}{\sigma_h(s,a)^2}\right] + \frac{\lambda}{K}\mathbf{I}_d
$$

$$
= \frac{1}{2}\boldsymbol{\Lambda}_h + \frac{\lambda}{K}\mathbf{I}_d. \tag{F.7}
$$

Finally combining (F.5), (F.6) and (F.7) yields

$$
\left\|\left(\frac{\widehat{\boldsymbol{\Lambda}}_h}{K}\right)^{-1}\right\| \le \frac{1}{\left(\frac{\iota_h}{2} + \frac{\lambda}{K}\right) - \frac{12\sqrt{2}}{\sqrt{K}\cdot\inf_{s,a}\sigma_h(s,a)^2}\cdot\sqrt{\log\left(\frac{2d}{\delta}\right)}} \le \frac{4}{\iota_h},
$$

where the second inequality follows from (F.3). $\qquad\square$

**Lemma F.4.** For any $h \in [H-1]$, let $\rho$ be some positive constant such that $\rho < (\eta_h + \sigma_r^2)/[12(H-h+1)]$. For any $\delta \in (0,1)$, suppose K satisfies that

$$
K \ge \frac{3600(H-h+1)^4 d^2}{\kappa_h^2 \inf_{s,a}\sigma_h(s,a)^2}\cdot\log\left(\frac{d(H-h+1)KL}{\iota_h\kappa_h\lambda\delta}\right). \tag{F.8}
$$

Then it holds with probability at least $1 - \delta$ that

$$
\left\|\left(\frac{\widehat{\boldsymbol{\Lambda}}_h}{K}\right)^{-1}\right\| \le \frac{8}{\iota_h}.
$$

for all $V \in \mathcal{V}_{h+1}(L) \cap \{V : \sup_{s\in\mathcal{S}}|V(s) - V_{h+1}^\pi(s)| \le \rho\}$.

*Proof of Lemma F.4.* Let $\epsilon > 0$ be a constant to be determined later and $\mathcal{C}_V$ be a $\epsilon-$cover of $\mathcal{V}_{h+1}(L) \cap \{V : \sup_{s\in\mathcal{S}}|V(s) - V_{h+1}^\pi(s)| \le \rho\}$. By Lemma F.3, the choice of K in (F.8) and a union bound, we have

$$
\left\|\left(\frac{\widehat{\boldsymbol{\Lambda}}_h}{K}\right)^{-1}\right\| \le \frac{4}{\iota_h} \tag{F.9}
$$

for all $V \in \mathcal{C}_V$, given K satisfies that

$$
K \ge \frac{3600(H-h+1)^4 d}{\kappa_h^2 \inf_{s,a}\sigma_h(s,a)^2}\cdot\log\left(\frac{Kd\mathcal{N}_\epsilon}{\lambda\delta}\right) \tag{F.10}
$$

where $\mathcal{N}_\epsilon$ is the $\epsilon$−covering number of $\mathcal{V}_{h+1}(L) \cap \{V : \sup_{s \in \mathcal{S}} |V(s) - V_{h+1}^\pi(s)| \le \rho\}$.

For any $V_1 \in \mathcal{V}_{h+1}(L) \cap \{V : \sup_{s \in \mathcal{S}} |V(S) - V_{h+1}^\pi(s)| \le \rho\}$, there exists $V_2 \in \mathcal{C}_V$ such that $\sup_{s \in \mathcal{S}} |V_1(s) - V_2(s)| \le \epsilon$. Let $\sigma_1$ and $\widehat{\mathbf{\Lambda}}_{h,1}$ be the variance estimator and the weighted covariance induced by $V_1$, and $\sigma_2$ and $\widehat{\mathbf{\Lambda}}_{h,2}$ that of $V_2$. Then we have

$$
\begin{aligned}
& \left| \sigma_1^2(s, a) - \sigma_2^2(s, a) \right| \\
\le{}& \left| \langle \phi(s,a), \widehat{\boldsymbol{\beta}}_{h,1} - \widehat{\boldsymbol{\beta}}_{h,2} \rangle \right| + 2(H - h + 1) \left| \langle \phi(s,a), \widehat{\boldsymbol{\theta}}_{h,1} - \widehat{\boldsymbol{\theta}}_{h,2} \rangle \right| \\
\le{}& \left| \widehat{\mathbf{\Sigma}}_h^{-1} \sum_{k=1}^{K} \phi(\check{s}_{k,h}, \check{a}_{k,h})(V_1^2(\check{s}_{k,h}') - V_2^2(\check{s}_{k,h}')) \right| + \left| \widehat{\mathbf{\Sigma}}_h^{-1} \sum_{k=1}^{K} \phi(\check{s}_{k,h}, \check{a}_{k,h})(V_1(\check{s}_{k,h}') - V_2(\check{s}_{k,h}')) \right| \\
\le{}& \frac{4(H - h + 2)^2 K}{\kappa_h} \cdot \epsilon,
\end{aligned}
\tag{F.11}
$$

where the second inequality is due to Assumption 2.1 and the third inequality follows from the fact that $V_1, V_2 \in \mathcal{V}_{h+1}(L)$.

Therefore, we can bound the difference between $\widehat{\mathbf{\Lambda}}_{h,1}$ and $\widehat{\mathbf{\Lambda}}_{h,2}$ as follows.

$$
\begin{aligned}
\left\| \frac{\widehat{\mathbf{\Lambda}}_{h,1}}{K} - \frac{\widehat{\mathbf{\Lambda}}_{h,2}}{K} \right\| &= \left\| \frac{1}{K} \sum_{k=1}^{K} \phi(s_{k,h}, a_{k,h}) \phi(s_{k,h}, a_{k,h})^\top \cdot \frac{\sigma_1(s,a)^2 - \sigma_2(s,a)^2}{\sigma_1(s,a)^2 \sigma_2(s,a)^2} \right\| \\
&\le \frac{1}{K} \sum_{k=1}^{K} \frac{|\sigma_1(s,a)^2 - \sigma_2(s,a)^2|}{\sigma_1(s,a)^2 \sigma_2(s,a)^2} \\
&\le \frac{4(H - h + 2)^2 K}{\kappa_h (\eta_h + \sigma_r^2)^2} \cdot \epsilon,
\end{aligned}
\tag{F.12}
$$

where the first inequality follows from Assumption 2.1, and the second inequality is due to (F.11). When $\epsilon$ is small enough, by (F.9) we have

$$
\lambda_{\min}(\widehat{\mathbf{\Lambda}}_{h,1}/K) \ge \lambda_{\min}(\widehat{\mathbf{\Lambda}}_{h,2}/K) - \|\widehat{\mathbf{\Lambda}}_{h,1} - \widehat{\mathbf{\Lambda}}_{h,2}\|/K \ge \frac{\iota_h}{4} - \frac{4(H - h + 2)^2 K}{\kappa_h(\eta_h + \sigma_r^2)} \cdot \epsilon,
$$

which further implies that

$$
\left\| \left( \frac{\widehat{\mathbf{\Lambda}}_{h,1}}{K} \right)^{-1} \right\| \le \left( \frac{\iota_h}{4} - \frac{4(H - h + 2)^2 K}{\kappa_h(\eta_h + \sigma_r^2)} \cdot \epsilon \right)^{-1} \le \frac{8}{\iota_h}
$$

if we choose $\epsilon = \iota_h \kappa_h (\eta_h + \sigma_r^2)/[32(H - h + 2)^2 K]$. In this case, by Lemma H.13, we have

$$
\log \mathcal{N}_\epsilon \le d \cdot \left( 1 + \frac{64 L(H - h + 2)^2 K}{\iota_h \kappa_h (\eta_h + \sigma_r^2)} \right).
\tag{F.13}
$$

Therefore, by (F.10), (F.12) and (F.13), it suffices to choose $K$ such that

$$
K \ge \frac{3600(H - h + 1)^4 d^2}{\kappa_h^2 \inf_{s,a} \sigma_h(s,a)^2} \cdot \log \left( \frac{d(H - h + 1)KL}{\iota_h \kappa_h \lambda \delta} \right).
$$

$\square$

## F.2 Bernstein Inequality for the Self-Normalized Martingales

**Lemma F.5.** For any $h \in [H - 1]$ and any fixed $\widehat{V}_{h+1}^\pi \in \mathcal{V}_{h+1}(L)$, let $\widehat{\sigma}_h$ be as defined in Line 6 of Algorithm 2 and $\widehat{\mathbf{\Lambda}}_h$ be as defined in (3.3). Suppose $K$ satisfies that

$$
K \ge \frac{1600(H - h + 1)^4 d}{\kappa_h^2 (\eta_h + \sigma_r^2)^2} \cdot \log \left( \frac{K}{\lambda \delta} \right)
\tag{F.14}
$$

Then for any $\delta \in (0, 1)$, it holds with probability at least $1 - \delta$ that

$$
\left\| \sum_{k=1}^{K} \widehat{\sigma}_h(s_{k,h}, a_{k,h})^{-2} \phi(s_{k,h}, a_{k,h}) \left( \mathbb{P}_h \widehat{V}_{h+1}^{\pi}(s_{k,h}, a_{k,h}) - \widehat{V}_{h+1}^{\pi}(s'_{k,h}) - \epsilon_{k,h} \right) \right\|_{\widehat{\mathbf{\Lambda}}_h^{-1}}
$$
$$
\leq \sqrt{2d \log \left( 1 + \frac{K}{\lambda d(\eta_h + \sigma_r^2)} \right) \cdot \log \left( \frac{4K^2}{\delta} \right)} + \frac{4(2H - 2h + 3)}{\sqrt{\eta_h + \sigma_r^2}} \log \left( \frac{4K^2}{\delta} \right)
$$

*Proof of Lemma F.5.* Let $\breve{\mathcal{E}}$ be the event given by Lemma F.2, on which it holds for all $(s, a) \in \mathcal{S} \times \mathcal{A}$ that

$$
\left| \widehat{\sigma}_h^2(s, a) - \sigma_r^2 - \max\{\eta_h, \mathbb{V}_h \widehat{V}_{h+1}^{\pi}(s, a)\} \right| \leq \frac{20(H - h + 1)^2 \sqrt{d}}{\kappa_h \sqrt{K}} \cdot \sqrt{\log \left( \frac{K}{\lambda\delta} \right)}. \tag{F.15}
$$

Now conditioning on $\breve{\mathcal{E}}$, it will not affect the distribution of $\{(s_{k,h}, a_{k,h}), k \in [K]\}$ by independence. In the following argument, we omit the explicit notation for conditioning on $\breve{\mathcal{E}}$ for simplicity.

Define $\mathbf{x}_k = \phi(s_{k,h}, a_{k,h})/\widehat{\sigma}_h(s_{k,h}, a_{k,h})$, which is a deterministic function of $(s_{k,h}, a_{k,h})$ since $\widehat{\boldsymbol{\theta}}_h$ and $\widehat{\boldsymbol{\beta}}_h$ are fixed. Define $\zeta_k = \left( \mathbb{P}_h \widehat{V}_{h+1}^{\pi}(s_{k,h}, a_{k,h}) - \widehat{V}_{h+1}^{\pi}(s'_{k,h}) - \epsilon_{k,h} \right)/\widehat{\sigma}_{k,h}$, which is a function of $s_{k,h}, a_{k,h}, \epsilon_{k,h}, s'_{k,h}$. Now we define the filtration $\{\mathcal{F}_k\}_{k=0}^{K}$ by $\mathcal{F}_0 = \sigma(s_{1,h}, a_{1,h})$, $\mathcal{F}_1 = \sigma(s_{1,h}, a_{1,h}, \epsilon_{1,h}, s'_{1,h}, s_{2,h}, a_{2,h})$, $\cdots$, $\mathcal{F}_k = \sigma(s_{1,h}, a_{1,h}, \epsilon_{1,h}, s'_{1,h}, \cdots, s_{k,h}, a_{k,h}, \epsilon_{k,h}, s'_{k,h}, s_{k+1,h}, a_{k+1,h})$ for $k = 1, \cdots, K - 1$, and $\mathcal{F}_K = \sigma(\mathcal{F}_{K-1}, \epsilon_{K,h}, s'_{K,h})$. Then we see that $\mathbf{x}_k$ is $\mathcal{F}_{k-1}$-measurable and $\zeta_k$ is $\mathcal{F}_k$-measurable. Furthermore, since $\mathbb{E}[\widehat{V}_{h+1}^{\pi}(s'_{k,h}) \mid \mathcal{F}_{k-1}] = \mathbb{P}_h \widehat{V}_{h+1}^{\pi}(s_{k,h}, a_{k,h})$, $\mathbb{E}[\epsilon_{k,h} \mid \mathcal{F}_{k-1}] = 0$ and $\widehat{\sigma}_{k,h}$ is $\mathcal{F}_{k-1}$-measurable, $\zeta_k \mid \mathcal{F}_{k-1}$ has zero-mean. Also, by construction we have $|\zeta_k| \leq (2H - 2h + 3)/\sqrt{\eta_h + \sigma_r^2}$, and it follows from (F.15) that

$$
\mathrm{Var}(\zeta_k \mid \mathcal{F}_{k-1}) \leq \frac{\mathbb{V}_h \widehat{V}_{h+1}^{\pi}(s_{k,h}, a_{k,h}) + \sigma_r^2}{\max\{\eta_h, \mathbb{V}_h \widehat{V}_{h+1}^{\pi}(s_{k,h}, a_{k,h})\} + \sigma_r^2 - \frac{20(H-h+1)^2\sqrt{d}}{\kappa_h\sqrt{K}} \cdot \sqrt{\log\left(\frac{K}{\lambda\delta}\right)}} \leq 2,
$$

as long as $K$ satisfies (F.14).

Then by Theorem H.10, with probability at least $1 - \delta$, we have

$$
\left\| \sum_{k=1}^{K} \mathbf{x}_k \zeta_k \right\|_{\widehat{\mathbf{\Lambda}}_h^{-1}} \leq \sqrt{2d \log \left( 1 + \frac{K}{\lambda d(\eta_h + \sigma_r^2)} \right) \cdot \log \left( \frac{4K^2}{\delta} \right)} + \frac{4(2H - 2h + 3)}{\sqrt{\eta_h + \sigma_r^2}} \log \left( \frac{4K^2}{\delta} \right).
$$

Since $\mathbb{P}(\breve{\mathcal{E}}) \geq 1 - \delta$, the overall probability is at least $(1 - \delta)^2 \geq 1 - 2\delta$ by independence. Finally replacing $\delta$ by $\delta/2$ completes the proof. $\square$

**Lemma F.6.** Let $\epsilon > 0$ be a constant. For any $h \in [H]$ and $\delta \in (0, 1)$, suppose $K$ satisfies that

$$
K \geq \frac{1600(H - h + 1)^4 d^2}{\kappa_h^2(\eta_h + \sigma_r^2)^2} \cdot \log \left( \frac{(H - h + 1)^2 KL}{\lambda\kappa_h(\eta_h + \sigma_r^2)\delta} \right) \tag{F.16}
$$

where $\mathcal{N}_\epsilon$ is the $\epsilon$-covering number of $\mathcal{V}_{h+1}(L)$. Then with probability at least $1 - \delta$, it holds for all function $V \in \mathcal{V}_{h+1}(L)$ that

$$
\left\| \sum_{k=1}^{K} \widehat{\sigma}_h(s_{k,h}, a_{k,h})^{-2} \phi(s_{k,h}, a_{k,h}) \left( \mathbb{P}_h V(s_{k,h}, a_{k,h}) - V(s'_{k,h}) - \epsilon_{k,h} \right) \right\|_{\widehat{\mathbf{\Lambda}}_h^{-1}}^2
$$
$$
\leq 50 \left( d + \frac{\sqrt{d}(H - h + 1)}{\sqrt{\eta_h + \sigma_r^2}} \right)^2 \cdot \log^2 \left( \frac{K(H - h + 1)^2 L}{\kappa_h(\eta_h + \sigma_r^2)\delta} \right). \tag{F.17}
$$

*Proof of Lemma F.6.* For the simplicity of presentation, we first define some notations. We define the following vector,

$$\mathbf{v}_V := \sum_{k=1}^{K} \frac{\boldsymbol{\phi}(s_{k,h}, a_{k,h}) \left( \mathbb{P}_h V(s_{k,h}, a_{k,h}) - V(s'_{k,h}) - \epsilon_{k,h} \right)}{\widehat{\sigma}_h(s_{k,h}, a_{k,h})^2}.$$

and the following matrix

$$\boldsymbol{\Gamma}_V := \sum_{k=1}^{K} \boldsymbol{\phi}(s_{k,h}, a_{k,h}) \boldsymbol{\phi}(s_{k,h}, a_{k,h})^\top / \widehat{\sigma}_h^2(s_{k,h}, a_{k,h}) + \lambda \mathbf{I}_d.$$

It remains to show that, with probability at least $1 - \delta$, for any function $V \in \mathcal{V}_{h+1}(L)$, $\mathbf{v}_V^\top \boldsymbol{\Gamma}_V^{-1} \mathbf{v}_V$ is no greater than the R.H.S. of (F.17). In the following argument, for $V_1 \in \mathcal{V}_{h+1}(L)$, we denote $\mathbf{v}_1 = \mathbf{v}_{V_1}, \boldsymbol{\Gamma}_1 = \boldsymbol{\Gamma}_{V_1}$ and $\sigma_1$ the variance estimator induced by $V_1$, and similar for $V_2$.

Let $\mathcal{C}_V$ be the smallest $\epsilon$-cover of $\mathcal{V}_{h+1}(L)$, and $\mathcal{N}_\epsilon = |\mathcal{C}_V|$ the $\epsilon$-covering number of $\mathcal{V}_{h+1}(L)$. For any $V_1 \in \mathcal{V}_{h+1}(L)$, there exists $V_2 \in \mathcal{C}_V$ such that $\mathrm{dist}(V_1, V_2) = \sup_s |V_1(s) - V_2(s)| \le \epsilon$. Note that we have the following decomposition:

$$\mathbf{v}_1^\top \mathbf{A}_1^{-1} \mathbf{v}_1 \le \mathbf{v}_2^\top \mathbf{A}_2^{-1} \mathbf{v}_2 + \left| \mathbf{v}_1^\top \mathbf{A}_1^{-1} \mathbf{v}_1 - \mathbf{v}_2^\top \mathbf{A}_2^{-1} \mathbf{v}_2 \right|. \tag{F.18}$$

By Lemma H.13, when $K$ satisfies (F.16), we have

$$K \ge \frac{1600(H - h + 1)^4 d}{\kappa_h^2 (\eta_h + \sigma_r^2)^2} \cdot \log\left( \frac{K \mathcal{N}_\epsilon}{\lambda \delta} \right)$$

Then by Lemma F.5 and a union bound, we have that, with probability at least $1 - \delta$,

$$\mathbf{v}_2^\top \mathbf{A}_2^{-1} \mathbf{v}_2 \le \left( \sqrt{2d \log\left( 1 + \frac{K}{\lambda d (\eta_h + \sigma_r^2)} \right) \cdot \log\left( \frac{4K^2 \mathcal{N}_\epsilon}{\delta} \right)} + \frac{4(2H - 2h + 3)}{\sqrt{\eta_h + \sigma_r^2}} \log\left( \frac{4K^2 \mathcal{N}_\epsilon}{\delta} \right) \right)^2. \tag{F.19}$$

It remains to bound the second term in (F.18). We first bound $\|\mathbf{v}_1 - \mathbf{v}_2\|_2$.

$$\|\mathbf{v}_1 - \mathbf{v}_2\|$$
$$= \left\| \sum_{k=1}^{K} \boldsymbol{\phi}(s_{k,h}, a_{k,h}) \left( \frac{\mathbb{P}_h V_1(s_{k,h}, a_{k,h}) - V_1(s'_{k,h}) - \epsilon_{k,h}}{\sigma_1^2(s_{k,h}, a_{k,h})} - \frac{\mathbb{P}_h V_2(s_{k,h}, a_{k,h}) - V_2(s'_{k,h}) - \epsilon_{k,h}}{\sigma_2^2(s_{k,h}, a_{k,h})} \right) \right\|$$
$$\le \sum_{k=1}^{K} \|\boldsymbol{\phi}(s_{k,h}, a_{k,h})\|_2 \cdot \left| \frac{\mathbb{P}_h V_1(s_{k,h}, a_{k,h}) - V_1(s'_{k,h}) - \epsilon_{k,h}}{\sigma_1^2(s_{k,h}, a_{k,h})} - \frac{\mathbb{P}_h V_2(s_{k,h}, a_{k,h}) - V_2(s'_{k,h}) - \epsilon_{k,h}}{\sigma_2^2(s_{k,h}, a_{k,h})} \right|$$
$$\le \sum_{k=1}^{K} \left| \frac{\left( \mathbb{P}_h V_1(s_{k,h}, a_{k,h}) - V_1(s'_{k,h}) - \epsilon_{k,h} \right)}{\sigma_1^2(s_{k,h}, a_{k,h})} - \frac{\left( \mathbb{P}_h V_2(s_{k,h}, a_{k,h}) - V_2(s'_{k,h}) - \epsilon_{k,h} \right)}{\sigma_2^2(s_{k,h}, a_{k,h})} \right| \tag{F.20}$$

where the first inequality follows from Cauchy-Schwartz inequality and the second inequality is due to Assumption 2.1.

Note that for any real-valued function $f_1(\cdot), f_2(\cdot)$ and positive function $g_1(\cdot), g_2(\cdot)$ bounded away from 0, we have

$$\left| \frac{f_1}{g_1} - \frac{f_2}{g_2} \right| = \left| \frac{f_1 g_2 - f_1 g_1 + f_1 g_1 - g_1 f_2}{g_1 g_2} \right|$$
$$\le \left| \frac{f_1(g_2 - g_1)}{g_1 g_2} \right| + \left| \frac{g_1(f_1 - f_2)}{g_1 g_2} \right|$$
$$\le \frac{1}{\inf g_1 \inf g_2} \left[ (\sup |f_1|) \cdot |g_2 - g_1| + (\sup g_1) \cdot |f_1 - f_2| \right]. \tag{F.21}$$

Now, by the construction we have $\sigma_1^2(\cdot, \cdot) \in [\eta_h + \sigma_r^2, (H - h + 1)^2 + \sigma_r^2]$, and $\mathbb{P}_h V_1(\cdot, \cdot) - V_1(\cdot) - \epsilon_{k,h} \in [-2H + 2h - 3, 2H - 2h + 3]$, and the same for $\sigma_2$ and $V_2$. Also note that for all $(s, a)$,

$$\left| \sigma_1^2(s, a) - \sigma_2^2(s, a) \right|$$

$$\leq \left| \langle \phi(s, a), \widehat{\boldsymbol{\beta}}_{h,1} - \widehat{\boldsymbol{\beta}}_{h,2} \rangle \right| + 2(H - h + 1) \left| \langle \phi(s, a), \widehat{\boldsymbol{\theta}}_{h,1} - \widehat{\boldsymbol{\theta}}_{h,2} \rangle \right|$$

$$\leq \left| \widehat{\boldsymbol{\Sigma}}_h^{-1} \sum_{k=1}^{K} \phi(\check{s}_{k,h}, \check{a}_{k,h})(V_1^2(\check{s}'_{k,h}) - V_2^2(\check{s}'_{k,h})) \right| + \left| \widehat{\boldsymbol{\Sigma}}_h^{-1} \sum_{k=1}^{K} \phi(\check{s}_{k,h}, \check{a}_{k,h})(V_1(\check{s}'_{k,h}) - V_2(\check{s}'_{k,h})) \right|$$

$$\leq \frac{4(H - h + 2)^2 K}{\kappa_h} \cdot \epsilon, \tag{F.22}$$

where the second inequality is due to Assumption 2.1 and the third inequality follows from the fact that $V_1, V_2 \in \mathcal{V}_{h+1}(L)$.

Denote $\mathbf{u} = \mathbf{v}_2 - \mathbf{v}_1$. Combining (F.20), (F.21) and (F.22) yields that

$$\|\mathbf{u}\| \leq \sum_{k=1}^{K} \frac{1}{(\eta_h + \sigma_r^2)^2} \left[ \frac{4(2H - 2h + 3)(H - h + 2)^2 K}{\kappa_h} \cdot \epsilon + ((H - h + 1)^2 + \sigma_r^2) \cdot 2\epsilon \right]$$

$$\leq \frac{10(H - h + 1)^2 K^2}{\kappa_h (\eta_h + \sigma_r^2)^2} \epsilon, \tag{F.23}$$

where the second inequality is due to the fact that $\sigma_r^2 \leq 1$.

Next, by Lemma H.7 and (F.22), we have

$$\|\mathbf{A}_1^{-1} - \mathbf{A}_2^{-1}\| \leq \frac{8K^2((H - h + 1)^2 + \sigma_r^2)}{\lambda^2 \kappa_h (\eta_h + \sigma_r^2)^2} \cdot \epsilon. \tag{F.24}$$

Also note that

$$\left| \mathbf{v}_1^\top \mathbf{A}_1^{-1} \mathbf{v}_1 - \mathbf{v}_2^\top \mathbf{A}_2^{-1} \mathbf{v}_2 \right| = \left| \mathbf{v}_1^\top \mathbf{A}_1^{-1} \mathbf{v}_1 - (\mathbf{v}_1 + \mathbf{u})^\top \mathbf{A}_2^{-1} (\mathbf{v}_1 + \mathbf{u}) \right|$$

$$\leq \left| \mathbf{v}_1^\top (\mathbf{A}_1^{-1} - \mathbf{A}_2^{-1}) \mathbf{v}_1 \right| + 2 \left| \mathbf{v}_1^\top \mathbf{A}_2^{-1} \mathbf{u} \right| + \left| \mathbf{u}^\top \mathbf{A}_2^{-1} \mathbf{u} \right|. \tag{F.25}$$

By the definition, we have $\|\mathbf{v}_1\|_2, \|\mathbf{v}_2\|_2 \leq (2H - 2h + 3)K/(\eta_h + \sigma_r^2)$, and $\|\mathbf{A}_1^{-1}\|, \|\mathbf{A}_2^{-1}\| \leq 1/\lambda$. It then follows from (F.24) and (F.25) that

$$\left| \mathbf{v}_1^\top \mathbf{A}_1^{-1} \mathbf{v}_1 - \mathbf{v}_2^\top \mathbf{A}_2^{-1} \mathbf{v}_2 \right| \leq \frac{(2H - 2h + 3)^2 K^2}{(\eta_h + \sigma_r^2)^2} \cdot \frac{8K^2((H - h + 1)^2 + \sigma_r^2)}{\lambda^2 \kappa_h (\eta_h + \sigma_r^2)^2} \cdot \epsilon$$

$$+ \frac{2(2H - 2h + 3)K}{\lambda(\eta_h + \sigma_r^2)} \cdot \frac{10K(H - h + 1)^2 K^2}{\kappa_h (\eta_h + \sigma_r^2)^2} \cdot \epsilon$$

$$+ \frac{100(H - h + 1)^4 K^4}{\lambda \kappa_h^2 (\eta_h + \sigma_r^2)^4} \cdot \epsilon^2$$

$$\leq \frac{200(H - h + 1)^4 K^4}{\kappa_h^2 (\eta_h + \sigma_r^2)^2} \cdot \epsilon,$$

by the choice of $\lambda = 1$. We then choose $\epsilon = \kappa_h^2 (\eta_h + \sigma_r^2)^2 / [200(H - h + 1)^4 K^5]$, and thus

$$\left| \mathbf{v}_1^\top \mathbf{A}_1^{-1} \mathbf{v}_1 - \mathbf{v}_2^\top \mathbf{A}_2^{-1} \mathbf{v}_2 \right| \leq \frac{1}{K} \tag{F.26}$$

Now by Lemma (H.13), we have

$$\mathcal{N}_\epsilon \leq \left( 1 + \frac{400(H - h + 1)^4 K^5 L}{\kappa_h^2 (\eta_h + \sigma_r^2)^2} \right)^d. \tag{F.27}$$

Then combining (F.18), (F.19), (F.26) and (F.27) yields

$$\left\| \sum_{k=1}^{K} \widehat{\sigma}_h(s_{k,h}, a_{k,h})^{-2} \phi(s_{k,h}, a_{k,h}) \left( \mathbb{P}_h V(s_{k,h}, a_{k,h}) - V(s'_{k,h}) - \epsilon_{k,h} \right) \right\|_{\widehat{\boldsymbol{\Lambda}}_h^{-1}}^2$$

$$\leq 50 \left( d + \frac{d(H - h + 1)}{\sqrt{\eta_h + \sigma_r^2}} \right)^2 \cdot \log^2 \left( \frac{K(H - h + 1)^2 L}{\kappa_h (\eta_h + \sigma_r^2) \delta} \right).$$

This completes the proof. □

## F.3 Bounding the error terms

Finally, we prove the following key lemma for completing the induction step in the proof of Theorem C.2.

**Lemma F.7.** Set $L = (1 + 1/H)d\sqrt{K/\lambda}$. For any $h \in [H-1]$, let $\rho$ be some positive constant such that $\rho < (\eta_h + \sigma_r^2)/[12(H - h + 1)]$. For any $\delta \in (0, 1)$, suppose $K$ satisfies that

$$K \geq \frac{3600(H - h + 1)^4 d^2}{\kappa_h^2 (\eta_h + \sigma_r^2)^2} \cdot \log\left(\frac{dHK}{\kappa_h \delta}\right) \tag{F.28}$$

Then the following two events hold simultaneously with probability at least $1 - \delta$:

1. $\widetilde{\mathcal{E}}_1$: for all $V \in \mathcal{V}_{h+1}(L) \cap \{V : \sup_{s \in \mathcal{S}} |V(s) - V_{h+1}^\pi(s)| \leq \rho\}$,

$$\left\| \left(\frac{\widehat{\boldsymbol{\Lambda}}_h}{K}\right)^{-1} \right\| \leq \frac{8}{\iota_h}; \tag{F.29}$$

2. $\widetilde{\mathcal{E}}_2$: for all function $V(\cdot) \in \mathcal{V}_{h+1}(L) \cap \{V : \sup_{s \in \mathcal{S}} |V(s) - V_{h+1}^\pi(s)| \leq \rho\}$ and all $(s, a)$ pairs,

$$\left| \phi(s, a)^\top \widehat{\boldsymbol{\Lambda}}_h^{-1} \sum_{k=1}^K \widehat{\sigma}_h(s_{k,h}, a_{k,h})^{-2} \phi(s_{k,h}, a_{k,h}) \left(\mathbb{P}_h V(s_{k,h}, a_{k,h}) - V(s'_{k,h}) - \epsilon_{k,h}\right) \right|$$

$$\leq \frac{20}{\sqrt{K}} \cdot \left(\frac{d}{\sqrt{\iota_h}} + \frac{d(H - h + 1)}{\sqrt{\iota_h(\eta_h + \sigma_r^2)}}\right) \cdot \log\left(\frac{d(H - h + 1)^2 K}{\kappa_h(\eta_h + \sigma_r^2)\delta}\right)$$

*Proof of Lemma F.7.* We want to show $\mathbb{P}\{\widetilde{\mathcal{E}}_1 \cap \widetilde{\mathcal{E}}_2\} \geq 1 - \delta$. It follows from Lemma F.4 and (F.28) that $\mathbb{P}(\widetilde{\mathcal{E}}_1) \geq 1 - \delta$.

To show that $\mathbb{P}(\widetilde{\mathcal{E}}_2) \geq 1 - \delta$, first by Lemma F.6, we have

$$\left\| \sum_{k=1}^K \widehat{\sigma}_h(s_{k,h}, a_{k,h})^{-2} \phi(s_{k,h}, a_{k,h}) \left(\mathbb{P}_h V(s_{k,h}, a_{k,h}) - V(s'_{k,h}) - \epsilon_{k,h}\right) \right\|_{\widehat{\boldsymbol{\Lambda}}_h^{-1}}^2$$

$$\leq 50 \left(d + \frac{d(H - h + 1)}{\sqrt{\eta_h + \sigma_r^2}}\right)^2 \cdot \log^2\left(\frac{K(H - h + 1)^2 L}{\kappa_h(\eta_h + \sigma_r^2)\delta}\right), \tag{F.30}$$

for all $V \in \mathcal{V}_{h+1}(L)$.

It follows from Cauchy-Schwartz inequality that

$$\phi(s, a)^\top \widehat{\boldsymbol{\Lambda}}_h^{-1} \sum_{k=1}^K \widehat{\sigma}_h(s_{k,h}, a_{k,h})^{-2} \phi(s_{k,h}, a_{k,h}) \left(\mathbb{P}_h V(s_{k,h}, a_{k,h}) - V(s'_{k,h}) - \epsilon_{k,h}\right)$$

$$\leq \|\phi(s, a)\|_{\widehat{\boldsymbol{\Lambda}}_h^{-1}} \cdot \left\| \sum_{k=1}^K \widehat{\sigma}_h(s_{k,h}, a_{k,h})^{-2} \phi(s_{k,h}, a_{k,h}) \left([\mathbb{P}_h V](s_{k,h}, a_{k,h}) - V(s'_{k,h}) - \epsilon_{k,h}\right) \right\|_{\widehat{\boldsymbol{\Lambda}}_h^{-1}}$$

$$\leq \|\widehat{\boldsymbol{\Lambda}}_h^{-1}\|^{1/2} \cdot \left\| \sum_{k=1}^K \widehat{\sigma}_h(s_{k,h}, a_{k,h})^{-2} \phi(s_{k,h}, a_{k,h}) \left([\mathbb{P}_h V](s_{k,h}, a_{k,h}) - V(s'_{k,h}) - \epsilon_{k,h}\right) \right\|_{\widehat{\boldsymbol{\Lambda}}_h^{-1}}$$

$$\leq \frac{20}{\sqrt{K}} \cdot \left(\frac{d}{\sqrt{\iota_h}} + \frac{d(H - h + 1)}{\sqrt{\iota_h(\eta_h + \sigma_r^2)}}\right) \cdot \log\left(\frac{d(H - h + 1)^2 K}{\kappa_h(\eta_h + \sigma_r^2)\delta}\right),$$

where the second inequality follows from Assumption 2.1 and the third inequality follows from (F.29) and (F.30). Note that this holds for all $(s, a) \in \mathcal{S} \times \mathcal{A}$ as we directly bound the operator norm of $\widehat{\boldsymbol{\Lambda}}_h$. Replacing $\delta$ by $\delta/2$ completes the proof. □

# G Lemmas for OPE Convergence

## G.1 Concentration of $\widehat{\sigma}$

Recall that in the algorithm, to estimate the variance, we use $\widehat{\beta}_h$ and $\widehat{\theta}_h$ which are estimated using the function $\widehat{V}_{h+1}^\pi$ and $\{\check{s}_{k,h}, \check{a}_{k,h}, \check{s}'_{k,h}\}_{k \in [K]}$.

For the next lemma we denote the function $\sigma(\cdot, \cdot)$ as computed from some function $V(\cdot)$ and data $\check{\mathcal{D}}_h$.

**Lemma G.1.** Let $\rho \geq 0$. For any $V \in \mathcal{V}_{h+1}(L) \cap \{V : \sup_s |V(s) - V_{h+1}^\pi(s)| \leq \rho\}$, with probability at least $1 - \delta$, we have

$$\left| \sigma^2(s,a) - \sigma_r^2 - \max\left\{ \eta_h, \ \mathbb{V}_h V_{h+1}^\pi(s,a) \right\} \right| \leq \frac{C_{K,h,\delta}(H - h + 1)^2 \sqrt{d}}{\sqrt{K}} + 4(H - h + 1) \cdot \rho,$$

for all $(s,a) \in \mathcal{S} \times \mathcal{A}$ where

$$C_{K,h,\delta} = 12\sqrt{2} \cdot \frac{1}{\sqrt{\kappa_h}} \cdot \left[ \frac{1}{2} \log\left( \frac{\lambda + K}{\lambda} \right) + \frac{1}{d} \log \frac{4}{\delta} \right]^{1/2} + 12\lambda \cdot \frac{1}{\kappa_h}.$$

*Proof of Lemma G.1.* By Lemma F.2, with probability at least $1 - \delta$, we have

$$\left| \sigma^2(s,a) - \sigma_r^2 - \max\left\{ \eta_h, \ \mathbb{V}_h V(s,a) \right\} \right| \leq \frac{C_{K,h,\delta}(H - h + 1)^2 \sqrt{d}}{\sqrt{K}}.$$

Note that if two functions $f_1, f_2 : \mathcal{S} \to \mathbb{R}$ satisfies $\sup_s |f_1(s) - f_2(s)| \leq \rho$, $\sup_s |f_1(s)| \leq H - h + 1$, and $\sup_s |f_2(s)| \leq H - h + 1$, then for all $(s,a)$,

$$|\mathbb{V}_h f_1(s,a) - \mathbb{V}_h f_2(s,a)| \leq 4(H - h + 1) \cdot \rho.$$

Then using the triangular inequality and $|\mathbb{V}_h V(s,a) - \mathbb{V}_h V_{h+1}^\pi(s,a)| \leq 4(H - h + 1)\rho$ finishes the proof. $\qquad\square$

## G.2 Concentration of Weighted Sample Covariance Matrices

In this subsection, we study the concentration of the matrices $\widehat{\Lambda}_h$, $h \in [H]$ to their population counterparts. Recall from Algorithm 1 that for each $h \in [H]$, the matrix $\widehat{\Lambda}_h$ is generated using the function $\widehat{\sigma}_h(\cdot, \cdot)$ and the dataset $\mathcal{D}_h = \{(s_{k,h}, a_{k,h}, r_{k,h}, s'_{k,h})\}_{k \in [K]}$. Since the function $\widehat{\sigma}_h(\cdot, \cdot)$ itself is generated by $\widehat{V}_{h+1}(\cdot)$ and the dataset $\check{\mathcal{D}}_h = \{(\check{s}_{k,h}, \check{a}_{k,h}, \check{r}_{k,h}, \check{s}'_{k,h})\}_{k \in [K]}$, we can equivalently view $\widehat{\Lambda}_h$ as generated by $\widehat{V}_{h+1}(\cdot)$ and the datasets $\check{\mathcal{D}}_h$ and $\mathcal{D}_h$. In the remaining of the subsection, we will omit the subscript and superscript when it is clear and simply write

$$\widehat{\Lambda}_h = \sum_{k=1}^K \phi(s_{k,h}, a_{k,h}) \phi^\top(s_{k,h}, a_{k,h}) / \sigma^2(s_{k,h}, a_{k,h}) + \lambda \mathbf{I}_d,$$

where $\sigma(\cdot, \cdot)$ is generated using the function $V(\cdot)$ and the dataset $\check{\mathcal{D}}_h$ as described in Algorithm 1. We also denote

$$\sigma_V^2(\cdot, \cdot) := \max\left\{ \eta_h, \ \mathbb{V}_h V(\cdot, \cdot) \right\} + \sigma_r^2.$$

By Lemma F.2, we know that with high probability, $\sigma^2(\cdot, \cdot)$ will be a good estimator for $\sigma_V^2(\cdot, \cdot)$. This will be used to show the concentration of the matrix $\widehat{\Lambda}_h$. We start from the next lemma.

**Lemma G.2.** For any $h \in [H]$, conditioning on $\sigma(\cdot, \cdot) \in \mathcal{T}_h(L_1, L_2)$ being fixed, with conditional probability at least $1 - \delta$,

$$\left\| \frac{\widehat{\Lambda}_h}{K} - \mathbb{E}_{\bar{\pi}, h}\left[ \frac{\phi(s,a)\phi(s,a)^\top}{\sigma^2(s,a)} \right] \right\| \leq \frac{4\sqrt{2}}{(\eta_h + \sigma_r^2)\sqrt{K}} \cdot \left( \log \frac{2d}{\delta} \right)^{1/2} + \frac{\lambda}{K}.$$

*Proof of Lemma G.2.* Since $\sigma(\cdot,\cdot)$ is a function of $V$ and the dataset $\check{\mathcal{D}}_h$ which is independent of $\mathcal{D}_h$, conditioning on $\sigma(\cdot,\cdot)$ won't change the distribution of $\mathcal{D}_h$. In other words, $\phi(s_{k,h},a_{k,h})/\sigma(s_{k,h},a_{k,h})$, $k \in [K]$ can be viewed as independent random vectors. Then by Lemma H.4, we have that, with conditional probability at least $1-\delta$,

$$\left\| \frac{\widehat{\boldsymbol{\Lambda}}_h}{K} - \mathbb{E}_{\bar{\pi},h}\left[\frac{\widehat{\boldsymbol{\Lambda}}_h}{K}\right] \right\| \le \frac{4\sqrt{2}}{(\eta_h + \sigma_r^2)\sqrt{K}} \cdot \sqrt{\log\left(\frac{2d}{\delta}\right)},$$

and thus

$$\left\| \frac{\widehat{\boldsymbol{\Lambda}}_h}{K} - \mathbb{E}_{\bar{\pi},h}\left[\frac{\phi(s,a)\phi(s,a)^\top}{\sigma^2(s,a)}\right] \right\| \le \left\| \frac{\widehat{\boldsymbol{\Lambda}}_h}{K} - \mathbb{E}_{\bar{\pi},h}\left[\frac{\widehat{\boldsymbol{\Lambda}}_h}{K}\right] \right\| + \left\| \mathbb{E}_{\bar{\pi},h}\left[\frac{\phi(s,a)\phi(s,a)^\top}{\sigma^2(s,a)}\right] - \mathbb{E}_{\bar{\pi},h}\left[\frac{\widehat{\boldsymbol{\Lambda}}_h}{K}\right] \right\|$$

$$\le \frac{4\sqrt{2}}{(\eta_h + \sigma_r^2)\sqrt{K}} \cdot \sqrt{\log\left(\frac{2d}{\delta}\right)} + \frac{\lambda}{K}.$$

$\square$

Next, combine Lemma G.2 and the event that $\sigma^2(\cdot,\cdot)$ is a good estimator for $\sigma_V^2(\cdot,\cdot)$, we get the following lemma.

**Lemma G.3.** For any $h \in [H]$, condition on $V \in \mathcal{V}_{h+1}(L)$ being fixed, with conditional probability at least $1-\delta$,

$$\left\| \frac{\widehat{\boldsymbol{\Lambda}}_h}{K} - \mathbb{E}_{\bar{\pi},h}\left[\frac{\phi(s,a)\phi(s,a)^\top}{\sigma_V^2(s,a)}\right] \right\|$$

$$\le \frac{4\sqrt{2}}{(\eta_h + \sigma_r^2)\sqrt{K}} \cdot \left(\log\frac{4d}{\delta}\right)^{1/2} + \frac{\lambda}{K} + \frac{1}{(\eta_h + \sigma_r^2)^2} \cdot \frac{C_{K,h,\delta}(H-h+1)^2\sqrt{d}}{\sqrt{K}},$$

where

$$C_{K,h,\delta} = 12\sqrt{2} \cdot \frac{1}{\sqrt{\kappa_h}} \cdot \left[\frac{1}{2}\log\left(\frac{\lambda+K}{\lambda}\right) + \frac{1}{d}\log\frac{8}{\delta}\right]^{1/2} + 12\lambda \cdot \frac{1}{\kappa_h}.$$

*Proof of Lemma G.3.* First note that condition on $\sigma(\cdot,\cdot) \in \mathcal{T}_h(L_1, L_2)$ such that $\sup_{s,a}|\sigma^2(s,a) - \sigma_V^2(s,a)| \le \rho$ for some $\rho \ge 0$, we have

$$\left\| \mathbb{E}_{\bar{\pi},h}\left[\frac{\phi(s,a)\phi(s,a)^\top}{\sigma^2(s,a)}\right] - \mathbb{E}_{\bar{\pi},h}\left[\frac{\phi(s,a)\phi(s,a)^\top}{\sigma_V^2(s,a)}\right] \right\|$$

$$\le \mathbb{E}_{\bar{\pi},h}\left[\|\phi(s,a)\phi(s,a)^\top\| \sup_{s,a}\left(\frac{1}{\sigma^2(s,a)} - \frac{1}{\sigma_V^2(s,a)}\right)\right]$$

$$\le \frac{1}{(\eta_h + \sigma_r^2)^2} \cdot \rho,$$

since $\sigma^2(s,a)$ and $\sigma_V^2(s,a)$ are lower bounded by $\eta_h + \sigma_r^2$. Then by Lemma G.2, we have that, conditioning on fixed $\sigma(\cdot,\cdot)$ s.t. $\sup_{s,a}|\sigma^2(s,a) - \sigma_V^2(s,a)| \le \rho$, with conditional probability at least $1-\delta$,

$$\left\| \frac{\widehat{\boldsymbol{\Lambda}}_h}{K} - \mathbb{E}_{\bar{\pi},h}\left[\frac{\phi(s,a)\phi(s,a)^\top}{\sigma_V^2(s,a)}\right] \right\| \le \frac{4\sqrt{2}}{(\eta_h + \sigma_r^2)\sqrt{K}} \cdot \left(\log\frac{2d}{\delta}\right)^{1/2} + \frac{\lambda}{K} + \frac{1}{(\eta_h + \sigma_r^2)^2} \cdot \rho. \tag{G.1}$$

Since conditioning on $V(\cdot)$ won't change the distribution of $\check{\mathcal{D}}_h$ under Assumption C.1, by Lemma F.2, with probability at least $1-\delta$, it holds for all $(s,a) \in \mathcal{S} \times \mathcal{A}$ that

$$\left|\sigma^2(s,a) - \sigma_V^2(s,a)\right| \le \frac{C_{K,h,\delta}(H-h+1)^2\sqrt{d}}{\sqrt{K}}, \tag{G.2}$$

where

$$C_{K,h,\delta} = 12\sqrt{2} \cdot \frac{1}{\sqrt{\kappa_h}} \cdot \left[\frac{1}{2}\log\left(\frac{\lambda+K}{\lambda}\right) + \frac{1}{d}\log\frac{4}{\delta}\right]^{1/2} + 12\lambda \cdot \frac{1}{\kappa_h}.$$

Combine (G.1) and (G.2), and we get that, condition on $V$, with probability at least $1 - 2\delta$,

$$\left\|\frac{\widehat{\mathbf{\Lambda}}_h}{K} - \mathbb{E}_{\bar{\pi},h}\left[\frac{\phi(s,a)\phi(s,a)^\top}{\sigma_V^2(s,a)}\right]\right\|$$

$$\leq \frac{4\sqrt{2}}{(\eta_h+\sigma_r^2)\sqrt{K}} \cdot \left(\log\frac{2d}{\delta}\right)^{1/2} + \frac{\lambda}{K} + \frac{1}{(\eta_h+\sigma_r^2)^2} \cdot \frac{C_{K,h,\delta}(H-h+1)^2\sqrt{d}}{\sqrt{K}}.$$

Replacing $\delta$ with $\delta/2$ finishes the proof. $\qquad\square$

Finally, combining Lemma G.3 and the event of uniform convergence, we can bound the distance between $\widehat{\mathbf{\Lambda}}_h$ and its population counterpart $\mathbf{\Lambda}_h$.

**Lemma G.4.** For any $h \in [H]$, condition on $V \in \mathcal{V}_{h+1}(L) \cap \{V : \sup_s |V(s) - V_{h+1}^\pi(s)| \leq \rho\}$, with conditional probability at least $1 - \delta$, we have

$$\left\|\frac{\widehat{\mathbf{\Lambda}}_h}{K} - \mathbf{\Lambda}_h\right\|$$

$$\leq \frac{4\sqrt{2}}{(\eta_h+\sigma_r^2)\sqrt{K}} \cdot \left(\log\frac{4d}{\delta}\right)^{1/2} + \frac{\lambda}{K} + \frac{1}{(\eta_h+\sigma_r^2)^2} \cdot \left(\frac{C_{K,h,\delta}(H-h+1)^2\sqrt{d}}{\sqrt{K}} + 4(H-h+1)\cdot\rho\right),$$

where

$$C_{K,h,\delta} = 12\sqrt{2} \cdot \frac{1}{\sqrt{\kappa_h}} \cdot \left[\frac{1}{2}\log\left(\frac{\lambda+K}{\lambda}\right) + \frac{1}{d}\log\frac{8}{\delta}\right]^{1/2} + 12\lambda \cdot \frac{1}{\kappa_h}.$$

*Proof of Lemma G.4.* First note that by Lemma G.3, with probability at least $1 - \delta$,

$$\left\|\frac{\widehat{\mathbf{\Lambda}}_h}{K} - \mathbb{E}_{\bar{\pi},h}\left[\frac{\phi(s,a)\phi(s,a)^\top}{\sigma_V^2(s,a)}\right]\right\|$$

$$\leq \frac{4\sqrt{2}}{(\eta_h+\sigma_r^2)\sqrt{K}} \cdot \left(\log\frac{4d}{\delta}\right)^{1/2} + \frac{\lambda}{K} + \frac{1}{(\eta_h+\sigma_r^2)^2} \cdot \frac{C_{K,h,\delta}(H-h+1)^2\sqrt{d}}{\sqrt{K}}.$$

On the other hand, by $\sup_s |V(s) - V_{h+1}^\pi(s)| \leq \rho$ and $|V(s)|, |V_{h+1}^\pi(s)| \leq H - h + 1$, we have $\sup_s |\sigma_V^2(s) - \sigma_h^2(s)| \leq 4(H-h+1)\rho$. It implies that

$$\left\|\mathbb{E}_{\bar{\pi},h}\left[\frac{\phi(s,a)\phi(s,a)^\top}{\sigma_V^2(s,a)}\right] - \mathbb{E}_{\bar{\pi},h}\left[\frac{\phi(s,a)\phi(s,a)^\top}{\sigma_h^2(s,a)}\right]\right\| \leq \frac{1}{(\eta_h+\sigma_r^2)^2} \cdot 4(H-h+1)\cdot\rho.$$

Then by triangular inequality, we conclude that

$$\left\|\frac{\widehat{\mathbf{\Lambda}}_h}{K} - \mathbb{E}_{\bar{\pi},h}\left[\frac{\phi(s,a)\phi(s,a)^\top}{\sigma_h^2(s,a)}\right]\right\|$$

$$\leq \frac{4\sqrt{2}}{(\eta_h+\sigma_r^2)\sqrt{K}} \cdot \left(\log\frac{4d}{\delta}\right)^{1/2} + \frac{\lambda}{K} + \frac{1}{(\eta_h+\sigma_r^2)^2} \cdot \left(\frac{C_{K,h,\delta}(H-h+1)^2\sqrt{d}}{\sqrt{K}} + 4(H-h+1)\cdot\rho\right).$$

Finally, recall the definition of $\mathbf{\Lambda}_h$ given by (2.6). $\qquad\square$

### G.3 Bound for the self-normalized martingales

**Lemma G.5.** For any $h \in [H]$, condition on $\widehat{V}_{h+1}^{\pi} \in \mathcal{V}_{h+1}(L)$ s.t. $\sup_s \left| \widehat{V}_{h+1}^{\pi}(s) \right| \leq B$, with conditional probability at least $1 - \delta$,

$$
\left\| \sum_{k=1}^{K} \phi(\check{s}_{k,h}, \check{a}_{k,h}) \left[ (\widehat{V}_{h+1}^{\pi})(\check{s}'_{k,h}) - \mathbb{P}_h(\widehat{V}_{h+1}^{\pi})(\check{s}_{k,h}, \check{a}_{k,h}) \right] \right\|_{\widehat{\mathbf{\Sigma}}_h^{-1}}^2 \leq 8B^2 \left[ \frac{d}{2} \log \left( \frac{\lambda + K}{\lambda} \right) + \log \frac{2}{\delta} \right],
$$

$$
\left\| \sum_{k=1}^{K} \phi(\check{s}_{k,h}, \check{a}_{k,h}) \left[ (\widehat{V}_{h+1}^{\pi})^2(\check{s}'_{k,h}) - \mathbb{P}_h(\widehat{V}_{h+1}^{\pi})^2(\check{s}_{k,h}, \check{a}_{k,h}) \right] \right\|_{\widehat{\mathbf{\Sigma}}_h^{-1}}^2 \leq 8B^4 \left[ \frac{d}{2} \log \left( \frac{\lambda + K}{\lambda} \right) + \log \frac{2}{\delta} \right].
$$

*Proof of Lemma G.5.* Denote $\mathbf{x}_k = \phi(\check{s}_{k,h}, \check{a}_{k,h})$, and $\eta_k = (\widehat{V}_{h+1}^{\pi})(\check{s}'_{k,h}) - \mathbb{P}_h(\widehat{V}_{h+1}^{\pi})(\check{s}_{k,h}, \check{a}_{k,h})$.

Define the filtration $\{\mathcal{F}_k\}_{k=0}^{K}$ by $\mathcal{F}_0 = \sigma(\check{s}_{1,h}, \check{a}_{1,h})$, $\mathcal{F}_1 = \sigma(\check{s}_{1,h}, \check{a}_{1,h}, \check{s}'_{1,h}, \check{s}_{2,h}, \check{a}_{2,h})$, $\cdots$, $\mathcal{F}_k = \sigma(\check{s}_{1,h}, \check{a}_{1,h}, \check{s}'_{1,h}, \cdots, \check{s}_{k,h}, \check{a}_{k,h}, \check{s}'_{k,h}, \check{s}_{k+1,h}, \check{a}_{k+1,h})$ for $k = 1, \cdots, K-1$, and $\mathcal{F}_K = \sigma(\mathcal{F}_{K-1}, \check{s}'_{K,h})$. Then we see that $\mathbf{x}_k$ is $\mathcal{F}_{k-1}$-measurable, and $\eta_k$ is $\mathcal{F}_k$-measurable. Furthermore, since $\mathbb{E}[(\widehat{V}_{h+1}^{\pi})^2(\check{s}'_{k,h}) \mid \mathcal{F}_{k-1}] = \mathbb{P}_h(\widehat{V}_{h+1}^{\pi})^2(\check{s}_{k,h}, \check{a}_{k,h})$, $\eta_k \mid \mathcal{F}_{k-1}$ is zero-mean. Also, $|\eta_k| \leq 2B$, which implies that $\eta_k \mid \mathcal{F}_{k-1}$ is $2B$-subgaussian. Then by H.9, with probability at least $1 - \delta/2$,

$$
\left\| \sum_{k=1}^{K} \mathbf{x}_k \eta_k \right\|_{\widehat{\mathbf{\Sigma}}_h^{-1}}^2 \leq 8B^2 \log \left( \frac{\det(\widehat{\mathbf{\Sigma}}_h)^{1/2} \det(\lambda I)^{-1/2}}{\delta/2} \right).
$$

Recall that $\widehat{\mathbf{\Sigma}}_h = \sum_{k=1}^{K} \phi(\check{s}_{k,h}, \check{a}_{k,h}) \phi^{\top}(\check{s}_{k,h}, \check{a}_{k,h}) + \lambda \mathbf{I}_d$ where $\|\phi\| \leq 1$. It follows that

$$
\det(\widehat{\mathbf{\Sigma}}_h) \leq (\lambda + K)^d.
$$

We then conclude that

$$
\left\| \sum_{k=1}^{K} \mathbf{x}_k \eta_k \right\|_{\widehat{\mathbf{\Sigma}}_h^{-1}}^2 \leq 8B^2 \left[ \frac{d}{2} \log \left( \frac{\lambda + K}{\lambda} \right) + \log \frac{2}{\delta} \right].
$$

The second inequality is similar. Taking a union bound finishes the proof. $\qquad\square$

## H  Auxiliary Lemmas

### H.1  Concentration Inequalities

**Lemma H.1** (Matrix McDiarmid inequality, Tropp 41). Let $\mathbf{z}_k$, $k = 1, \cdots, K$ be independent random vectors in $\mathbb{R}^d$, and let $\mathbf{H}$ be a function that maps $K$ vectors to a $d \times d$ symmetric matrix. Assume there exists a sequence of fixed symmetric matrices $\{\mathbf{A}_k\}_{k \in [K]}$ such that

$$
\left( \mathbf{H}(\mathbf{z}_1, \cdots, \mathbf{z}_k, \cdots, \mathbf{z}_K) - \mathbf{H}(\mathbf{z}_1, \cdots, \mathbf{z}'_k, \cdots, \mathbf{z}_K) \right)^2 \preceq \mathbf{A}_k^2,
$$

where $\mathbf{z}_k, \mathbf{z}'_k$ ranges over all possible values for each $k \in [K]$. Define $\sigma^2$ as

$$
\sigma^2 := \left\| \sum_k \mathbf{A}_k^2 \right\|.
$$

Then, for any $t > 0$,

$$
\mathbb{P}\left\{ \lambda_{\max}(\mathbf{H}(\mathbf{z}) - \mathbb{E}\mathbf{H}(\mathbf{z})) \geq t \right\} \leq d \cdot \exp\left( \frac{-t^2}{8\sigma^2} \right),
$$

where $\mathbf{z} = (\mathbf{z}_1, \cdots, \mathbf{z}_K)$.

**Lemma H.2** (Freedman's inequality for martingales, Freedman [13]). Consider a martingale difference sequence $\{e_k, \ k = 1, 2, 3, \cdots\}$ with filtration $\mathcal{F}_k := \sigma\{e_1, \cdots, e_{k-1}\}$, for $k = 1, 2, \cdots$. Assume $e_k$ is uniformly bounded:

$$|e_k| \leq R \quad \text{almost surely for} \quad k = 1, 2, 3, \cdots$$

Then for all $\epsilon \geq 0$ and $\sigma^2 > 0$,

$$\mathbb{P}\left\{\exists K > 0 : \left|\sum_{k=1}^{K} e_k\right| \geq \epsilon, \ \sum_{k=1}^{K} \text{Var}[e_k \mid \mathcal{F}_k] \leq \sigma^2\right\} \leq 2 \exp\left(\frac{-\epsilon^2/2}{\sigma^2 + R\epsilon/3}\right).$$

## H.2 Basic Matrix Inequalities

**Lemma H.3.** Assume $\mathbf{G}_1$ and $\mathbf{G}_2 \in \mathbb{R}^{d \times d}$ are two positive semi-definite matrices. Then we have

$$\left\|\mathbf{G}_1^{-1}\right\| \leq \left\|\mathbf{G}_2^{-1}\right\| + \left\|\mathbf{G}_1^{-1}\right\| \cdot \left\|\mathbf{G}_2^{-1}\right\| \cdot \left\|\mathbf{G}_1 - \mathbf{G}_2\right\|$$

and

$$\|\mathbf{u}\|_{\mathbf{G}_1^{-1}} \leq \left[1 + \sqrt{\left(\left\|\mathbf{G}_2^{-1}\right\| \cdot \|\mathbf{G}_2\|\right)^{1/2} \cdot \left\|\mathbf{G}_1^{-1}\right\| \cdot \left\|\mathbf{G}_1 - \mathbf{G}_2\right\|}\right] \cdot \|\mathbf{u}\|_{\mathbf{G}_2^{-1}},$$

for all $\mathbf{u} \in \mathbb{R}^d$.

*Proof of Lemma H.3.* The first inequality is by

$$\left\|\mathbf{G}_1^{-1}\right\| \leq \left\|\mathbf{G}_2^{-1}\right\| + \left\|\mathbf{G}_2^{-1} - \mathbf{G}_1^{-1}\right\| \leq \left\|\mathbf{G}_2^{-1}\right\| + \left\|\mathbf{G}_2^{-1}\right\| \cdot \left\|\mathbf{G}_2 - \mathbf{G}_1\right\| \cdot \left\|\mathbf{G}_1^{-1}\right\|.$$

To prove the second one, note that

$$
\begin{aligned}
\|\mathbf{u}\|_{\mathbf{G}_1^{-1}} &= \sqrt{\mathbf{u}^\top \mathbf{G}_1^{-1} \mathbf{u}} \\
&= \sqrt{\mathbf{u}^\top (\mathbf{G}_1^{-1} - \mathbf{G}_2^{-1})\mathbf{u} + \mathbf{u}^\top \mathbf{G}_2^{-1} \mathbf{u}} \\
&= \sqrt{\mathbf{u}^\top \mathbf{G}_2^{-1/2} \left[\mathbf{I} + (\mathbf{G}_2^{1/2} \mathbf{G}_1^{-1} \mathbf{G}_2^{1/2} - \mathbf{I})\right] \mathbf{G}_2^{-1/2} \mathbf{u}} \\
&\leq \left(1 + \left\|\mathbf{G}_2^{1/2} \mathbf{G}_1^{-1} \mathbf{G}_2^{1/2} - \mathbf{I}\right\|^{1/2}\right) \cdot \|\mathbf{u}\|_{\mathbf{G}_2^{-1}},
\end{aligned}
$$

and the rest follows from

$$
\begin{aligned}
\left\|\mathbf{G}_2^{1/2} \mathbf{G}_1^{-1} \mathbf{G}_2^{1/2} - \mathbf{I}\right\| &= \left\|\mathbf{G}_2^{1/2} (\mathbf{G}_1^{-1} - \mathbf{G}_2^{-1})\mathbf{G}_2^{1/2}\right\| \\
&= \left\|\mathbf{G}_2^{1/2} \mathbf{G}_1^{-1}(\mathbf{G}_1 - \mathbf{G}_2)\mathbf{G}_2^{-1}\mathbf{G}_2^{1/2}\right\| \\
&\leq \left(\left\|\mathbf{G}_2^{-1}\right\| \cdot \|\mathbf{G}_2\|\right)^{1/2} \cdot \left\|\mathbf{G}_1^{-1}\right\| \cdot \left\|\mathbf{G}_1 - \mathbf{G}_2\right\|.
\end{aligned}
$$

$\square$

**Lemma H.4.** Let $\varphi : \mathcal{S} \times \mathcal{A} \to \mathbb{R}^d$ be a bounded function such that $|\varphi(s, a)| \leq C$ for all $(s, a) \in \mathcal{S} \times \mathcal{A}$. For any $K > 0$ and $\lambda > 0$, define $\bar{\mathbf{G}}_K = \sum_{k=1}^{K} \varphi(s_k, a_k)\varphi(s_k, a_k)^\top + \lambda \mathbf{I}_d$ where $(s_k, a_k)$'s are i.i.d samples from some distribution $\nu$ over $\mathcal{S} \times \mathcal{A}$. Then with probability at least $1 - \delta$, it holds that

$$\left\|\frac{\bar{\mathbf{G}}_K}{K} - \mathbb{E}_\nu\left[\frac{\bar{\mathbf{G}}_K}{K}\right]\right\| \leq \frac{4\sqrt{2}C^2}{\sqrt{K}}\left(\log \frac{2d}{\delta}\right)^{1/2}.$$

*Proof of Lemma H.4.* Denote $\mathbf{x}_k = \boldsymbol{\varphi}(s_k, a_k)$. Denote $\widetilde{\boldsymbol{\Sigma}}_h$ as the matrix obtained by replacing the $k$-th vector $\mathbf{x}_k$ in $\widehat{\boldsymbol{\Sigma}}_h$ by $\widetilde{\mathbf{x}}_k$ and leaving the rest $K-1$ vectors unchanged. Then we have

$$\left( \frac{\widehat{\boldsymbol{\Sigma}}_h}{K} - \frac{\widetilde{\boldsymbol{\Sigma}}_h}{K} \right)^2 = \left( \frac{\mathbf{x}_k \mathbf{x}_k^\top - \widetilde{\mathbf{x}}_k \widetilde{\mathbf{x}}_k^\top}{K} \right)^2$$

$$\preceq \frac{1}{K^2} \left( 2\mathbf{x}_k \mathbf{x}_k^\top \mathbf{x}_k \mathbf{x}_k^\top + 2\widetilde{\mathbf{x}}_k \widetilde{\mathbf{x}}_k^\top \widetilde{\mathbf{x}}_k \widetilde{\mathbf{x}}_k^\top \right)$$

$$\preceq \frac{1}{K^2} \left( 2C^4 \mathbf{I}_d + 2C^4 \mathbf{I}_d \right)$$

$$= \frac{4C^4}{K^2} \cdot \mathbf{I}_d$$

$$:= \boldsymbol{A}_k^2,$$

where the first inequality uses the fact that $(\boldsymbol{A} - \boldsymbol{B})^2 \preceq 2\boldsymbol{A}^2 + 2\boldsymbol{B}^2$ for all p.s.d. matrices $\boldsymbol{A}$ and $\boldsymbol{B}$, the second inequality is from $\|\boldsymbol{\varphi}\| \leq C$. Note that we have

$$\left\| \sum_k \boldsymbol{A}_k^2 \right\| = \frac{4C^4}{K}.$$

Then by Lemma H.1, we have: for all $t > 0$,

$$\mathbb{P}\left\{ \left\| \frac{\widehat{\boldsymbol{\Sigma}}_h}{K} - \mathbb{E}\left[ \frac{\widehat{\boldsymbol{\Sigma}}_h}{K} \right] \right\| \geq t \right\} \leq 2d \cdot \exp\left( \frac{-t^2 K}{32 C^4} \right).$$

Equivalently, with probability at least $1 - \delta$,

$$\left\| \frac{\widehat{\boldsymbol{\Sigma}}_h}{K} - \mathbb{E}\left[ \frac{\widehat{\boldsymbol{\Sigma}}_h}{K} \right] \right\| \leq \frac{4\sqrt{2}C^2}{\sqrt{K}} \left( \log \frac{2d}{\delta} \right)^{1/2}.$$

This completes the proof. $\qquad \square$

**Lemma H.5.** Let $\boldsymbol{\varphi} : \mathcal{S} \times \mathcal{A} \to \mathbb{R}^d$ be a bounded function such that $\|\boldsymbol{\varphi}(s, a)\|_2 \leq C$ for all $(s, a) \in \mathcal{S} \times \mathcal{A}$. For any $K > 0$ and $\lambda > 0$, define $\bar{\mathbf{G}}_K = \sum_{k=1}^{K} \boldsymbol{\varphi}(s_k, a_k)\boldsymbol{\varphi}(s_k, a_k)^\top + \lambda \mathbf{I}_d$ where $(s_k, a_k)$'s are i.i.d samples from some distribution $\nu$ over $\mathcal{S} \times \mathcal{A}$. Let $\mathbf{G} = \mathbb{E}_\nu[\boldsymbol{\varphi}(s, a)\boldsymbol{\varphi}(s, a)^\top]$. Then for any $\delta \in (0, 1)$, if $K$ satisfies that

$$K \geq \max\left\{ 512 C^4 \|\mathbf{G}^{-1}\|^2 \log\left( \frac{2d}{\delta} \right), 4\lambda \|\mathbf{G}^{-1}\| \right\}. \tag{H.1}$$

Then with probability at least $1 - \delta$, it holds simultaneously for all $\mathbf{u} \in \mathbb{R}^d$ that

$$\|\mathbf{u}\|_{\bar{\mathbf{G}}_K^{-1}} \leq \frac{2}{\sqrt{K}} \|\mathbf{u}\|_{\mathbf{G}^{-1}}.$$

*Proof of Lemma H.5.* Note that

$$\|\mathbf{u}\|_{\bar{\mathbf{G}}_K^{-1}} = \frac{1}{\sqrt{K}} \sqrt{\mathbf{u}^\top \mathbf{G}^{-1} \mathbf{u} + \mathbf{u}^\top \left[ \left( \frac{\bar{\mathbf{G}}_K}{K} \right)^{-1} - \mathbf{G}^{-1} \right] \mathbf{u}}$$

$$= \frac{1}{\sqrt{K}} \sqrt{\mathbf{u}^\top \mathbf{G}^{-1} \mathbf{u} + \mathbf{u}^\top \mathbf{G}^{-1/2} \left[ \mathbf{G}^{1/2} \left( \frac{\bar{\mathbf{G}}_K}{K} \right)^{-1} \mathbf{G}^{1/2} - \mathbf{I}_d \right] \mathbf{G}^{-1/2} \mathbf{u}}$$

$$\leq \frac{1}{\sqrt{K}} \left( 1 + \left\| \mathbf{G}^{1/2} \left( \frac{\bar{\mathbf{G}}_K}{K} \right)^{-1} \mathbf{G}^{1/2} - \mathbf{I}_d \right\|^{1/2} \right) \|\mathbf{u}\|_{\mathbf{G}^{-1}}, \tag{H.2}$$

where the last inequality follows from Cauchy-Schwartz inequality.

It then reduces to bound $\left\| \mathbf{G}^{1/2} \left( \bar{\mathbf{G}}_K / K \right)^{-1} \mathbf{G}^{1/2} - \mathbf{I}_d \right\|$, which can be further bounded by

$$\left\| \mathbf{G}^{1/2} \left( \frac{\bar{\mathbf{G}}_K}{K} \right)^{-1} \mathbf{G}^{1/2} - \mathbf{I}_d \right\| \leq \left\| \left[ \mathbf{G}^{-1/2} \frac{\bar{\mathbf{G}}_K}{K} \mathbf{G}^{-1/2} \right]^{-1} \right\| \cdot \left\| \mathbf{I}_d - \mathbf{G}^{-1/2} \frac{\bar{\mathbf{G}}_K}{K} \mathbf{G}^{-1/2} \right\|. \quad \text{(H.3)}$$

By Lemma H.4, we have

$$\left\| \frac{\bar{\mathbf{G}}_K}{K} - \mathbb{E} \left[ \frac{\bar{\mathbf{G}}_K}{K} \right] \right\| \leq \frac{4\sqrt{2}C^2}{\sqrt{K}} \left( \log \frac{2d}{\delta} \right)^{1/2}$$

with probability at least $1 - \delta$, and thus

$$\left\| \mathbf{I} - \mathbf{G}^{-1/2} \frac{\bar{\mathbf{G}}_K}{K} \mathbf{G}^{-1/2} \right\| \leq \left[ \left\| \frac{\bar{\mathbf{G}}_K}{K} - \mathbb{E} \left[ \frac{\bar{\mathbf{G}}_K}{K} \right] \right\| + \left\| \mathbb{E} \left[ \frac{\bar{\mathbf{G}}_K}{K} \right] - \mathbf{G} \right\| \right] \cdot \| \mathbf{G}^{-1} \|$$

$$\leq \frac{4\sqrt{2}C^2 \| \mathbf{G}^{-1} \|}{\sqrt{K}} \sqrt{\log \frac{2d}{\delta}} + \frac{\lambda \| \mathbf{G}^{-1} \|}{K}$$

$$\leq \frac{1}{2} \quad \text{(H.4)}$$

where the last inequality follows from the assumption (H.1). Therefore,

$$\lambda_{\min} \left( \mathbf{G}^{-1/2} \frac{\bar{\mathbf{G}}_K}{K} \mathbf{G}^{-1/2} \right) \geq 1 - \left\| \mathbf{I} - \mathbf{G}^{-1/2} \frac{\bar{\mathbf{G}}_K}{K} \mathbf{G}^{-1/2} \right\| \geq \frac{1}{2}$$

with probability at least $1 - \delta$. This further implies that

$$\left\| \left[ \mathbf{G}^{-1/2} \frac{\bar{\mathbf{G}}_K}{K} \mathbf{G}^{-1/2} \right]^{-1} \right\| = \lambda_{\min} \left( \mathbf{G}^{-1/2} \frac{\bar{\mathbf{G}}_K}{K} \mathbf{G}^{-1/2} \right)^{-1} \leq 2. \quad \text{(H.5)}$$

Combining (H.3), (H.4) and (H.5) yields that

$$\left\| \mathbf{G}^{1/2} \left( \frac{\bar{\mathbf{G}}_K}{K} \right)^{-1} \mathbf{G}^{1/2} - \mathbf{I}_d \right\| \leq 1 \quad \text{(H.6)}$$

with probability at least $1 - \delta$. Then plug (H.6) back into (H.2), and we obtain that

$$\| \mathbf{u} \|_{\bar{\mathbf{G}}_K^{-1}} \leq \frac{2}{\sqrt{K}} \| \mathbf{u} \|_{\mathbf{G}^{-1}}$$

with probability at least $1 - \delta$. Note that in the above argument we only need to bound $\left\| \mathbf{G}^{1/2} \left( \bar{\mathbf{G}}_K / K \right)^{-1} \mathbf{G}^{1/2} - \mathbf{I}_d \right\|$ which is independent of the choice of $\mathbf{u}$, thus it holds for all $\mathbf{u} \in \mathbb{R}^d$ simultaneously. This completes the proof. □

### H.3 Inequalities for Sample Covariance Matrices

Here we introduce some useful lemmas about the inverse Gram matrix.

**Lemma H.6** (Lemma D.1, Jin et al. 17). Let $\mathbf{\Lambda}_t = \sum_{i=1}^{t} \mathbf{x}_i \mathbf{x}_i^\top + \lambda I$ where $\lambda > 0$ and $\mathbf{x}_i \in \mathbb{R}^d$. Then

$$\sum_{i=1}^{t} \mathbf{x}_i^\top \mathbf{\Lambda}_t^{-1} \mathbf{x}_i \leq d.$$

*Proof of Lemma H.6.* Note that

$$\sum_{i=1}^{t} \mathbf{x}_i^\top \mathbf{\Lambda}_t^{-1} \mathbf{x}_i = \sum_{i=1}^{t} \text{tr}(\mathbf{x}_i^\top \mathbf{\Lambda}_t^{-1} \mathbf{x}_i) = \text{tr} \left( \mathbf{\Lambda}_t^{-1} \sum_{i=1}^{t} \mathbf{x}_i \mathbf{x}_i^\top \right).$$

Using the eigen-decomposition $\sum_{i=1}^{t} \mathbf{x}_i \mathbf{x}_i^\top = \mathbf{U} \text{diag}(\lambda_1, \cdots, \lambda_d) \mathbf{U}^\top$, we have $\mathbf{\Lambda}_t = \mathbf{U} \text{diag}(\lambda_1 + 1, \cdots, \lambda_d + 1) \mathbf{U}^\top$, and it follows that

$$\text{tr} \left( \mathbf{\Lambda}_t^{-1} \sum_{i=1}^{t} \mathbf{x}_i \mathbf{x}_i^\top \right) = \sum_{j=1}^{d} \frac{\lambda_j}{\lambda_j + \lambda} \leq d.$$

□

**Lemma H.7.** For any $h \in [H]$ and $L_1, L_2 > 0$, let $\sigma_1, \sigma_2 \in \mathcal{T}_h(L_1, L_2)$ such that $\sup_{s,a} |\sigma_1(s, a) - \sigma_2(s, a)| \le \epsilon$. Define

$$\mathbf{\Lambda}_1 := \sum_{k=1}^K \phi(s_{k,h}, a_{k,h})\phi(s_{k,h}, a_{k,h})^\top / \sigma_1(s_{k,h}, a_{k,h})^2 + \lambda \mathbf{I}_d \,,$$

$$\mathbf{\Lambda}_2 := \sum_{k=1}^K \phi(s_{k,h}, a_{k,h})\phi(s_{k,h}, a_{k,h})^\top / \sigma_2(s_{k,h}, a_{k,h})^2 + \lambda \mathbf{I}_d \,.$$

Then under Assumption 2.1, it holds that

$$\|\mathbf{\Lambda}_1 - \mathbf{\Lambda}_2\| \le \frac{2K\sqrt{(H - h + 1)^2 + \sigma_r^2} \cdot \epsilon}{(\eta_h + \sigma_r^2)^2},$$

and

$$\|\mathbf{\Lambda}_1^{-1} - \mathbf{\Lambda}_2^{-1}\| \le \frac{2K\sqrt{(H - h + 1)^2 + \sigma_r^2} \cdot \epsilon}{\lambda^2(\eta_h + \sigma_r^2)^2}.$$

*Proof of Lemma H.7.* We have

$$\mathbf{\Lambda}_1 - \mathbf{\Lambda}_2 = \sum_{k=1}^K \phi(s_{k,h}, a_{k,h})\phi^\top(s_{k,h}, a_{k,h}) \left( \frac{1}{\sigma_1^2(s_{k,h}, a_{k,h})} - \frac{1}{\sigma_2^2(s_{k,h}, a_{k,h})} \right)$$

and thus

$$\|\mathbf{\Lambda}_1 - \mathbf{\Lambda}_2\| \le \sum_{k=1}^K \|\phi(s_{k,h}, a_{k,h})\phi^\top(s_{k,h}, a_{k,h})\| \cdot \left| \frac{1}{\sigma_1^2(s_{k,h}, a_{k,h})} - \frac{1}{\sigma_2^2(s_{k,h}, a_{k,h})} \right|$$

$$\le \sum_{k=1}^K \left| \frac{1}{\sigma_1^2(s_{k,h}, a_{k,h})} - \frac{1}{\sigma_2^2(s_{k,h}, a_{k,h})} \right|$$

$$= \sum_{k=1}^K \left| \frac{|\sigma_1(s_{k,h}, a_{k,h}) + \sigma_2(s_{k,h}, a_{k,h})| \cdot |\sigma_1(s_{k,h}, a_{k,h}) - \sigma_2(s_{k,h}, a_{k,h})|}{\sigma_1^2(s_{k,h}, a_{k,h})\sigma_2^2(s_{k,h}, a_{k,h})} \right|$$

$$\le K \cdot \frac{2\sqrt{(H - h + 1)^2 + \sigma_r^2}}{(\eta_h + \sigma_r^2)^2} \cdot \epsilon$$

where the first inequality is from the assumption that $\|\phi(s, a)\| \le 1$ for all $(s, a) \in \mathcal{S} \times \mathcal{A}$ and the second inequality is by $\sigma^2(\cdot) \in [\eta_h + \sigma_r^2, (H - h + 1)^2 + \sigma_r^2]$. It then follows that

$$\|\mathbf{\Lambda}_1^{-1} - \mathbf{\Lambda}_2^{-1}\| = \|\mathbf{\Lambda}_1^{-1}(\mathbf{\Lambda}_1 - \mathbf{\Lambda}_2)\mathbf{\Lambda}_2^{-1}\|$$

$$\le \|\mathbf{\Lambda}_1^{-1}\| \cdot \|\mathbf{\Lambda}_1 - \mathbf{\Lambda}_2\| \cdot \|\mathbf{\Lambda}_2^{-1}\|$$

$$\le \frac{2K\sqrt{(H - h + 1)^2 + \sigma_r^2} \cdot \epsilon}{\lambda^2(\eta_h + \sigma_r^2)^2},$$

where in the last inequality we use $\|\mathbf{\Lambda}_1^{-1}\|, \|\mathbf{\Lambda}_1^{-2}\| \le 1/\lambda$. $\qquad\square$

**Lemma H.8.** For any $h \in [H]$ and $L_1, L_2 > 0$, let $\sigma_1, \sigma_2 \in \mathcal{T}_h(L_1, L_2)$ such that $\sup_{s,a} |\sigma_1^2(s, a) - \sigma_2^2(s, a)| \le \epsilon$. Then it holds that

$$\|\mathbf{\Lambda}_1 - \mathbf{\Lambda}_2\| \le \frac{K}{(\eta + \sigma_r^2)^2} \cdot \epsilon, \ \|\mathbf{\Lambda}_1^{-1} - \mathbf{\Lambda}_2^{-1}\| \le \frac{K}{\lambda^2 (\eta + \sigma_r^2)^2} \cdot \epsilon.$$

*Proof of Lemma H.8.* Note that

$$\|\mathbf{\Lambda}_1 - \mathbf{\Lambda}_2\| \le \sum_{k=1}^K \|\phi(s_{k,h}, a_{k,h})\phi^\top(s_{k,h}, a_{k,h})\| \cdot \left| \frac{1}{\sigma_1^2(s_{k,h}, a_{k,h})} - \frac{1}{\sigma_2^2(s_{k,h}, a_{k,h})} \right|$$

$$\le \sum_{k=1}^K \left| \frac{\sigma_2^2(s_{k,h}, a_{k,h}) - \sigma_1^2(s_{k,h}, a_{k,h})}{\sigma_1^2(s_{k,h}, a_{k,h}) \cdot \sigma_2^2(s_{k,h}, a_{k,h})} \right|,$$

and the rest follows from the proof of Lemma H.7. $\qquad\square$

### H.4 Bounds for self-normalized vector-valued martingales

Here we introduce some concentration inequalities that can be applied to bound the self-normalized martingales.

**Theorem H.9** (Hoeffding inequality for self-normalized martingales, Abbasi-Yadkori et al. [1]). Let $\{\eta_t\}_{t=1}^{\infty}$ be a real-valued stochastic process. Let $\{\mathcal{F}_t\}_{t=0}^{\infty}$ be a filtration, such that $\eta_t$ is $\mathcal{F}_t$-measurable. Assume $\eta_t \mid \mathcal{F}_{t-1}$ is zero-mean and $R$-subgaussian for some $R > 0$, i.e.,

$$\forall \lambda \in \mathbb{R}, \quad \mathbb{E}\left[e^{\lambda \eta_t | \mathcal{F}_{t-1}}\right] \leq e^{\lambda^2 R^2 / 2}.$$

Let $\{\mathbf{x}_t\}_{t=1}^{\infty}$ be an $\mathbb{R}^d$-valued stochastic process where $\mathbf{x}_t$ is $\mathcal{F}_{t-1}$-measurable. Assume $\mathbf{\Lambda}_0$ is a $d \times d$ positive definite matrix, and define $\mathbf{\Lambda}_t = \mathbf{\Lambda}_0 + \sum_{s=1}^{t} \mathbf{x}_s \mathbf{x}_s^{\top}$. Then, for any $\delta > 0$, with probability at least $1 - \delta$, for all $t > 0$,

$$\left\| \sum_{s=1}^{t} \mathbf{x}_s \eta_s \right\|_{\mathbf{\Lambda}_t^{-1}}^2 \leq 2R^2 \log \left( \frac{\det(\mathbf{\Lambda}_t)^{1/2} \det(\mathbf{\Lambda}_0)^{-1/2}}{\delta} \right).$$

**Theorem H.10** (Bernstein inequality for self-normalized martingales, Zhou et al. [53]). Let $\{\eta_t\}_{t=1}^{\infty}$ be a real-valued stochastic process. Let $\{\mathcal{F}_t\}_{t=0}^{\infty}$ be a filtration, such that $\eta_t$ is $\mathcal{F}_t$-measurable. Assume $\eta_t$ also satisfies

$$|\eta_t| \leq R, \ \mathbb{E}[\eta_t \mid \mathcal{F}_{t-1}] = 0, \ \mathbb{E}[\eta_t^2 \mid \mathcal{F}_{t-1}] \leq \sigma^2.$$

Let $\{\mathbf{x}_t\}_{t=1}^{\infty}$ be an $\mathbb{R}^d$-valued stochastic process where $\mathbf{x}_t$ is $\mathcal{F}_{t-1}$-measurable and $\|\mathbf{x}_t\| \leq L$. Let $\mathbf{\Lambda}_t = \lambda \mathbf{I}_d + \sum_{s=1}^{t} \mathbf{x}_s \mathbf{x}_s^{\top}$. Then, for any $\delta > 0$, with probability at least $1 - \delta$, for all $t > 0$,

$$\left\| \sum_{s=1}^{t} \mathbf{x}_s \eta_s \right\|_{\mathbf{\Lambda}_t^{-1}} \leq 8\sigma \sqrt{d \log \left(1 + \frac{tL^2}{\lambda d}\right) \cdot \log \left(\frac{4t^2}{\delta}\right)} + 4R \log \left(\frac{4t^2}{\delta}\right).$$

### H.5 Auxiliary Results for Self-normalized Martingales

Assume the function $\sigma_1(\cdot, \cdot)$ is computed using the function $V_1(\cdot)$ in the same way $\widehat{\sigma}_h$ is computed using $\widehat{V}_{h+1}^{\pi}$ as in Algorithm 1. In this way, we can view $\sigma_1$ as a function parameterized by $V_1$. And similar for $\sigma_2$ and $V_2$.

**Lemma H.11.** Assume $V_1$ and $V_2 \in \mathcal{V}_{h+1}(L)$ and satisfy $\sup_s |V_1(s) - V_2(s)| \leq \epsilon$. Then

$$\sup_{s,a} |\sigma_1(s,a) - \sigma_2(s,a)| \leq 2\sqrt{\frac{K(H-h+1)}{\lambda}} \cdot \sqrt{\epsilon},$$

$$\sup_{s,a} |\sigma_1^2(s,a) - \sigma_2^2(s,a)| \leq \frac{4K(H-h+1)}{\lambda}\epsilon.$$

*Proof of Lemma H.11.* By the proof of Lemma H.14, we have

$$\sup_{s,a} |\sigma_1(s,a) - \sigma_2(s,a)| \leq \sup_{s,a} \sqrt{|\sigma_1^2(s,a) - \sigma_2^2(s,a)|} \leq \sqrt{\|\boldsymbol{\beta}_1 - \boldsymbol{\beta}_2\| + 2(H-h+1) \cdot \|\boldsymbol{\theta}_1 - \boldsymbol{\theta}_2\|}.$$

Note that

$$\|\boldsymbol{\theta}_1 - \boldsymbol{\theta}_2\| = \left\| (\widehat{\mathbf{\Sigma}}_h)^{-1} \sum_{k=1}^{K} \boldsymbol{\phi}(\check{s}_{k,h}, \check{a}_{k,h})(V_1 - V_2)(\check{s}'_{k,h}) \right\| \leq \frac{K}{\lambda}\epsilon,$$

where we use $\left\| (\widehat{\mathbf{\Sigma}}_h)^{-1} \right\| \leq 1/\lambda$ and $\|\boldsymbol{\phi}(s,a)\| \leq 1$ for all $(s,a)$. Similarly, we can show

$$\|\boldsymbol{\beta}_1 - \boldsymbol{\beta}_2\| \leq \left\| (\widehat{\mathbf{\Sigma}}_h)^{-1} \sum_{k=1}^{K} \boldsymbol{\phi}(\check{s}_{k,h}, \check{a}_{k,h})(V_1(\check{s}'_{k,h})^2 - V_2(\check{s}'_{k,h})^2) \right\|$$

$$\leq \left\| (\widehat{\mathbf{\Sigma}}_h)^{-1} \right\| \cdot K \cdot 2(H-h+1)\epsilon \leq \frac{2K(H-h+1)}{\lambda}\epsilon.$$

Altogether, we have

$$\sup_{s,a} |\sigma_1(s,a) - \sigma_2(s,a)| \leq 2\sqrt{\frac{K(H-h+1)}{\lambda}} \cdot \sqrt{\epsilon},$$

and

$$\sup_{s,a} |\sigma_1^2(s,a) - \sigma_2^2(s,a)| \leq \frac{4K(H-h+1)}{\lambda}\epsilon.$$

$\square$

### H.6 Covering numbers of the function classes

Here we compute the covering numbers of the function classes $\mathcal{V}_h$ and $\mathcal{T}_h$.

**Lemma H.12** (Covering number of the Euclidean Ball). For any $\epsilon > 0$, the $\epsilon$-covering number of the ball of radius $r$ under the Euclidean norm satisfies $\mathcal{N}_\epsilon \leq (1 + 2r/\epsilon)^d$.

A proof of this classical result can be found, for example, in the work by Vershynin [42]. Now we give the covering number of the function class $\mathcal{V}_h(L)$ for all $h \in [H]$ and $L > 0$.

**Lemma H.13.** For any $h \in [H]$ and any $L > 0$, let $\mathcal{V}_h(L)$ be as defined in (C.2). Let $\mathcal{N}_\epsilon$ denote the $\epsilon$-covering number of $\mathcal{V}_h(L)$ with respect to the distance $\text{dist}(V_1, V_2) = \sup_s |V_1(s) - V_2(s)|$. Then under Assumption 2.1, it holds that

$$\mathcal{N}_\epsilon \leq \left(1 + \frac{2L}{\epsilon}\right)^d.$$

*Proof of Lemma H.13.* For any $V_1, V_2 \in \mathcal{V}_h(L)$ parametrized by $\mathbf{w}_1$ and $\mathbf{w}_2$ respectively, we have

$$\text{dist}(V_1, V_2) = \sup_s |\langle \phi_h^\pi(s), \mathbf{w}_1 - \mathbf{w}_2 \rangle| \leq \|\mathbf{w}_1 - \mathbf{w}_2\|_2 \cdot \sup_s \|\phi_h^\pi(s)\|_2 \leq \|\mathbf{w}_1 - \mathbf{w}_2\|_2,$$

where the first inequality is by Cauchy-Schwarz inequality and the second inequality uses the assumption that $\|\phi(s,a)\| \leq 1$.

Let $\mathcal{C}_\mathbf{w}(\epsilon)$ be an $\epsilon-$cover of the Euclidean ball $\{\mathbf{w} \in \mathbb{R}^d | \|\mathbf{w}\|_2 \leq L\}$. Then for any $V_1 \in \mathcal{V}_h(L)$, there exists a $V_2 \in \mathcal{V}_h(L)$ parametrized by $\mathbf{w}_2 \in \mathcal{C}_\mathbf{w}(\epsilon)$ such that $\text{dist}(V_1, V_2) \leq \epsilon$. Then we see that

$$\mathcal{N}_\epsilon \leq |\mathcal{C}_\mathbf{w}(\epsilon)| \leq \left(1 + \frac{2L}{\epsilon}\right)^d,$$

where the second inequality follows from Lemma H.12. $\square$

**Lemma H.14.** For any $h \in [H]$ and $L_1, L_2 > 0$, let $\mathcal{T}_h(L_1, L_2)$ be as defined in (C.3). Let $\mathcal{N}_\epsilon$ denote the $\epsilon$-covering number of $\mathcal{T}_h(L_1, L_2)$ with respect to the distance $\text{dist}(\sigma_1, \sigma_2) = \sup_{s,a} |\sigma_1(s,a) - \sigma_2(s,a)|$. Then under Assumption 2.1, it holds that

$$\mathcal{N}_\epsilon \leq \left(1 + \frac{4L_1}{\epsilon^2}\right)^d \cdot \left(1 + \frac{8(H-h+1)L_2}{\epsilon^2}\right)^d.$$

*Proof of Lemma H.14.* For any $\sigma_1, \sigma_2 \in \mathcal{T}$ which are parameterized by $(\boldsymbol{\beta}_1, \boldsymbol{\theta}_1)$ and $(\boldsymbol{\beta}_2, \boldsymbol{\theta}_2)$ respectively, we have

$\text{dist}(\sigma_1, \sigma_2)$

$= \sup_{s,a} |\sigma_1(s,a) - \sigma_2(s,a)|$

$\leq \sup_{s,a} \sqrt{|\sigma_1^2(s,a) - \sigma_2^2(s,a)|}$

$\leq \sup_{s,a} \sqrt{\left|\langle \boldsymbol{\phi}(s,a), \boldsymbol{\beta}_1 \rangle - \langle \boldsymbol{\phi}(s,a), \boldsymbol{\beta}_2 \rangle\right| + \left|[\langle \boldsymbol{\phi}(s,a), \boldsymbol{\theta}_1 \rangle_{[0,H-h+1]}]^2 - [\langle \boldsymbol{\phi}(s,a), \boldsymbol{\theta}_2 \rangle_{[0,H-h+1]}]^2\right|}$

$\leq \sup_{s,a} \sqrt{\left|\langle \boldsymbol{\phi}(s,a), \boldsymbol{\beta}_1 \rangle - \langle \boldsymbol{\phi}(s,a), \boldsymbol{\beta}_2 \rangle\right| + 2(H-h+1) \cdot |\langle \boldsymbol{\phi}(s,a), \boldsymbol{\theta}_1 \rangle - \langle \boldsymbol{\phi}(s,a), \boldsymbol{\theta}_2 \rangle|}$

$\leq \sqrt{\|\boldsymbol{\beta}_1 - \boldsymbol{\beta}_2\| + 2(H-h+1) \cdot \|\boldsymbol{\theta}_1 - \boldsymbol{\theta}_2\|}.$

where the first inequality uses the fact that $|a - b| \leq \sqrt{|a^2 - b^2|}$ for any $a, b \geq 0$, the second and the third inequalities follows from the fact that $\max\{\eta_h, \cdot\}$ and the clipping $\{\cdot\}_{[0,(H-h+1)^2]}$, $\{\cdot\}_{[0,H-h+1]}$ are all contraction maps, and the last inequality is by Cauchy-Schwarz inequality and the assumption that $\|\phi(s, a)\| \leq 1$.

In order to have $\mathrm{dist}(\sigma_1, \sigma_2) \leq \epsilon$, it suffices to have $\|\boldsymbol{\beta}_1 - \boldsymbol{\beta}_2\| \leq \epsilon^2/2$ and $2(H-h+1)\|\boldsymbol{\theta}_1 - \boldsymbol{\theta}_2\| \leq \epsilon^2/2$. By Lemma H.12, in order to $\epsilon^2/2$-cover $\{\boldsymbol{\beta} : \|\boldsymbol{\beta}\| < L_1\}$ and $\epsilon^2/(4(H - h + 1))$-cover $\{\boldsymbol{\theta} : \|\boldsymbol{\theta}\| \leq L_2\}$ we need

$$\mathcal{N}_\beta \leq \left(1 + \frac{4L_1}{\epsilon^2}\right)^d , \mathcal{N}_\theta \leq \left(1 + \frac{8(H-h+1)L_2}{\epsilon^2}\right)^d .$$

Altogether, to $\epsilon$-cover $\mathcal{T}$, we have

$$\mathcal{N}_\epsilon \leq \mathcal{N}_\beta \cdot \mathcal{N}_\theta \leq \left(1 + \frac{4L_1}{\epsilon^2}\right)^d \cdot \left(1 + \frac{8(H-h+1)L_2}{\epsilon^2}\right)^d .$$

$\square$

## H.7  Bounds for the Regression Estimators

**Lemma H.15.** Assume $\sup_s \left|\widehat{V}_{h+1}^\pi(s)\right| \leq B$ for some $B \geq 0$. Then $\widehat{\boldsymbol{\theta}}_h$, $\widehat{\boldsymbol{\beta}}_h$ and $\widehat{\mathbf{w}}_h^\pi$ in Algorithm 1 satisfy the following:

$$\|\widehat{\boldsymbol{\theta}}_h\| \leq B\sqrt{\frac{Kd}{\lambda}}, \quad \|\widehat{\boldsymbol{\beta}}_h\| \leq B^2\sqrt{\frac{Kd}{\lambda}}, \quad \|\widehat{\mathbf{w}}_h^\pi\| \leq \frac{B+1}{\sqrt{\eta + \sigma_r^2}}\sqrt{\frac{Kd}{\lambda}}.$$

*Proof of Lemma H.15.* For any vector $\mathbf{v} \in \mathbb{R}^d$, we have

$$|\mathbf{v}^\top \widehat{\boldsymbol{\theta}}_h| = \left|\mathbf{v}^\top (\widehat{\boldsymbol{\Sigma}}_h)^{-1} \sum_{k=1}^K \phi(\check{s}_{k,h}, \check{a}_{k,h}) \widehat{V}_{h+1}^\pi(\check{s}_{k,h}')\right|$$

$$\leq \sum_{k=1}^K |\mathbf{v}^\top (\widehat{\boldsymbol{\Sigma}}_h)^{-1} \phi(\check{s}_{k,h}, \check{a}_{k,h})| \cdot \sup_s |\widehat{V}_{h+1}^\pi(s)|$$

$$\leq B \cdot \sqrt{\left[\sum_{k=1}^K \mathbf{v}^\top (\widehat{\boldsymbol{\Sigma}}_h)^{-1} \mathbf{v}\right] \cdot \left[\sum_{k=1}^K \phi(\check{s}_{k,h}, \check{a}_{k,h})^\top (\widehat{\boldsymbol{\Sigma}}_h)^{-1} \phi(\check{s}_{k,h}, \check{a}_{k,h})\right]}$$

$$\leq B\|\mathbf{v}\|_2 \sqrt{\frac{K}{\lambda}} \cdot \sqrt{d},$$

where the second inequality is by Cauchy-Schwarz inequality, and the last inequality uses $\|(\widehat{\boldsymbol{\Sigma}}_h)^{-1}\| \leq 1/\lambda$ and Lemma H.6. It follows that $\|\widehat{\boldsymbol{\theta}}_h\| \leq B\sqrt{\frac{Kd}{\lambda}}$. Similarly, we have $\|\widehat{\boldsymbol{\beta}}_h\| \leq B^2\sqrt{\frac{Kd}{\lambda}}$ since $\sup_s |\widehat{V}_{h+1}^\pi(s)|^2 \leq B^2$. To bound $\|\widehat{\mathbf{w}}_h^\pi\|$, note that

$$|\mathbf{v}^\top \widehat{\mathbf{w}}_h^\pi| = \left|\mathbf{v}^\top \widehat{\boldsymbol{\Lambda}}_h^{-1} \sum_{k=1}^K \phi(s_{k,h}, a_{k,h}) Y_{k,h}/\widehat{\sigma}_{k,h}^2\right|$$

$$\leq \frac{B+1}{\sqrt{\eta_h + \sigma_r^2}} \cdot \sum_{k=1}^K \left|\mathbf{v}^\top \widehat{\boldsymbol{\Lambda}}_h^{-1} \frac{\phi(s_{k,h}, a_{k,h})}{\widehat{\sigma}_{k,h}}\right|$$

$$\leq \frac{B+1}{\sqrt{\eta_h + \sigma_r^2}} \cdot \sqrt{\left[\sum_{k=1}^K \mathbf{v}^\top (\widehat{\boldsymbol{\Lambda}}_h)^{-1} \mathbf{v}\right] \cdot \left[\sum_{k=1}^K \frac{\phi(s_{k,h}, a_{k,h})^\top}{\widehat{\sigma}_{k,h}} (\widehat{\boldsymbol{\Lambda}}_h)^{-1} \frac{\phi(s_{k,h}, a_{k,h})}{\widehat{\sigma}_{k,h}}\right]}$$

$$\leq \frac{B+1}{\sqrt{\eta_h + \sigma_r^2}} \cdot \|\mathbf{v}\|_2 \sqrt{\frac{K}{\lambda}} \cdot \sqrt{d}$$

$$= \left(\frac{B+1}{\sqrt{\eta_h + \sigma_r^2}}\sqrt{\frac{Kd}{\lambda}}\right) \|\mathbf{v}\|_2,$$

where the first inequality comes from

$$\left| \frac{Y_{k,h}}{\widehat{\sigma}_{k,h}} \right| = \left| \frac{r_{k,h} + \widehat{V}^{\pi}_{h+1}(s'_{k,h})}{\widehat{\sigma}_{k,h}} \right| \leq (B+1) \cdot \frac{1}{\sqrt{\eta_h + \sigma_r^2}},$$

and note that by assumption $|r_{k,h}| \leq 1$ $a.s.$, and by the clipping in the algorithm, $\widehat{\sigma}_{k,h} \geq \sqrt{\eta_h + \sigma_r^2}$.

$\square$