# OpenReview forum: "Variance-Aware Off-Policy Evaluation with Linear Function Approximation"
_NeurIPS.cc/2021/Conference — NeurIPS 2021 Poster_

### Official Review · Reviewer_7eNH · 2021-07-14

**Rating:** 6
**Confidence:** 4

**Summary:**

This paper proposes a new off-policy evaluation (OPE) algorithm in the finite horizon MDP setting. The reviewer finds the new results are sightly incremental compared with the existing results and thus suggests not accepting the paper.

**Limitations And Societal Impact:**

yes

**Main Review:**

This paper proposes a new off-policy evaluation (OPE) algorithm in the finite horizon MDP setting. Compared with the regular LSVI-type algorithm, the proposed algorithm involves an extra estimation step on the variance of the next-step state value and uses it to reweight the sample in the least-square regression for estimating value functions. The theoretical analysis shows that the leading term of the OPE error bound now depends on the weighted coverage matrix rather than the original coverage matrix.

My main concern is that the new theoretical result is, most of the time, equal to the previous result, while it is hard to say that it can be better than the original results with regard to the order of H. This is because the variance $\sigma(s, a)$ in average is just of order H. There might be certain state-action pairs having close-zero state-action values or having a deterministic transition distribution to the next state, but the difference showing up in the new error bound is not clearly an improvement since it does not change the order and the constants can not be directly compared.

**Time Spent Reviewing:**

2

---

> ### Author Response · Authors · 2021-08-10
> **Response to Reviewer 7eNH**
>
> Thank you for your comments.
>
> ---
> It looks that there are a few misunderstandings in your review. We will explain it in detail in the following.
>
> **Q1**: “the new theoretical result is, most of the time, equal to the previous result.”
>
> **A1**: This is a huge misjudgment of our results. We believe that the reviewer overlooks the significance of our improvement over the previous results in [Duan et al., 2020]. Our results are new and stronger than the previous results in the following aspects:
>
> (1) We consider a more general setting (time-inhomogeneous MDPs, where the transition probability can vary step by step within the same episode).
>
> (2) We propose a new algorithm that can utilize the variance information.
>
> (3) Importantly, our error bound for OPE is problem-dependent.
>
> (4) Because of dealing with the more general time-inhomogeneous setting, the analysis in the previous work for time-homogeneous setting does not work anymore. Thus, we develop a novel analysis of OPE for time-inhomogeneous MDPs. (See Appendix B.1)
>
> Note that even in the worst case where the variance of the value function $\sigma^2(s,a)\approx H^2$, our algorithm and theoretical results still are not equal to (or degenerate to) the previous results.
>
> ---
> **Q2**: “It is hard to say that it can be better than the original results with regard to the order of H. This is because $\sigma(s,a)$  in average is just of order $H$.”
>
> **A2**: This is a huge misunderstanding. $O(H^2)$ is merely a trivial upper bound of the variance and is the worst case bound. In practice, the variance $\sigma^2(s,a)$ can be much smaller than $H^2$. For example, for deterministic MDPs the variance is 0. In many real-life MDP problems, there might be only one non-zero reward which is given at the final stage $h=H$, and in this case the variance is O(1). For both aforementioned cases, the previous result would still suffer from an $O(H^2)$ factor, while our result improves it to $O(H)$. Again, we want to emphasize that our proposed algorithm can adapt to MDPs with different variances.
>
> Furthermore, even if we consider the case where the average variance is of order $H^2$, our variance-aware algorithm can still outperform the FQI-OPE in [Duan et al. 2020] empirically, since VA-OPE assigns more weight to those high-quality (i.e., low-variance) data points. This can be explained by thinking about the advantage of weighted linear regression versus ordinary linear regression. This is also verified by our numerical simulations (See Appendix A for the detailed numerical results).
>
> ---
> **Q3**: “There might be certain state-action pairs having close-zero state-action values or having a deterministic transition distribution to the next state, but the difference showing up in the new error bound is not clearly an improvement since it does not change the order and the constants can not be directly compared.”
>
> **A3**: Again, this is a misunderstanding.
> The improvement of our results comes from the term $\sum_{h=1}^H ||v_h||_{\Lambda_h^{-1}}$ in the bound of Theorem 4.1.
>
> In sharp contrast, the previous result in [Duan et al. 2020] depends on $\sum_{h=1}^H (H-h+1)||v_h||_{\Sigma_h^{-1}} = O(H^2)$ (See Appendix B.6 in their Arxiv version).
>
> As a comparison, for deterministic MDPs, the variance is 0, and thus $\sigma_h = \sqrt{2}$ for all $h$. Substituting this into our term, we can obtain $\sqrt{2} \sum_{h=1}^H ||v_h||_{\Sigma_h^{-1}} = O(H)$.
>
> Duan et al. 's term is still $\sum_{h=1}^H (H-h+1)||v_h||_{\Sigma_h^{-1}} = O(H^2)$.
>
> Similarly, when the state-action pairs have close-to-zero values, it is clear that our bound is again $O(H)$, and theirs is $O(H^2)$.
> Therefore, in the best case, the improvement of our algorithm is by a factor of $H$, which is significant in practical RL as $H$ is often a large number.
>
> The only case where our bound is equal to (i.e. recovers) the previous result is when all state-action pairs have $O(H^2)$ variance, which is very unlikely in practice.
> It is important to note that our bound is problem-dependent and is at worst $O(H^2)$, while their bound is always $O(H^2)$. This is a huge difference.
>
> ---
> We hope the above response addresses your concerns.

---

> > ### Comment · Reviewer_7eNH · 2021-08-24
> > **Response**
> >
> > Thanks for your explanation. I have changed my grade to acceptance.

---

> > > ### Author Response · Authors · 2021-08-25
> > > **Thank you for your positive feedback!**
> > >
> > > We are glad that our response has addressed your concerns.

---

### Official Review · Reviewer_c9Y4 · 2021-07-16

**Rating:** 7
**Confidence:** 5

**Summary:**

The variance-aware OPE method uses the estimated variance of the value function to reweight the Bellman residual in Fitted Q-Iteration. It shows that the algorithm achieves a tighter error bound than the best-known result.

**Ethical Concerns:**

No.

**Limitations And Societal Impact:**

Yes.

**Main Review:**

It is a really good paper with high quality. Especially, such a variance-aware structure helps people to understand what is essential in the linear MDP model and improves over the existing Duan et al. in a non-trivial way (from the statistical sense). The use of two folds of data to estimate $\widehat{\sigma}$ is neat and the fisher information interpretation for $\Lambda^{-1}$ is appropriate. I think it makes a very good contribution to the OPE community for the purpose of understanding instance-optimal / problem-dependent RL.

Only one small concern: it would be better if $s_{k, h}^{\prime} \neq s_{k, h+1}$ can be got rid of (especially Yin et al. does not really use this, their data is coming in a sequential way). But I can understand the authors since without this the analysis could be nasty due to the data dependence within the episodes.

An additional comment: since your bound include the variance structure, in the special case of tabular MDP, could your result match the dominate term in Yin & Wang 20 (square root of their MSE bound)? It would interesting if your result can achieve that since their bound is statistically optimal (matching the Cramer-Rao lower bound) for tabular RL.

This is within one of the best in my batch therefore I will choose my score as acceptance.

**Time Spent Reviewing:**

15 Hours

---

> ### Author Response · Authors · 2021-08-10
> **Response to Reviewer c9Y4**
>
> Thank you for your positive comments!
>
> ---
> **Q1**: “Since your bound includes the variance structure, in the special case of tabular MDP, could your result match the dominant term in Yin & Wang 20 (square root of their MSE bound)? It would be interesting if your result can achieve that since their bound is statistically optimal (matching the Cramer-Rao lower bound) for tabular RL.”
>
> **A1**: As one way to compare the results, we can represent a tabular MDP as a linear MDP by using one-hot feature mappings, i.e., $\phi(s,a) =  e(s,a) \in \mathbb{R}^{|S|\times|A|}$ where the $e(s,a)$ is the $(s,a)^{th}$ canonical basis.
>
> Rewriting $\Lambda_h$ and $v_h$ using the notation in [Yin & Wang 20], we have
> $\Lambda_h= \sum_{s_h,a_h} \frac{d_h^{\bar\pi}(s_h) \bar\pi(a_h|s_h)}{\mathbb{V_h} V_{h+1}^\pi (s_h,a_h)}  \cdot e(s_h,a_h) e(s_h,a_h)^\top$
> which is a diagonal matrix, and $v_h = \sum_{s_h,a_h} d_h^\pi(s_h) \pi(a_h|s_h) e(s_h,a_h)$.
>
> Then one can verify that the dominant term in our error bound (ignoring the constants and the logarithmic terms) is given by
> $\sum_{h=1}^H ||v_h||_{\Lambda_h^{-1}} $
>
> $= \sum_{h=1}^H  \sqrt{\sum_{s_h,a_h} \frac{d_h^\pi(s_h)^2 \pi(a_h|s_h)^2}{d_h^{\bar\pi}(s_h) \bar\pi(a_h|s_h)} \cdot \mathbb{V_h} V_{h+1}^\pi(s,a)}$.
>
>
> Compared with the bound in Theorem 3.1 in [Yin & Wang 20], our bound seems to have an extra $\sqrt{H}$ factor (which can be shown by Cauchy-Schwarz inequality).
> We think the reason is that the above representation of the tabular MDP as a linear MDP using the canonical basis is not the most efficient way.

---

### Official Review · Reviewer_ASrb · 2021-07-16

**Rating:** 6
**Confidence:** 4

**Summary:**

This paper studies the off-policy evaluation (OPE) problem in reinforcement learning with linear function approximation. They propose to incorporate the variance information of the value function to improve the sample efficiency of OPE. This has better theoretical properties.

**Main Review:**

1.As for the final variance estimator \hat{\sigma}_h, why use a mechanism similar to clip? And using it, we can get \hat{\sigma}_h>1, this is very different from variance. so what is the motivation to set such range?
2. The paper said "Theorem 4.1 suggests that Algorithm 1 provably achieves a tighter instance-dependent error bound for OPE than that in [10]."  But from formula (B.1), the dominant term in error bound is not better than [10]. And What about the rest of error bound?
3. in experiments, From Fig.1. ,  the variance of the proposed methods is very large. In general, the larger the variance, the more unstable the OPE error.

**Time Spent Reviewing:**

5

---

> ### Author Response · Authors · 2021-08-10
> **Response to Reviewer ASrb**
>
> Thank you for your positive comments!
>
> ---
> **Q1**: “As for the final variance estimator $\hat{\sigma}_h$, why use a mechanism similar to clip? And using it, we can get $\hat{\sigma}_h>1$, this is very different from variance. So what is the motivation?”
>
> **A1**: The purpose of the clipping max{1, $\cdot$ } is to avoid dividing by a quantity which is close to zero, so that in Algorithm 1 it will not blow up when reweighting data points by the inverse of $\hat{\sigma}_h$. The additional “+1” under the square root is used to account for the variance of the reward function (note that the reward function is also random in our paper. So the randomness of the value function comes from both the transition and the reward). Please refer to Appendix C.1 in the supplementary material for a more detailed explanation.
>
> ---
> **Q2**: “The paper said "Theorem 4.1 suggests that Algorithm 1 provably achieves a tighter instance-dependent error bound for OPE than that in [10]." But from formula (B.1), the dominant term in error bound is not better than [10]. And What about the rest of the error bound?”
>
> **A2**: This seems to be a misunderstanding. We apologize if (B.1) causes a confusion. Note that the formula (B.1) is derived when the variance of the value function $\mathbb{V}V_h$ is of order $(H-h-1)^2$, which is the worst case. So (B.1) indicates that even in the worst case, our error bound is no worse than that in [10]. To see why our error bound can be much better than that in [10] (by a factor of $H$), please refer to the rest of Appendix B.2 for more details.
>
> ---
> **Q3**: “In experiments, from Fig.1. , the variance of the proposed methods is very large. In general, the larger the variance, the more unstable the OPE error.”
>
> **A3**: Actually, our variance is roughly of the same order as that of FQI-OPE. It looks much larger because in Figure 1 the y-axis is log-scaled (base 10).
>
> In Figure 1 (a), we see that the upper confidence interval of VA-OPE is just slightly wider than that of FQI-OPE. The lower confidence interval looks a lot wider because of the log-scale.
> In Figure 1 (b) and (c), when $H$ gets larger, VA-OPE begins to show its advantage. In (b), we see that both the upper and lower confidence bounds of VA-OPE are smaller than those of FQI-OPE. Also, because of the log-scale, the seemingly wide confidence interval of VA-OPE is actually in the same order as that of FQI-OPE, at least for all moderately large $K$.
> In (c), the upper confidence bound of VA-OPE is even smaller than the lower confidence bound of FQI-OPE.
> Actually, if we plot without the log-scaling, then one would see that the variance of VA-OPE is almost the same as that of FQI-OPE in (b) and (c), and is only slightly larger in (a). Also, it is important to note that the average error curve of VA-OPE is almost always below that of FQI-OPE.
>
> We are sorry for the confusion caused by the log-scaling. But please note that the plot of error rate versus $K$ would be otherwise cluttered without such scaling.

---

### Official Review · Reviewer_PGyL · 2021-07-18

**Rating:** 6
**Confidence:** 3

**Summary:**

The paper focuses on OPE for linear MDPS in an offline setup. The proposed algorithm accounts for the value-function variance in the dataset, and bounds the resulting value error as a function of that variance.

**Limitations And Societal Impact:**

Irrelevant.

**Main Review:**

The paper is written fairly well, though there are several grammar errors or typos. As an example, line 85: "there exists \gamma_h and \mu_h...". Nonetheless, I appreciate the idea of using the variance to improve data efficiency. The results seem like they make sense. But, specifically, I have the following several questions to the authors:
1. Why exactly do you split the data? In Sec. 2 you suggest this would be explained in Sec. 3, but I didn't find such an explanation.
2. The expectation of the weighted covariance matrix in (2.6) is taken w.r.t. \tilde{\pi}. If I understand correctly, it means that directly using the data to estimate it is straightforward. However, the variance inside that equation is taken w.r.t. \pi. How is that done? I can't find a place in the text or algorithm which show how you reweigh the variance to match the target distribution.
3. I don't understand how the norm in Remark 4.3 characterizes the distribution shift. I guess this relates to my previous question.

After I understand the answers to the above questions I can decide whether I think this paper should be published or not. In other words, I'm open to change my score either positively or negatively.

**Time Spent Reviewing:**

3

---

> ### Author Response · Authors · 2021-08-10
> **Response to Reviewer PGyL**
>
> Thank you for your feedback!
>
> ---
> **Q1**: “Why exactly do you split the data? In Sec. 2 you suggest this would be explained in Sec. 3, but I didn't find such an explanation.”
>
> **A1**: We split the data because the datasets $D$ and $\check{D}$ will be used for two different purposes in Algorithm 1 which we have explained in Section 3.
> Note that at the beginning of Section 3, we define the notations $\phi_{k,h}$ and $\check\phi_{k,h}$, where $\phi_{k,h}$ is computed using the dataset $D$ and $\check\phi_{k,h}$ is computed using the dataset $\check{D}$. Then in Lines 3 to 5 of Algorithm 1, $\check\phi$ is used to estimate the variance parameters $\theta_h$ and $\beta_h$, whereas in Lines 7 to 9 the dataset $D$ is used to estimate the model parameter $w_h^\pi$ by weighted regression.
> To sum up, the two data splits are used for two separate estimation procedures in Algorithm 1.
>
> Thanks for pointing this out. In the revision, we will add a few sentences in the beginning of Section 3 to make it clearer.
>
> ---
> **Q2**: “The expectation of the weighted covariance matrix in (2.6) is taken w.r.t. $\bar{\pi}$. If I understand correctly, it means that directly using the data to estimate it is straightforward. However, the variance inside that equation is taken w.r.t. $\pi$. How is that done? I can't find a place in the text or algorithm which shows how you reweigh the variance to match the target distribution.”
>
> **A2**: You are correct. The expectation of the weighted covariance matrix in (2.6) is indeed taken w.r.t. $\bar{\pi}$ and the variance inside that equation is indeed defined w.r.t. $\pi$.
>
> However, there is no contradiction between these two expectations. This is because: here the outer expectation is over the $(s,a)$ pair generated by executing the behavior policy $\bar{\pi}$. Although the definition of the function $\sigma_h(\cdot,\cdot)$ depends on the target policy $\pi$, it is still a deterministic function and the $(s,a)$ pair has nothing to do with $\pi$.
> Since the function $\sigma_h(\cdot,\cdot)$ is unknown, we then estimate it using the dataset $\check{D}$. The details of the estimation are explained in Line 177 to 189.
>
> ---
>
> **Q3**: “I don't understand how the norm in Remark 4.3 characterizes the distribution shift. I guess this relates to my previous question.”
>
> **A3**: The norm in Remark 4.3 characterizes the distribution shift because:
> By definition the vector $v_h^\pi$ is the expectation of $\phi(s_h,a_h)$ under the occupancy measure induced by the target policy $\pi$. In contrast, the matrix $\Lambda_h$ is the expectation of  $\phi(s_h,a_h)\phi(s_h,a_h)^\top$ under the occupancy measure induced by the behavior policy $\bar\pi$, further weighted by $\sigma^2(s_h,a_h)$.
>
> The intuition is that, if the two distributions differ by a lot, then the contribution in the norm of $\Lambda_h$ from the direction of $v_h^\pi$ will be small, and consequently, $||v_h^\pi||_{\Lambda_h^{-1}}$ would be large, which indicates a large distribution shift. This is not a rigorously defined distance between the induced distributions of the behavior policy and the target policy, but a very intuitive surrogate of the distribution mismatch.
>
> ---
> We hope the above response would address your questions.

---

### Decision · Program_Chairs · 2021-09-27

**Decision:**

Accept (Poster)

**Comment:**

This paper studies off-policy evaluation in linear MDPs. The authors extend the FQI work of [10] by reweighing the Bellman residual by an estimate of the value function variance. This slightly improves the instance-dependent sample efficiency, by up to a factor of H.
The paper is fairly incremental compared to [10] and of somewhat limited scope, but executed well overall to warrant acceptance.